# Human mobility networks reveal increased segregation in large cities

Hamed Nilforoshan[1,7], Wenli Looi[1,7], Emma Pierson[2,7], Blanca Villanueva[3], Nic Fishman[1], Yiling Chen[1], John Sholar[1], Beth Redbird[4,5], David Grusky[6] & Jure Leskovec[1 ✉]

A long-standing expectation is that large, dense and cosmopolitan areas support socioeconomic mixing and exposure among diverse individuals[1–6]. Assessing this hypothesis has been difficult because previous measures of socioeconomic mixing have relied on static residential housing data rather than real-life exposures among people at work, in places of leisure and in home neighbourhoods[7,8]. Here we develop a measure of exposure segregation that captures the socioeconomic diversity of these everyday encounters. Using mobile phone mobility data to represent 1.6 billion real-world exposures among 9.6 million people in the United States, we measure exposure segregation across 382 metropolitan statistical areas (MSAs) and 2,829 counties. We find that exposure segregation is 67% higher in the ten largest MSAs than in small MSAs with fewer than 100,000 residents. This means that, contrary to expectations, residents of large cosmopolitan areas have less exposure to a socioeconomically diverse range of individuals. Second, we find that the increased socioeconomic segregation in large cities arises because they offer a greater choice of differentiated spaces targeted to specific socioeconomic groups. Third, we find that this segregation-increasing effect is countered when a city's hubs (such as shopping centres) are positioned to bridge diverse neighbourhoods and therefore attract people of all socioeconomic statuses. Our findings challenge a long-standing conjecture in human geography and highlight how urban design can both prevent and facilitate encounters among diverse individuals.

In the United States, economic segregation is very high, with income affecting where one lives[9], who one marries[10], and who one meets and befriends[11]. This extreme segregation is costly. It reduces economic mobility[12–15], fosters a wide range of health problems[16–18] and increases political polarization[19–22]. Although there are all manner of reforms designed to reduce economic segregation (such as subsidized housing), it has long been argued that one of the most powerful segregation-reducing dynamics is rising urbanization[23] and the resulting happenstance mixing that it induces[1–6]. This 'cosmopolitan mixing hypothesis' anticipates that, in large cities, the combination of increased population diversity, constrained space and accessible public transportation will bring diverse individuals into close physical proximity with one another[2], reducing everyday socioeconomic segregation. The New York City Subway has been lauded, for example, as a mixing bowl in which a diverse set of people cross paths each day[24].

As plausible as the cosmopolitan mixing hypothesis might seem, big cities also provide new opportunities for self-segregation, because they are large enough to enable people to seek out and find others who are similar to themselves[25]. These contrasting hypotheses about the relationship between urbanization and socioeconomic mixing remain untested because it has been difficult to measure real-world exposures that take the form of path crossings and encounters among individuals[7,26,27]. It becomes possible to measure such exposures when mobile phone geolocation data are analysed at the device level. Although mobile phone data have been used for many research purposes[28–38], a nationwide study of socioeconomic mixing and urbanization has not been undertaken because of difficulties in ascertaining individual-level socioeconomic status (SES), determining when dyadic exposures occur, and amassing the data needed to compare across cities or counties[28–30,32–36].

Here we carefully test the cosmopolitan mixing hypothesis and the dynamics underlying it. To assess this hypothesis and understand the relationship between urbanization and segregation, we use mobile phone mobility data in the form of de-identified GPS location pings (see the 'SafeGraph' section of the Methods). From this data, we capture geolocated individual-level exposures between individuals of similar or different SES. This enables us to develop city-level and county-level measures of segregation that capture where people go, when they go there and whom they encounter on the way.

We first determine the SES of a person by identifying their home location and its monthly rent value. We next construct a dynamic network that captures each individual's exposures to other individuals in their everyday life. Our network contains 1,570,782,460 edges (representing exposures in physical space) among 9,567,559 nodes (representing

[1]Department of Computer Science, Stanford University, Stanford, CA, USA. [2]Department of Computer Science, Cornell Tech, New York, NY, USA. [3]Department of Biomedical Data Science, Stanford University, Stanford, CA, USA. [4]Institute for Policy Research, Northwestern University, Evanston, IL, USA. [5]Department of Sociology, Northwestern University, Evanston, IL, USA. [6]Department of Sociology, Stanford University, Stanford, CA, USA. [7]These authors contributed equally: Hamed Nilforoshan, Wenli Looi, Emma Pierson. ✉e-mail: jure@cs.stanford.edu

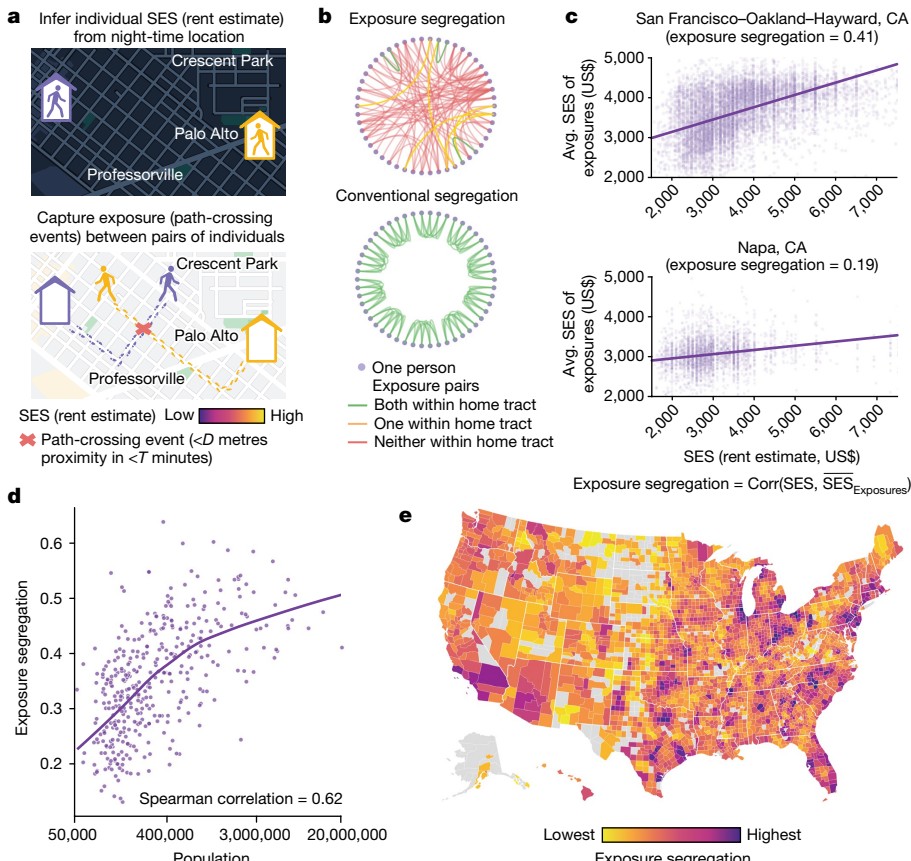

**Fig. 1 | Exposure segregation captures the likelihood of exposure between people of different socioeconomic backgrounds and reveals increased segregation in highly populated metropolitan areas. a**, For 9.6 million individuals (mobile phones), we infer their SES (rent or rent equivalent) from their home address on the basis of their location at night (see the 'Inferring home location' section of the Methods). We then capture path-crossing events (that is, being at the same location at the same time) to identify pairs of individuals who were exposed to each other (see the 'Constructing exposure network' section of the Methods). **b**, The nationwide network of 1.6 billion exposures spans 2,829 counties and 382 MSAs. Our exposure network contrasts with a conventional measure of economic segregation, the neighbourhood sorting index, which assumes that individuals are exposed to other residents only within their home census tract. Graphs pertain to a sample community of 50 individuals residing in ten census tracts in San Francisco, CA. Nodes represent individuals; edges represent exposures. This sample illustrates the importance of capturing cross-tract exposures, which are undetected by conventional segregation measures. **c**, For each geographical region (either MSA or county), we estimate exposure segregation, defined as the correlation between an individual's SES and the mean SES of those with whom they cross paths; 1 signifies perfect segregation and 0 signifies no segregation. This definition is equivalent to the conventional neighbourhood sorting index, but with the key difference that it leverages real-life exposure from mobility data instead of synthetic exposures

from individuals grouped by census tracts. For two MSAs, we show the raw data; each point represents one individual. San Francisco–Oakland–Hayward, CA, is 2.2× more segregated ($P < 10^{-4}$, 95% CI = 1.6–2.8×; two-sided bootstrap; see the 'Hypothesis testing' section of the Methods) than Napa, CA. **d,e**, Contrary to the hypothesis that highly populated metropolitan areas support diverse exposures and socioeconomic mixing, we find that larger MSAs are more segregated (**d**). Exposure segregation presented as a function of population size; each dot represents one MSA; the purple line indicates the LOWESS fit. An upward slope reveals that urbanization is associated with higher exposure segregation (Spearman correlation = 0.62, $n = 382$, $P < 10^{-4}$; two-sided Student's $t$-test; see the 'Hypothesis testing' section of the Methods). The top ten largest MSAs by population size are 67% more segregated ($P < 10^{-4}$, 95% CI = 49–87%; two-sided bootstrap; see the 'Hypothesis testing' section of the Methods) than small MSAs with fewer than 100,000 residents. Associations are robust to controlling for potential confounding factors and are similar for population density and exposure segregation (Extended Data Table 1 and Supplementary Table 7). **e**, Exposure segregation across the 2,829 US counties. The analysis was limited to counties with at least 50 individuals present in the dataset. Exposure segregation varies substantially across counties in the United States. Moreover, as with MSA-level segregation, county-level exposure segregation is also positively associated with both population size and population density (Extended Data Fig. 4).

individuals, that is, mobile phones) across 382 MSAs and 2,829 counties in the United States. Every timestamped edge between a pair of nodes signifies that the two individuals crossed paths with and encountered each other (that is, they were at the same location at the same time). We analysed these data to estimate the amount of exposure segregation, defined as the extent to which individuals of different economic statuses are exposed to one another within each geographical area (MSAs and counties) in the United States. Our measure of exposure segregation extends a traditional static segregation measure by capturing the diversity of person-to-person exposures localized in space and time.

## A more realistic measure of segregation

To estimate each person's SES, we first infer their home location from night-time mobile phone location pings (Fig. 1a; see the 'Inferring home location' section of the Methods), and we then recover the estimated monthly rent value of the home at this location (Fig. 1a; see the 'Inferring SES' section of the Methods). This method is more accurate in estimating individual SES than the conventional approach of using neighbourhood-level census averages[30,31]. We next identify each instance when a pair of individuals crossed paths and were thus exposed to each other, defined as their two devices being within

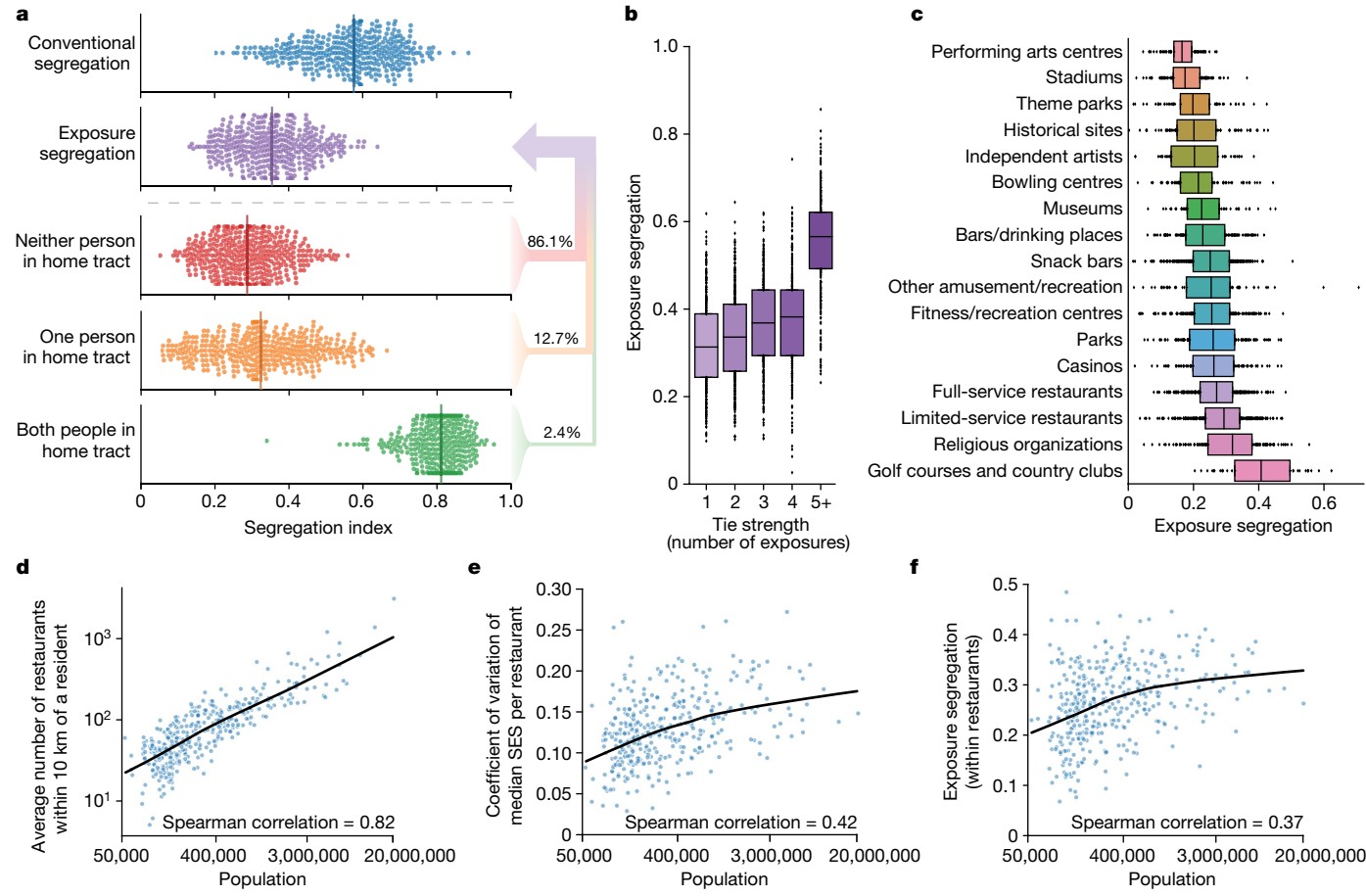

**Fig. 2 | Exploring the dynamics of exposure segregation reveals that socioeconomic differentiation of spaces accounts for increased segregation in large cities. a**, Each point represents the segregation estimate in one of the $n = 382$ MSAs; the vertical coloured lines represent the median across MSAs. Top, exposure segregation is 38% lower ($P < 10^{-4}$, 95% CI = 37–41%; two-sided bootstrap; see the 'Hypothesis testing' section of the Methods) than the conventional segregation measure—the neighbourhood sorting index. Bottom, a breakdown of exposure segregation into its component parts. Exposures in which both people are within their home census tract (green) are most segregated, reflecting the homophily effect in which people preferentially encounter those of a similar SES in their home tracts. Out-of-tract exposures (orange and red) are less segregated, reflecting the visitor effect in which entering other tracts exposes individuals to economically diverse individuals. As a small minority (2.4%, 95% CI = 2.4–2.4%; two-sided bootstrap; see the 'Hypothesis testing' section of the Methods) of exposures happen within the home tract, the visitor effect dominates the homophily effect and exposure segregation is therefore lower than the conventional neighbourhood sorting index. **b**,**c**, Exposure segregation varies by tie strength and location type. Each point represents segregation in one of $n = 382$ MSAs using only exposure pairs occurring with a specific tie strength (**b**) or in a given location type (**c**). The boxes indicate the interquartile range across MSAs. Segregation increases with tie strength and is especially high for the strongest ties (5+ exposures; median exposure segregation, 0.57). Segregation is highest at golf courses and country clubs (median exposure segregation, 0.42) and lowest at performing arts centres (median exposure segregation, 0.16) and stadiums (median exposure segregation, 0.17). **d**–**f**, A case study of full-service restaurants illustrates the relationship between urbanization and exposure segregation. Highly populated metropolitan areas are more segregated not only because they offer a wider choice of venues but also because these venues are more socioeconomically differentiated. **d**, Larger MSAs have more restaurants within 10 km of the average resident, giving residents more options to self-segregate. **e**, Moreover, restaurants in larger MSAs vary more in the median SES of their visitors, meaning that a greater choice of socioeconomically differentiated restaurants is offered. The coefficient of variation across restaurant SES (that is, the median SES of a restaurant's visitors) in the ten largest MSAs is 63% more ($P < 10^{-4}$, 95% CI = 37–100%; two-sided bootstrap; see the 'Hypothesis testing' section of the Methods) than the coefficient of variation in small MSAs (with fewer than 100,000 residents). **f**, Consequently, exposure segregation within restaurants is higher in larger MSAs. These relationships are also detectable at the scale of city hubs (defined as higher-level clusters of POIs such as plazas and shopping centres) as well as at the neighbourhood level (Extended Data Figs. 5 and 6).

$D$ metres of each other within $T$ minutes (see the 'Constructing exposure network' section of the Methods). Although our key findings are robust to the precise choice of $D$ and $T$ (Supplementary Figs. 5–8), our primary analyses use $D = 50$ metres and $T = 5$ minutes because the cosmopolitan mixing hypothesis pertains to visual exposure[1,2]. This approach, to our knowledge, provides the highest-resolution measure of exposure to date, compared with previous GPS-based studies[30,31,39].

The economic segregation of each geographical region is measured by the correlation between a person's SES and the mean SES of everyone to whom they are exposed through a path crossing (see the 'Exposure segregation' section of the Methods). This correlation is estimated by fitting a linear mixed-effects model that eliminates attenuation bias and secures unbiased estimates of exposure segregation even when observed exposures are sparse (Extended Data Fig. 1; see the 'Estimating exposure segregation' section of the Methods). The resulting measure of exposure segregation (Fig. 1b,c), which ranges from 0 (perfect integration) to 1 (complete segregation), is a generalization of a widely used measure of socioeconomic segregation—the neighbourhood sorting index[7]. The neighbourhood sorting index is equivalent to the correlation between each person's SES and the mean SES of all of the people in their home census tract, whereas our measure of exposure segregation is equivalent to the correlation between each person's SES

and the mean SES of all of the people who they encounter (either inside or outside their home census tract). Thus, the key difference between these two measures is that the neighbourhood sorting index assumes that exposures occur uniformly and only among co-residents of the same home tract, whereas exposure segregation captures real-world exposures among people as they navigate their daily lives.

## Extreme segregation in large cities

We find that, contrary to the cosmopolitan mixing hypothesis, exposure segregation is higher in large MSAs (Fig. 1d). The Spearman correlation between MSA population and MSA segregation is 0.62 ($P < 10^{-4}$), and the ten largest MSAs by population size are 67% more segregated ($P < 10^{-4}$, 95% confidence interval (CI) = 49–87%) than small MSAs with fewer than 100,000 residents. This result is robust. We validated it by recalculating the correlation with a measure of density rather than population size (Spearman correlation = 0.45, $P < 10^{-4}$; Supplementary Table 7), by controlling for potential confounding factors (Extended Data Table 1 and Supplementary Table 7), by varying the granularity of the analysis (Fig. 1e and Extended Data Fig. 4) and by testing a variety of specifications of exposure segregation (Supplementary Table 6 and Supplementary Figs. 2–10). The consistent result that larger, denser cities are more segregated runs counter to the hypothesis that such cities promote socioeconomic mixing by attracting diverse individuals and constraining space in ways that oblige them to encounter one other[1–6]. Our results support the opposite hypothesis: big cities allow their inhabitants to seek out people who are more like themselves. The key advance that enables this finding is our fine-grained measure of proximity with respect to both time and space (Supplementary Fig. 66).

## Exploring exposure segregation

Our methodology further allows for comparisons between a conventional static measure of segregation (neighbourhood sorting index) and our dynamic measure. The median level of exposure segregation across all MSAs is 38% lower ($P < 10^{-4}$, 95% CI = 37–41%) than the corresponding value for a conventional static estimate[31] (neighbourhood sorting index; Fig. 2a (top)). We explain this result by disaggregating our measure into components pertaining to exposures in which both, one or neither individual was within their home census tract (Fig. 2a (bottom)). Exposure segregation is lower because, when people venture outside their home tracts, they experience more diversity. For example, exposures are 50% less segregated ($P < 10^{-4}$, 95% CI 48–53%) when both people are outside the home census tract than when both people are within their home tract. Within their own neighbourhood, people cross paths with neighbours who are socioeconomically most similar to them, but this has little effect on overall exposure segregation because only 2.4% of exposures (95% CI = 2.4–2.4%) occur when both individuals are within their home tract. Finally, we observe that not only is overall exposure segregation elevated in large cities, but also each of its components is elevated in large cities (Supplementary Fig. 10).

We quantify variability in exposure segregation both by tie strength (Fig. 2b,c) and across different points of interest (POIs). Stronger ties are more segregated[40,41] (Fig. 2b). We also find much variability in POI-level segregation[11,30] (Fig. 2c; see the 'Decomposing segregation by activity' section of the Methods). We explain this variability in POI-level segregation (Fig. 2c) by the extent to which a POI category (such as restaurants) contains differentiated POIs that service small and thereby socioeconomically homogeneous communities (for example, Michelin star restaurants). We operationalize the extent of a POI category's differentiation using the average travel distance to the nearest POI[30] and the total number of POIs (Spearman correlation = −0.75, $P < 0.001$ (travel distance); Spearman correlation = 0.69, $P < 0.01$ (number of POIs); Extended Data Fig. 3a,b). For example, in the median MSA, religious organizations require 92% less travel distance ($P < 10^{-4}$, 95% CI = 92–93%)

and are 16× more numerous ($P < 10^{-4}$, 95% CI = 8–18×) than stadiums. Because religious organizations can therefore target more narrowly defined socioeconomic communities, they are 75% more segregated ($P < 10^{-4}$, 95% CI = 58–87%) than stadiums. In rare cases, a POI category with only a small number of POIs may still exhibit substantial segregation (such as golf courses) owing to economic differentiation among its POIs caused by other factors (such as a public–private distinction; Extended Data Fig. 3c). Below, we show that this link between the socioeconomic differentiation of spaces and segregation is also critical to explaining why large cities are more segregated.

## Differentiation of space in large cities

To understand why large metropolitan areas support segregation, we present an example of segregation within leisure POIs. Full-service restaurants provide an illustrative example (Fig. 2d–f) of a segregation-inducing dynamic that holds widely across other leisure sites (Supplementary Fig. 22) and other scales of analysis (Extended Data Figs. 5 and 6). We find that larger MSAs offer their residents a greater number of leisure choices: the average resident of one of the ten largest MSAs has 22× more restaurants ($P < 10^{-4}$, 95% CI = 11–39×) within 10 km of their home compared with an average resident of a small MSA (where a 'small MSA' is defined as one with fewer than 100,000 residents; Fig. 2d). These choices are also more socioeconomically differentiated. When a restaurant's SES is defined as the median SES of all people who visited it and encountered another person, the coefficient of variation of 'restaurant SES' in the ten largest MSAs is 63% greater ($P < 10^{-4}$, 95% CI = 37–100%) than that in small ones (Fig. 2e). Thus, not only do large MSAs offer their residents a larger choice of restaurants, but these restaurants are also more socioeconomically differentiated. For example, in large cities such as New York, one can spend US$10, US$100 or US$1,000 on a meal, depending on the choice of restaurant[42,43]. These processes mean that exposure segregation in restaurants is 29% higher ($P < 10^{-3}$, 95% CI 8–49%) in the ten largest MSAs than in small MSAs (Fig. 2f). We find analogous results across many POI types (Supplementary Fig. 22) and at higher levels of scale pertaining to city hubs (for example, plazas, shopping centres, boardwalks) as well as neighbourhoods (Extended Data Figs. 5 and 6).

## Mitigating segregation through urban design

Our results suggest that segregation could be mitigated when frequently visited POIs, which we refer to as 'hubs', are positioned in close proximity to diverse neighbourhoods. These hubs would serve as bridges between residents of nearby high-SES and low-SES neighbourhoods, enabling them to easily visit the hubs[44–46] and encounter one another (Fig. 3c). We developed the bridging index (see the 'Bridging index' section of the Methods) to measure whether hubs are located in such bridging positions. Our index measures the economic diversity of the groups that would encounter each other if everybody visited only their nearest hub. It is computed by clustering individuals by the nearest hub to their home and then measuring the economic diversity within these clusters (Extended Data Fig. 7). The resulting index ranges from 0 to 1, where 0 means that individuals near each hub have a uniform SES, and 1 means that individuals near each hub are as diverse as the overall area (Extended Data Fig. 8). We compute our bridging index for commercial centres (such as plazas, shopping centres, boardwalks) because we find that they are common hubs of exposure: the majority (56.9%, 95% CI = 56.9–56.9%) of exposures across all 382 MSAs occur in close proximity (within 1 km) to a commercial centre, even though only 2.5% of land area is within 1 km of a commercial centre (Fig. 3c). The results show that our bridging index is strongly associated with exposure segregation (Spearman correlation = −0.78, $P < 10^{-4}$; Fig. 3d). The top ten MSAs with the highest bridging index are 53.1% less segregated ($P < 10^{-4}$, 95% CI = 44–60%) than the ten MSAs with the lowest bridging

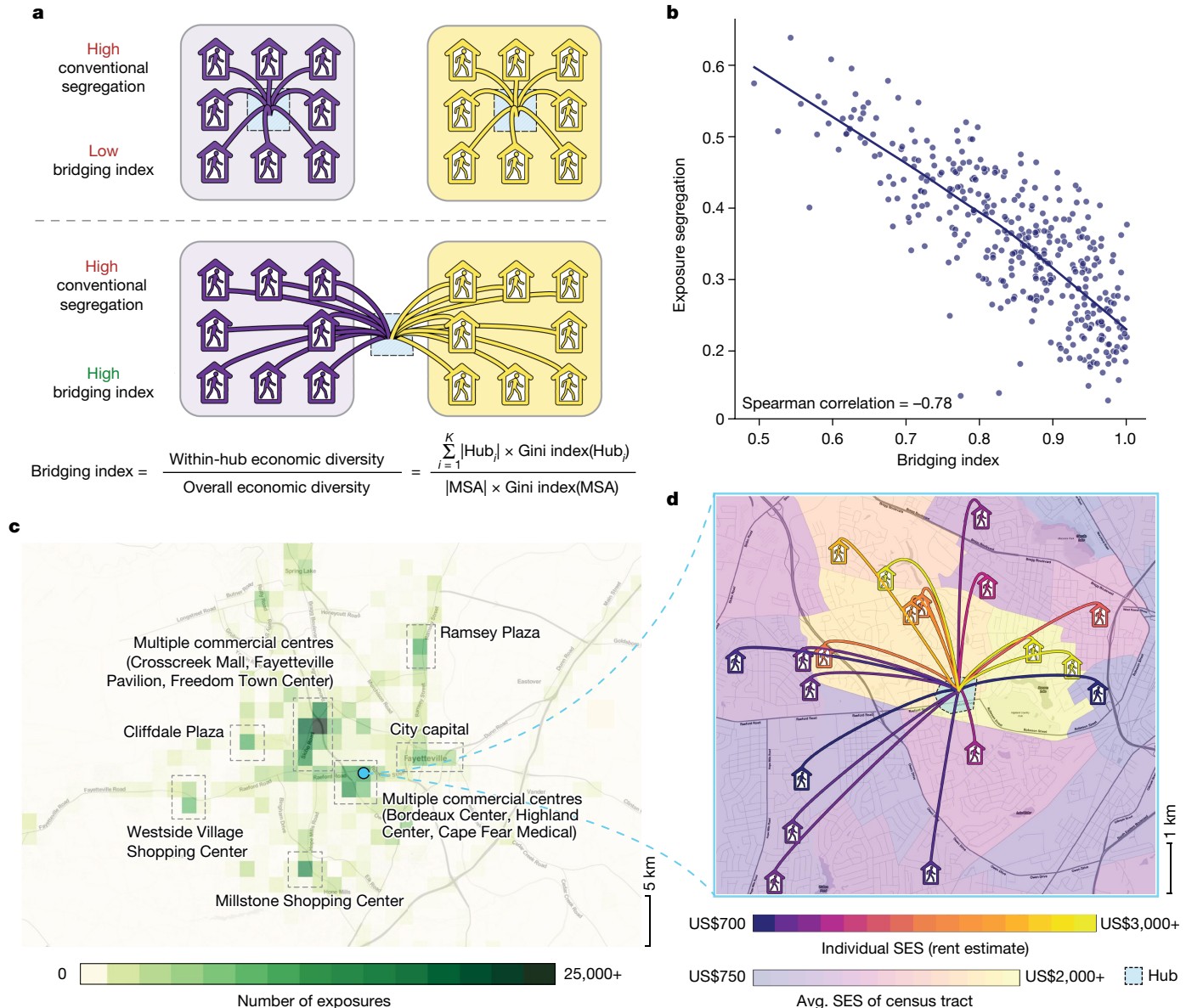

**Fig. 3 | Exposure segregation is lower when frequently visited hubs bridge socioeconomically diverse neighbourhoods. a**, We developed an index (see the 'Bridging index' section of the Methods) to quantify the extent to which highly visited hubs bridge socioeconomically diverse neighbourhoods. The metric was constructed by clustering homes by the nearest hub, then measuring the within-cluster diversity of SES. Two plots illustrate that the bridging index is distinct from conventional measures of residential segregation such as the neighbourhood sorting index. The bridging index ranges from 0 (no bridging; top) to 1 (perfect bridging; bottom), while residential segregation is constant (high-SES and low-SES individuals are highly segregated by census tract, denoted by purple and yellow bounding boxes). We compute our bridging index with hubs defined as commercial centres (such as shopping centres and plazas) because the majority (56.9%, 95% CI = 56.9–56.9%; bootstrapping; see the 'Hypothesis testing' section of the Methods) of exposures across all 382 MSAs occur in close proximity (within 1 km) to a commercial centre, even though only 2.5% of land area is within 1 km of a commercial centre. **b**, Our bridging index strongly predicts exposure segregation (Spearman correlation = −0.78, $n = 382$, $P < 10^{-4}$; two-sided Student's $t$-test; see the 'Hypothesis testing' section of the Methods). The top ten MSAs with the highest bridging index are 53.1% less segregated ($P < 10^{-4}$, 95% CI = 44–60%; two-sided bootstrap; see the

'Hypothesis testing' section of the Methods) than the ten MSAs with the lowest bridging index. The bridging index predicts segregation more accurately ($P < 10^{-4}$; two-sided Steiger's $Z$-test; see the 'Hypothesis testing' section of the Methods) than population size, SES inequality, neighbourhood sorting index and race, and is significantly associated ($P < 10^{-4}$; two-sided Student's $t$-test; see the 'Hypothesis testing' section of the Methods) with exposure segregation after controlling for these variables and other potential confounding factors (Extended Data Tables 2 and 3). **c,d**, A case study of Fayetteville, North Carolina, an MSA with low exposure segregation (21st percentile) despite having an above-median population size (64th percentile) and income inequality (60th percentile). **c**, Exposure heat map of Fayetteville; all visually discernible hubs are associated with one or more commercial centres. **d**, Hubs are located in accessible proximity to both high-SES and low-SES census tracts (bridging index = 0.90, 62nd percentile), leading to diverse exposures. An illustrative example of one hub (Highland Center) in Fayetteville and a random sample of ten exposures occurring inside of it. The home icons demarcate home locations of individuals (up to 100 m of random noise was added for anonymity); the colours denote individual and mean tract SES. The maps in **c** and **d** were generated using OpenStreetMap data.

index. This finding is again robust: the hub-bridging effect is strong and significant ($P < 10^{-4}$) even after including controls for race, population size, economic inequality and many other variables (Extended Data Tables 2 and 3, Supplementary Table 6 and Supplementary Figs. 2, 8 and 13). It follows that zoning laws and related policies that encourage developers to locate hubs, such as shopping centres, between diverse residential neighbourhoods may reduce exposure segregation. We have identified several large cities that increase integration in this manner (Supplementary Table 21) and present an illustrative example (Fig. 3c,d) in which well-placed hubs bridge diverse individuals in Fayetteville, North Carolina.

## Discussion

As big cities continue to grow and spread, it is important to examine whether they encourage socioeconomic mixing. Although it is often argued that big cities promote mixing by increasing density, we find that exposure diversity and city size are negatively related. This result means that scale matters. We have shown that, because large cities can sustain venues that are targeted to thin socioeconomic slices of the population, they have become homophily-generating machines that are far more segregated than small cities. We also find that some cities are able to mitigate this segregative effect because their hubs are located in bridging zones that can draw in people from diverse neighbourhoods. We were able to detect these pockets of homophily (and the counteracting effects of bridging hubs) because we have developed a dynamic measure of economic segregation that captures everyday socioeconomic mixing at home, work and leisure.

This new methodology for measuring exposure segregation, while an improvement over static approaches, has limitations. For example, it is difficult to ascertain how weak or strong the ties are, as we are obliged to use physical proximity as a proxy for exposure[47]. It is reassuring in this regard that our core results persist under stricter time, distance and tie-strength thresholds (Supplementary Table 6 and Supplementary Figs. 5–8), and are associated with key downstream outcomes (Extended Data Fig. 2 and Supplementary Fig. 24). It is likewise important to locate and analyse supplementary datasets that cover subpopulations (for example, subpopulations of homeless individuals) that are not as well represented in our dataset[48]. The available evidence indicates that our sample is well balanced on many key racial, economic and demographic variables[49], but mobile phone market penetration is still not complete, and GPS ping data are unevenly distributed by time. Finally, our measure of SES relies on housing consumption, an indicator that does not exhaust the concept of SES. It is again reassuring that our analytical approach, which improves on conventional neighbourhood-level imputations, is robust under a range of alternative measures of SES (Supplementary Fig. 3).

This is all to suggest that dynamic segregation data are rich enough to overcome many seeming limitations. The dynamic approach that we have taken here could further be extended to examine cross-population differences in the sources of segregation and to develop a more complete toolkit of approaches for reducing segregation and improving urban design.

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

# Methods

The Methods is structured as follows. In the 'Datasets' section, we explain the datasets used in our analysis; in the 'Data processing' section, we explain the data processing procedures that we use to infer SES and exposures; and in the 'Analysis' section, we explain the analyses underlying our main results.

## Datasets

**SafeGraph.** Our primary mobility and location data comprise GPS locations from a sample of adult smartphone users in the United States, provided by the company SafeGraph. The data are de-identified GPS location pings from smartphone applications that are collected and transmitted to SafeGraph by participating users[50]. As described by SafeGraph in the public documentation, SafeGraph data are collected by "partner[ing] with mobile applications that obtain opt-in consent from users to collect anonymous location data. This data is not associated with any name or email address". SafeGraph ensures that its mobile application partners obtain consent for data to be used for commercial and research purposes, including academic publication. SafeGraph users are able to opt out of data collection at any time.

Although the sample is not random, previous work has demonstrated that SafeGraph data are geographically well balanced (that is, an approximately unbiased sample of different census tracts within each state) and well balanced along the dimensions of race, income and education[49,51]. Furthermore, SafeGraph data are a widely used standard in large-scale studies of human mobility across many different areas including COVID-19 modelling[51], political polarization[39] and consumer preference tracking[52]. All data provided by SafeGraph were stored on a secure server behind a firewall. Data handling and analysis was conducted in accordance with SafeGraph policies and in accordance with the guidelines of the Stanford University Institutional Review Board.

The raw data consist of 91,755,502 users and 61,730,645,084 pings from three evenly spaced months in 2017: March, July and November. Each ping consists of a latitude, longitude, timestamp, and de-identified user ID. The mean number of raw pings associated with a user is 667 and the median number of pings is 12. We applied several filters to improve the reliability of the SafeGraph data, and subsequently linked each user to an estimated rent (that is, Zillow Zestimate) using their inferred home location (that is, CoreLogic address), as described in the 'Inferring home location' and 'Inferring SES' sections.

We applied several filters to improve the reliability of the SafeGraph data. To ensure that the locations are reliable, we excluded pings with location estimates less accurate than 100 m, as recommended by SafeGraph[53]. We filtered out users with fewer than 500 pings, as these are largely noise. We also filtered out users for whom we were unable to infer a home, because we rely on home rent values to measure SES. Finally, to avoid duplicate users, we removed users if more than 80% of their pings had identical latitudes, longitudes and timestamps to those of another user; this could potentially occur if, for example, a single person in the real world carries multiple mobile devices. After these initial filters, we were able to infer home locations for 12,183,523 users in the United States (50 states and Washington DC), leveraging the CoreLogic database. Of users for whom we could infer a home location, we were able to successfully link 9,576,650 to an estimated rent value through the Zillow API. The 'Inferring home location' and 'Inferring SES' sections provide full details on the use of CoreLogic database to infer home locations and the use of the Zillow API to link these home locations to estimated rent values. Finally, after removing users for whom >80% of their pings were duplicates with another user, we reduced the number of users from 9,576,650 to 9,567,559 (that is, we removed less than 0.1% of users through de-duplication).

**CoreLogic.** We use the CoreLogic real estate database to link users to home locations[54]. The database provides information covering over 99% of US residential properties (145 million properties), over 99% of commercial real estate properties (26 million properties) and 100% of US county, municipal and special tax districts (3,141 counties). The CoreLogic real estate database includes the latitude and longitude of each home, in addition to its full address: street name, number, county, state and zip code.

**Zillow.** We used the Zillow property database to query for rent estimates[55] (our primary measure of SES). The Zillow database contains rent data (rent Zestimate) for 119 million US residential properties. We were able to determine a rent Zestimate, the primary measure of SES used in our analysis, for 9,576,650 out of 12,183,523 inferred SafeGraph user homes (a 79% hit rate).

**SafeGraph Places.** Our database of US business establishment boundaries and annotations comes from the SafeGraph Places database[50], which indexes the names, addresses, categories, latitudes, longitudes and geographical boundary polygons of 5.5 million US POIs in the United States. SafeGraph includes the North American Industry Classification System (NAICS) category of each POI, which is standard taxonomy used by the Federal government to classify business establishments[56]. For example, the NAICS code 722511 indicates full-service restaurants. We identified relevant leisure sites using the prefix 7, which includes arts, entertainment, recreation, accommodation and food services, and supplemented these POIs with the prefix 8131 to include religious organizations such as churches. We restricted our analysis of leisure sites to the top-most frequently visited POI categories within these NAICS code prefixes (Fig. 2c): full-service restaurants, snack bars, limited-service restaurants, stadiums and so on. SafeGraph Places also includes higher-level 'parent' POI polygons that encapsulate smaller POIs. Specifically, we identified hubs with the NAICS code 531120 (lessors of non-residential real estate), which we find in practice corresponds to commercial centres such as shopping centres, plazas, boardwalks and other clusters of businesses. We provide illustrative examples of such hubs in Supplementary Figs. 16–18.

**US census.** We extracted demographic and geographical features from the five-year 2013–2017 American Community Survey[57]. This enables us, as described below, to link mobile phone locations to geographical areas including census block group (CBG), census tract and MSA, as well as to infer demographic features corresponding to those demographic areas including median household income.

A CBG is a statistical division of a census tract. CBGs are generally defined to contain between 600 and 3,000 people. A CBG can be identified at the national level by the unique combination of state, county, tract and block group codes.

A census tract is a statistical subdivision of a county containing an average of around 4,000 inhabitants. Census tracts range in population from 1,200 to 8,000 inhabitants. Each tract is identified by a unique numeric code within a county. A tract can be identified at the national level by the unique combination of state, county and tract codes.

Census tracts and block groups typically cover a contiguous geographical area, although this is not a constraint on the shape of the tract or block group. Census tract and block group boundaries generally persist over time so that temporal and geographical analysis is possible across multiple censuses.

Most census tracts and CBGs are delineated by inhabitants who participate in the Census Bureau's Participant Statistical Areas Program. The Census Bureau determines the boundaries of the remaining tracts and block groups when delineation by inhabitants, local governments or regional organizations is not possible[58].

An MSA is a US geographical area defined by the Office of Management and Budget (OMB) and is one of two types of Core Based Statistical Area (CBSA). A CBSA comprises a county or counties associated with a core urbanized area with a population of at least 10,000 inhabitants and

adjacent counties with a high degree of social and economic integration with the core area. Social and economic integration is measured through commuting ties between the adjacent counties and the core. A micropolitan statistical area is a CBSA of which the core has a population of between 10,000 and 50,000; an MSA is a CBSA of which the core has a population of over 50,000. In our primary analysis, we follow a previous study[31] and focus on MSAs, excluding micropolitan statistical areas owing to data sparsity concerns.

**TIGER.** Road and transportation feature annotations come from the census-curated Topologically Integrated Geographic Encoding and Referencing system (TIGER) database[59]. The TIGER databases are an extract of selected geographical and cartographic information from the US Census Bureau's Master Address File/Topologically Integrated Geographic Encoding and Referencing (MAF/TIGER) Database (MTDB). We used the MAF/TIGER Feature Class Code (MTFCC) from the TIGER Roads and TIGER Rails databases to identify road and railways. TIGER data are in the format of Shapefiles, which provide the exact boundaries of roads and railways as latitude/longitude coordinates.

## Data processing
For each individual, we first infer their home location and subsequently estimate their SES on the basis of their home rent value (see the 'Inferring home location' and subsequently 'Inferring SES' sections). We then calculate all exposures between individuals (see the 'Constructing exposure network' section). We then annotate exposures according to the location in which they occurred. Specifically, we annotate whether the exposure took place in both individuals' home tract, in one individual's home tract, or in neither home tract. We also determine whether it occurred inside a fine-grained POI, such as a specific restaurant, as well as whether it took place within a parent POI, like a hub (see the 'Annotating exposures' section). Details on all inferences and exposure calculations are provided below.

**Inferring home location.** We first infer a user's home latitude and longitude using the latitude and longitude coordinates of their pings during local night-time (and early-morning) hours, based on best practices established by SafeGraph[60]. We first remove users with fewer than 500 pings to ensure that we have enough data to reliably infer home locations. We then interpolate each person's location for each 1 h window (for example, 18:00–19:00, 19:00–20:00 and 20:00–21:00) using linear interpolation of latitudes and longitudes to ensure that we have time series at a constant time resolution. We perform interpolation using the interpolate package of the scipy library. We filter for hours between 18:00 and 09:00 during which the person moves less than 50 m until the next hour; these stationary night-time (and early-morning) pings represent cases in which the person is more likely to be at home. We filter for users who have such stationary pings on at least three dates and with at least 60% of pings within a 50 m radius. Finally, we infer home latitude and longitude as the median latitude and longitude of these stationary pings (after removing outliers outside the 50 m radius). We choose the thresholds above because they yield a good compromise between inferring the home location of most users and inferring home locations with high confidence. Overall, we are able to infer home locations for 70% of users with more than 500 pings, and these locations are inferred with high confidence; 89% of stationary night-time observations are within 50 m of the inferred home latitude and longitude. Our key findings are robust to the exact choice of threshold for home identification (Supplementary Fig. 62).

**Inferring SES.** Having inferred each user's home location, we link their latitude and longitude to a large-scale housing database (Zillow) to infer the estimated rent of each individual's home, which we use as a measure of SES. We do this in two steps. First, we link the inferred user's home latitude and longitude to the CoreLogic property database (see the 'CoreLogic' section), a comprehensive database of properties in the United States, by taking the closest CoreLogic residential property (single family residence, condominium, duplex or apartment) to the user's inferred home latitude and longitude. Second, we use the CoreLogic address to query the Zillow database (see the 'Zillow' section), which provides an estimated home rent and price for each individual (the Zillow database does not allow for queries using raw latitude and longitude, which is why it is necessary to leverage to CoreLogic to obtain an address for each user). We use Zillow's estimated rent for the user's home as our main measure of SES. We apply several quality-control filters to ensure that the final set of individuals that we use in our main analyses have reliably inferred home locations and SES: (1) we remove a small number of users whose median latitude and longitude at home are identical to another user's, as we empirically observe that these people have unusual ping patterns; (2) we remove users for whom we are lacking a Zillow rent estimate, as this constitutes our primary SES measure; (3) we winsorize Zillow rent estimates that are greater than US$20,000 to avoid spurious results from a small number of outliers; (4) we remove a small number of users who are missing census demographic information for their inferred home location; (5) we remove users whose Zillow home location is further than 100 m from their CoreLogic home location, or whose CoreLogic home location is further than 100 m from their median latitude and longitude at home; (6) we remove a small number of users in single family residences who are mapped to the exact same single family residence as more than 10 other people, as this may indicate a data error in the Zillow database.

The set of users who pass these filters constitutes our final analysis set of 9,567,559 users. We confirm that the census demographic statistics of these users' inferred home locations are similar to those of the US population in terms of income, age, sex and race.

Any individual quantitative measure provides only a partial picture of a person's SES. Recognizing this, we conduct robustness checks in which, rather than using the Zillow estimated rent of the user's home as a proxy for SES, we use (1) the median CBG household income in that area; and (2) the percentile-scored rent of the home, to account for long-tailed rent distributions. Our main results are robust to using these alternative measures of SES (Supplementary Fig. 3).

**Constructing the exposure network.** We constructed a fine-grained, dynamic exposure network $\mathcal{G}$ between all 9,567,559 individuals across 382 MSAs and 2,829 counties, which is represented as an undirected graph $\mathcal{G} = (\mathcal{V}, \mathcal{E})$ with time-varying edges. Each node $v_i \in \mathcal{V}$ in the graph represents one of the $n = 9{,}567{,}559$ individuals in our study, such that the set of nodes is $V = \{v_1, v_2, ..., v_n\}$. Each node $v_i$ has a single attribute $x_i$, representing the inferred SES (estimated rent) of the individual.

Individuals $v_i$ and $v_j$ are connected by one edge $e_{i,j,k} \in \mathcal{E}$ per exposure, with $k$ indicating the $k$th exposure between individuals $v_i$ and $v_j$. Each edge $e_{i,j,k}$ has three attributes $t_{i,j,k}$, $\text{lat}_{i,j,k}$ and $\text{lon}_{i,j,k}$, indicating the timestamp, latitude and longitude of the exposure, respectively. We now focus our discussion on explaining how each of the exposure edges of the network is calculated.

We define an exposure to occur when two users have GPS pings that are close (according to a fixed threshold) in both physical proximity and time. Specifically if user $v_i$ has a GPS ping with $t_i$, $\text{lat}_i$, $\text{lon}_i$ (indicating the timestamp, latitude and longitude of the ping respectively), and user $v_j$ has a GPS ping with $t_j$, $\text{lat}_j$, $\text{lon}_j$, then the users are said to have crossed paths if $|t_i - t_j| < T$ and $\text{distance}((\text{lat}_i, \text{lat}_i), (\text{lat}_j, \text{lat}_j)) < D$, where $T$ represents the time threshold (that is, the maximum time distance the two pings can be apart to count as an exposure) and $D$ represents the distance threshold (that is, the maximum physical distance that the two pings can be apart to count as an exposure). We filter for both distance and time simultaneously to ensure that our exposure network includes only pairs of users who are likely to have crossed paths with each other. This high-resolution definition of exposure contrasts with other methods that consider all individuals that visit the same location, irrespective

of time[30], to have an equal likelihood of exposure, an unrealistic assumption because the SES of visitors to a given location can vary significantly by time (Supplementary Fig. 63)[61]. This fine-grained measure of proximity with respect to both time and space is the key advance that enables our findings (Supplementary Fig. 66). We use a threshold $T$ of 5 min, which is a stringent threshold on time as the mean number of pings per person per hour during day time is approximately one ping. We use a distance threshold $D$ of 50 m, because the cosmopolitan mixing hypothesis pertains to visual exposure[1–3] and following previous work showing that even exposure to individuals from afar is linked to long-term outcomes[19]. Our network is validated by correlation to external, gold-standard datasets (Extended Data Fig. 2). Furthermore, we show through a series of robustness checks that our key results in Figs. 1–3 are highly robust to varying thresholds (that is, 1 min or 2 min time thresholds, as well as 10 m or 25 m distance thresholds), as well as additional criteria to increase the tie strength (that is, requiring prolonged exposures, or multiple exposures on unique days). Under all of these different definitions of exposure, our main findings remain consistent (Supplementary Table 6 and Supplementary Figs. 2–8).

To efficiently calculate exposures that occurred among all users, we implement our exposure threshold as a $k$-dimensional ($k$-d) tree[62], a data structure that enables one to efficiently identify all pairs of points within a given distance of each other in a $k$-dimensional space. In total, we identify 1,570,782,460 exposures. The timestamp $t_{i,j,k}$ of the exposure is the minimum ping timestamp in the pair of individuals' ping timestamps ($t_i, t_j$). The location $\text{lat}_{i,j,k}, \text{lon}_{i,j,k}$ of the exposure is the average latitude and longitude of pair of pings belonging to the two individuals ($\text{lat}_i, \text{lat}_j$) and ($\text{lon}_i, \text{lon}_j$). We implement our exposure detection system to parallelize across multiple cores, enabling us to efficiently construct the network using a single supercomputer (with 12 TB RAM and 288 cores) in under a week. By contrast, a naive implementation (without $k$-d trees or parallelization) would necessitate on the order of ~10 years of computing time. The key challenge is accounting for proximity in time and space simultaneously, which results in an $O(n^2)$ time complexity for a naive implementation (where $n$ is the number of pings in the dataset), in contrast to previous work that is time agnostic and can therefore compute exposures using geohashes in $O(n)$ time[31,38].

**Annotating exposures.** Exposures are annotated to indicate whether they occurred at or near POIs, for example, at a user's home, or within a restaurant. Annotations are not mutually exclusive in that an exposure may be simultaneously tagged as having occurred near multiple POIs from multiple data sources. We describe the specific annotations below.

We annotate a user's exposure as having occurred at their home if it occurs within 50 m of the user's home location. An exposure is annotated with a TIGER road/railway if it occurs within 20 m from that feature. An exposure is annotated as having occurred within a SafeGraph Places POI if the exposure occurs within the polygon defined for the POI. Polygons are provided by the SafeGraph Places database for both fine-grained POIs (for example, individual restaurants) as well as parent POIs (such as hubs). We focus our analysis of fine-grained POIs (Fig. 2c and Extended Data Fig. 3) on the most visited fine-grained POIs, such as full-service restaurants, snack bars, limited-service restaurants (such as fast food) and stadiums (a full list is shown in Fig. 2c). These categories approximately align with those used in previous work[31].

## Analysis
**Exposure segregation.** We define the exposure segregation of a specified geographical area (that is, MSA or county) as the Pearson correlation between the SES of an individual residing in that geographical area and the mean SES of those who they encounter.

$$\text{Exposure segregation} = \text{Corr}(\text{SES}, \overline{\text{SES}}_{\text{exposures}}) = \frac{\text{cov}(\text{SES}, \overline{\text{SES}}_{\text{exposures}})}{\sigma_{\text{SES}} \sigma_{\overline{\text{SES}}_{\text{exposures}}}}$$

Our metric captures the extent to which an individual's SES predicts the SES of their immediate exposure network. Thus, in a perfectly integrated area in which individuals encounter others randomly regardless of SES, exposure segregation would equal 0.0. In a perfectly segregated area in which individuals encounter only those of the exact same SES, exposure segregation would equal 1.0. Our primary metric does not upweight repeated exposures to the same person (to avoid overly weighting strong ties such as housemates), although our key findings are robust to doing so (Supplementary Fig. 2).

Exposure segregation nests a classic definition of residential segregation, the neighbourhood sorting index[7], which is equivalent to the Pearson correlation between each person's SES and the mean SES in their census tract. The neighbourhood sorting index is widely used because it can be calculated directly from census data on the SES of people living in each tract. However, a fundamental limitation of the neighbourhood sorting index as a measure of segregation is that the census tract in which people live is a weak proxy for who they encounter. Census tracts are static and artificial boundaries that do not capture socioeconomic mixing as individuals move throughout the cityscape during work, leisure time and schooling.

We design our exposure segregation metric such that it accommodates any exposure network, and the neighbourhood sorting index is therefore a special case of our metric. Specifically, if exposure segregation is computed for a synthetic exposure network under the unrealistic assumptions that (1) people are exposed only to those in their home census tract; and (2) exposures occur uniformly at random, then it is equivalent to the neighbourhood sorting index (Supplementary Fig. 19). However, constructing such a synthetic exposure network from census tracts has limited applicability to measuring segregation in the real world, because people may also be exposed to more heterogeneous populations as they visit other census tracts for work, leisure or other activities, a phenomenon that we refer to as the visitor effect. Furthermore, even within the home tract, individuals may seek out people of similar SES; we refer to this as the homophily effect. We therefore instead leverage dynamic mobility data from mobile phones to capture the extent of contact between diverse individuals throughout the day, and apply our metric, exposure segregation, to this real-world exposure network. Our analyses reveal that our measure of exposure successfully captures both the visitor effect and the homophily effect (Fig. 2a). An advantage of our definition of exposure segregation is that it allows for direct comparability to the neighbourhood sorting index because both measures are of the same underlying statistical quantity, but differ in their definition of the exposure network. Our results indicate that this choice of exposure network matters; exposure segregation is a stronger predictor of upward economic mobility than the neighbourhood sorting index (Extended Data Fig. 2), and the two metrics are shown to be distinct (Supplementary Fig. 20).

To calculate the exposure segregation of a specified geographical area (that is, MSA or county), we first select the set of all individuals who reside in area $\mathcal{V}_A \subset \mathcal{V}$. For example, to calculate exposure segregation for Napa, California (Fig. 1c (top)), $\mathcal{V}_A$ is the 3,707 users with home locations inside the geographical boundary of Napa, CA. Subsequently, for each individual resident of the area $v_i \in \mathcal{V}_A$, we query the population exposure network ($\mathcal{G} = (\mathcal{V}, \mathcal{E})$) for the SES of the set of individuals who they cross paths with $\mathcal{Y}_i$: $\{x_j \in \mathcal{V} | e_{i,j,k} \in \mathcal{E}\}$. We then aim to estimate the Pearson correlation between the SES of each individual $x_i$ and the (unweighted) mean SES of those to whom they are exposed from all path crossings, $y_i = \text{mean}(\mathcal{Y}_i)$.

**Estimating exposure segregation.** Here we first motivate why a 'naive' approach to estimating exposure segregation through a sample Pearson correlation on the observed exposure network is problematic (resulting in downwardly biased estimates of exposure segregation). We then elaborate on how we leverage a linear mixed effects model

to compute a corrected Pearson correlation, enabling us to obtain unbiased estimates of exposure segregation even in areas where data are sparse.

A naive approach to estimate exposure segregation would be to first compute the observed sample mean SES of individuals who each person is exposed to. Exposure segregation could then be estimated using a sample Pearson correlation:

$$r_{xy} = \frac{\sum_{i=1}^{n}(x_i - \overline{x})(y_i - \overline{y})}{\sqrt{\sum_{i=1}^{n}(x_i - \overline{x})^2(y_i - \overline{y})^2}}$$

between an individual's SES ($x_i$) and the sample mean SES of those they are exposed to ($y_i$). This approach is problematic because naively computing such a correlation based on limited data (in counties or MSAs with low population sizes) will result in estimates that are downward biased. To illustrate why naive estimates of exposure segregation are downward biased, imagine that we compute the correlation between a person's SES and the 'true' mean SES of the people who they are exposed to. Now, we add noise to the mean SES values, which represents the noisy mean estimates given limited data. As the noise is increased, the correlation is decreased. Thus, because estimates of each person's mean SES will be more noisy in geographical areas with less data, there will be a downward bias to naive estimates of the Pearson correlation in these areas.

We instead compute a corrected Pearson correlation, using a linear mixed effects model to accurately estimate exposure segregation: the correlation between a person's SES and the mean SES of the people they are exposed to. Our linear mixed effects model is an unbiased estimator of the Pearson correlation. We compare the unbiased estimates from our linear mixed-effects model to the downwardly biased sample Pearson correlation estimates in Extended Data Fig. 1.

Our mixed model represents the distribution of datapoints ($x_i, y_{ij}$) through the following equation:

$$y_{ij} = ax_i + b + \epsilon_i^{(1)} + \epsilon_{ij}^{(2)},$$

where $x_i$ is the SES of person $i$, $y_{ij}$ is the SES of person $j$ who was exposed to person $i$, $a$ and $b$ are model parameters, $\epsilon_i^{(1)}$ is a person-specific noise term and $\epsilon_{ij}^{(2)}$ is the noise for each datapoint. Above, the true mean SES of the exposure set for each person is modelled as $ax_i + b + \epsilon_i^{(1)}$. Individual exposures $y_{ij}$ are then modelled as noisy draws from a distribution centred at this true mean. The Pearson correlation coefficient between person $i$'s SES and the mean SES of the people they were exposed to is then computed as follows. We assume that $x_i$ has a variance of 1 through data preprocessing and that $x_i$ is uncorrelated with $\epsilon_i^{(1)}$.

$$\begin{aligned}
\mathrm{corr}(x_i, ax_i + b + \epsilon_i^{(1)}) &= \mathrm{corr}(x_i, ax_i + \epsilon_i^{(1)}) \\
&= \frac{\mathrm{cov}(x_i, ax_i + \epsilon_i^{(1)})}{\sqrt{\mathrm{Var}(x_i)\mathrm{Var}(ax_i + \epsilon_i^{(1)})}} \\
&= \frac{\mathrm{cov}(x_i, ax_i)}{\sqrt{\mathrm{Var}(ax_i + \epsilon_i^{(1)})}} \\
&= \frac{a}{\sqrt{a^2 + \mathrm{Var}(\epsilon_i^{(1)})}}
\end{aligned}$$

We estimate $a$ and $\mathrm{Var}(\epsilon_i^{(1)})$ by fitting the mixed model using the R lme4 package, optimizing the restricted maximum-likelihood (REML) objective.

**Decomposing segregation by time.** Each exposure edge ($e_{i,j,k}$) in our exposure network is timestamped with a time of exposure $t_{i,j,k}$. This enables us to decompose our overall exposure segregation into

fine-grained estimates of segregation during different hours of the day by filtering for exposures that occurred within a specific hour. In Supplementary Fig. 21, we partition estimates of segregation by 3 h windows to illustrate how segregation varies throughout the day (Supplementary Information).

**Decomposing segregation by activity.** Each exposure edge ($e_{i,j,k}$) in our exposure network occurs at a specific location $\mathrm{lat}_{i,j,k}$, $\mathrm{lon}_{i,j,k}$. It is therefore possible to annotate exposures by the fine-grained POI (for example, specific restaurant) that they occurred in, as well as the by the higher-level parent POI (for example, shopping centre) in which the POI was located (see the 'Annotating exposures' section). This enables us to decompose our overall exposure segregation into fine-grained estimates of segregation by specific leisure activity. We do so by filtering the network for all exposures that occurred in a specific POI category, and recalculating exposure segregation for the MSA or county using only those exposures. In Fig. 2c, we show the variation in exposure segregation by leisure site, and further explain these variations in Extended Data Fig. 3.

**Bridging index.** We seek to identify a modifiable, extrinsic aspect of a city's built environment that may reduce exposure segregation. One promising candidate is the location of a city's highly visited POIs (that is, hubs). We define a new measure, the bridging index, which measures the extent to which a particular set of hubs ($\mathcal{P}$) may facilitate the integration of individuals of diverse SES within a geographical area (that is, MSA or county). The bridging index measures the economic diversity of the groups that would encounter one another if everybody visited only their nearest hub from $\mathcal{P}$, based on the observation that physical proximity significantly influences which hubs individuals visit[44–46].

The bridging index is computed through two steps (Extended Data Fig. 7):
(1) Cluster all individuals who live in an area (that is, MSA or county residents, $\mathcal{V}_A$) into $K$ clusters ($\mathcal{H}_1, \mathcal{H}_2, \ldots, \mathcal{H}_K$) according to the hub from $\mathcal{P}$ closest to their home location. $K$ is the number of hubs in $\mathcal{P}$.
(2) The bridging index is computed as the weighted average of the economic diversity (that is, Gini index) of these clusters of people, relative to the area's overall economic diversity.

$$\begin{aligned}
\text{Bridging index} &= \frac{\text{Within} - \text{hub economic diversity}}{\text{Overall economic diversity}} \\
&= \frac{\sum_{i=1}^{K}|\mathcal{H}_i| \times \text{Gini index}(\mathcal{H}_i)}{|\mathcal{V}_A| \times \text{Gini index}(\mathcal{V}_A)}
\end{aligned}$$

We illustrate the intuition for our bridging index and how it captures the relationship between home and hub locations in Extended Data Fig. 8. A bridging index of 1.0 indicates that, if everybody visits their nearest hub, each person will encounter a set of people as economically diverse as the overall city they reside in. Thus, a bridging index of 1.0 signifies perfect bridging, that is, even if individuals live in segregated neighbourhoods, hubs are located such that individuals must leave their neighbourhoods and encounter diverse others. On the other hand, a bridging index of 0.0 signifies the opposite extreme; a city with a bridging index of 0.0 is one in which, if everybody visits the nearest hub, each person will encounter only people of the exact same SES.

The economic diversity of each hub $\mathcal{H}_i$ is quantified using the Gini index: Gini index($\mathcal{H}_i$), a well-established measure of economic statistical dispersion[63] (Extended Data Fig. 7c), although results are robust to choice of economic diversity measure such as using variance instead of Gini index (Supplementary Fig. 14). The bridging index normalizes to the baseline economic diversity observed in the city, enabling direct comparisons between cities.

In our primary analysis, we identify hubs through commercial centres (such as shopping centres and plazas, which are higher-level clusters

of individual POIs) because they are associated with a high density of exposures. Specifically, the majority (56.9%) of exposures happen inside of or within 1 km of a commercial centre even though only 2.5% of the land area of MSAs is within 1 km of a commercial centre. We therefore compute our bridging index using the set $\mathcal{P}$ of all commercial centres within each MSA. We find that our bridging index strongly predicts exposure segregation (Spearman correlation = −0.78; Fig. 3d). The top 10 MSAs with the highest bridging index are 53.1% less segregated than the 10 MSAs with the lowest bridging index. The bridging index predicts segregation more accurately than population size, racial demographics SES inequality, the neighbourhood sorting index and racial demographics, and is significantly associated with segregation ($P < 10^{-4}$) after controlling for all aforementioned variables (Extended Data Tables 2 and 3).

**Hypothesis testing.** Unless otherwise noted, hypothesis tests and CIs were conducted using a bootstrap with 10,000 replications[64]. Steiger's *Z*-test was used to compare different predictors of segregation indices, and hypothesis tests for Spearman correlation coefficients were computed using two-sided Student's t-tests[65–67]. *P* values were not adjusted for multiple comparisons.

#### Reporting summary

Further information on research design is available in the Nature Portfolio Reporting Summary linked to this article.

## Data availability

Nationwide exposure segregation and bridging index measures are available online (http://segregation.stanford.edu). Raw mobility data are not publicly available to preserve privacy. Census data (5 year, 2013–2017 American Community Survey) are available online (https://www.census.gov/programs-surveys/acs). Zillow Rent Estimates are available online (https://www.zillow.com/howto/api/APIOverview.htm). TIGER data (TIGER Roads and Tiger Rails) are available online (https://www.census.gov/geographies/mapping-files/time-series/geo/tiger-geodatabase-file.html). The CoreLogic database is commercially available and may be requested for research use (https://www.corelogic.com/contact/). Individual mobile phone mobility data are not publicly available to preserve privacy, while mobility data aggregated to the CBG level and SafeGraph places data are commercially available and may be requested for research use (https://www.safegraph.com/contact-us).

## Code availability

Code is publicly available at GitHub (http://github.com/snap-stanford/exposure-segregation). All analysis was conducted using Python, except for the exposure segregation estimates, which were obtained using a mixed model implemented in R (see the 'Estimating exposure segregation' section of the Methods).

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

**Acknowledgements** We thank S. Chang, Y. Roohani, J. Gaebler, S. Goel, A. Chow, E. Jahanparast, S. Manek and A. Shirali for helpful conversations; and N. Singh, R. F. Squire, J. Williams-Holt, J. Wolf, R. Yang and other staff at SafeGraph for mobile phone mobility data and feedback. H.N. was supported by a Stanford Knight-Hennessy Scholarship and the National Science Foundation Graduate Research Fellowship under grant no. DGE-1656518; and E.P. by a Hertz Graduate Fellowship, a Google Research Scholar award, NSF CAREER 2142419, a CIFAR Azrieli Global scholarship, a LinkedIn Research Award, a Future Fund Regrant and the Abby Joseph Cohen Faculty Fund. This research was supported by DARPA under grant numbers HR00112190039 (TAMI) and N660011924033 (MCS); ARO under grant numbers W911NF-16-1-0342 (MURI) and W911NF-16-1-0171 (DURIP); NSF under grant numbers OAC-1835598 (CINES), OAC-1934578 (HDR) and CCF-1918940 (Expeditions); NIH under grant number 3U54HG010426-04S1 (HuBMAP); Stanford Data Science Initiative; Wu Tsai Neurosciences Institute; Amazon; GSK; Hitachi; Juniper Networks; KDDI; and Toshiba. The content is solely the responsibility of the authors and does not necessarily represent the official views of the funding entities. We thank the OpenStreetMap contributors, as the maps in Fig. 3 and Supplementary Fig. 65 were generated using OpenStreetMap data (https://www.openstreetmap.org/copyright).

**Author contributions** H.N., E.P., B.R., D.G. and J.L. conceived the study concept and design. H.N., W.L., E.P., B.V., N.F., Y.C. and J.S. conducted the computational analysis. All of the authors jointly interpreted the data and wrote the paper.

**Competing interests** The authors declare no competing interests.

**Additional information**
**Correspondence and requests for materials** should be addressed to Jure Leskovec.

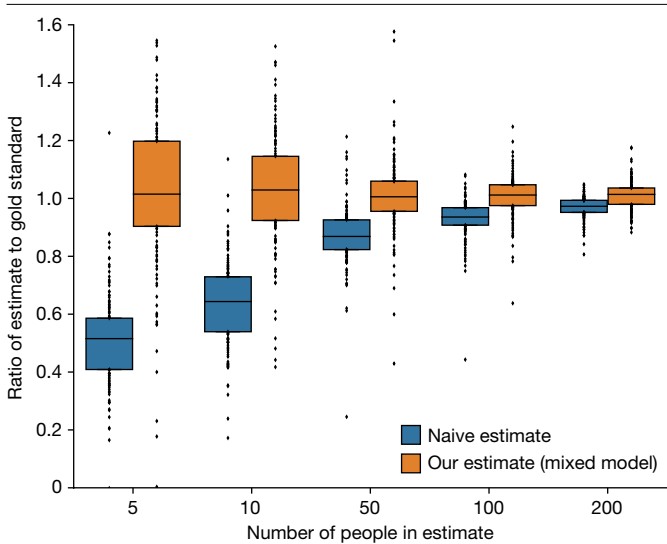

**Extended Data Fig. 1 | Unbiased estimates of exposure segregation using our mixed model compared with (downwardly biased) naive estimates using a sample Pearson correlation.** We first compute a gold standard estimate of exposure segregation. We do so by eliminating data sparsity (that is, restricting our analysis to individuals who crossed paths with at least 500 other people) and computing the 'naive' Pearson correlation coefficient between each individual's SES and the mean SES of those with whom they crossed paths (for each MSA). Next, for each person, we randomly downsample their path-crossings to 5, 10, 50, 100, and 200 (x-axis). On this noisy downsampled data, we estimate exposure segregation using both our mixed model (orange) and using the 'naive' Pearson correlation (blue). The y axis shows the ratio of these new estimates to the gold standard for each MSA. This analysis reveals that our mixed model enables us to obtain unbiased estimates of exposure segregation, whereas the 'naive' Pearson correlation is downwardly biased when observed path-crossings are sparse.

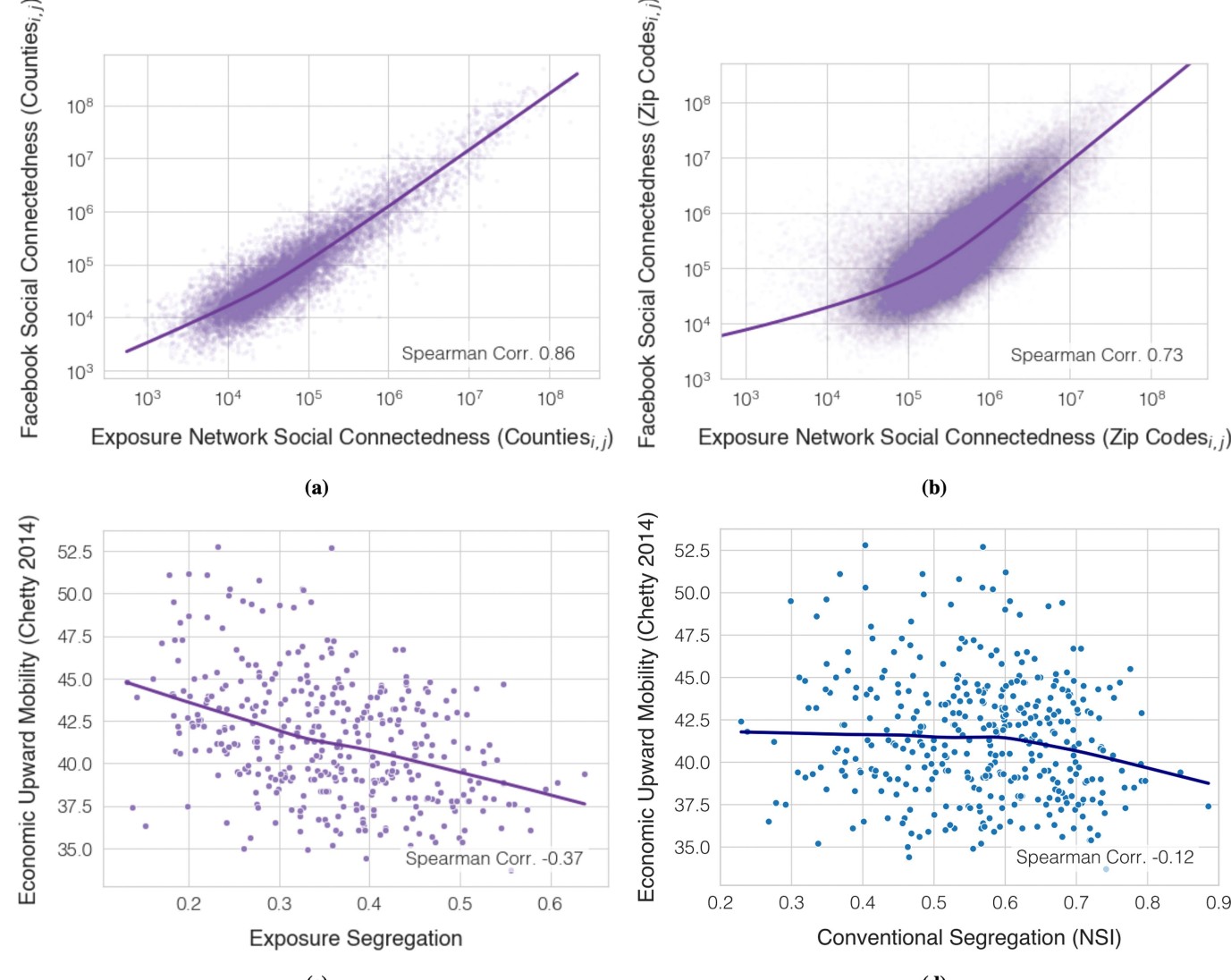

**(a)**

**(b)**

**(c)**

**(d)**

**Extended Data Fig. 2 | This studies' exposure network predicts population-scale friendship formation and upward economic mobility outcomes.**
We measure the external validity of our definition of exposure by linking our exposure network to outcomes across two gold-standard, large-scale, datasets. We find that at the zip code, county, and MSA-level, our exposure network mirrors population-scale outcomes resulting from dynamic human processes: **(a-b)** the Facebook social connectedness index[68] measures the relative probability of a Facebook friendship link between a given Facebook user in location $i$ and a given user in location $j$. FB social connectedness index has been used to study social segregation[69], and has also been linked to economic[70,71] and public health outcomes[72]. We reproduce the social connectedness index using our exposure network ($\frac{\#ExposurePairs_{i,j}}{\#Individuals_i \cdot \#Individuals_j}$) at the county **(a)** and zip code **(b)** level, and find strong correlations across county pairs (Spearman Correlation 0.85, $N = 121,595$, $p < 10^{-4}$; Two-sided Student's t-test; see the 'Hypothesis testing' section of

the Methods) and zip code pairs (Spearman Correlation 0.73, $N = 1,053,539$, $p < 10^{-4}$; Two-sided Student's t-test; see the 'Hypothesis testing' section of the Methods). Furthermore, we find that our exposure network is a stronger predictor of friendship formation than distance (Supplementary Tables 23-24). **(c-d)** The Chetty et al. intergenerational mobility dataset quantifies upward economic mobility from federal income tax records for each MSA as the mean income rank of children with parents in the bottom half of the income distribution[73]. We find that exposure segregation at the MSA-level **(c)** correlates to (absolute) upward economic mobility (Spearman Correlation -0.37, $N = 379$, $p < 10^{-4}$; Two-sided Student's t-test; see the 'Hypothesis testing' section of the Methods), and does so significantly more strongly ($p < 10^{-4}$; Two-sided Steiger's Z-test; see the 'Hypothesis testing' section of the Methods) than **(d)** the conventional segregation measure, neighbourhood sorting index (Spearman Correlation -0.12, $N = 379$, $p < 0.05$).

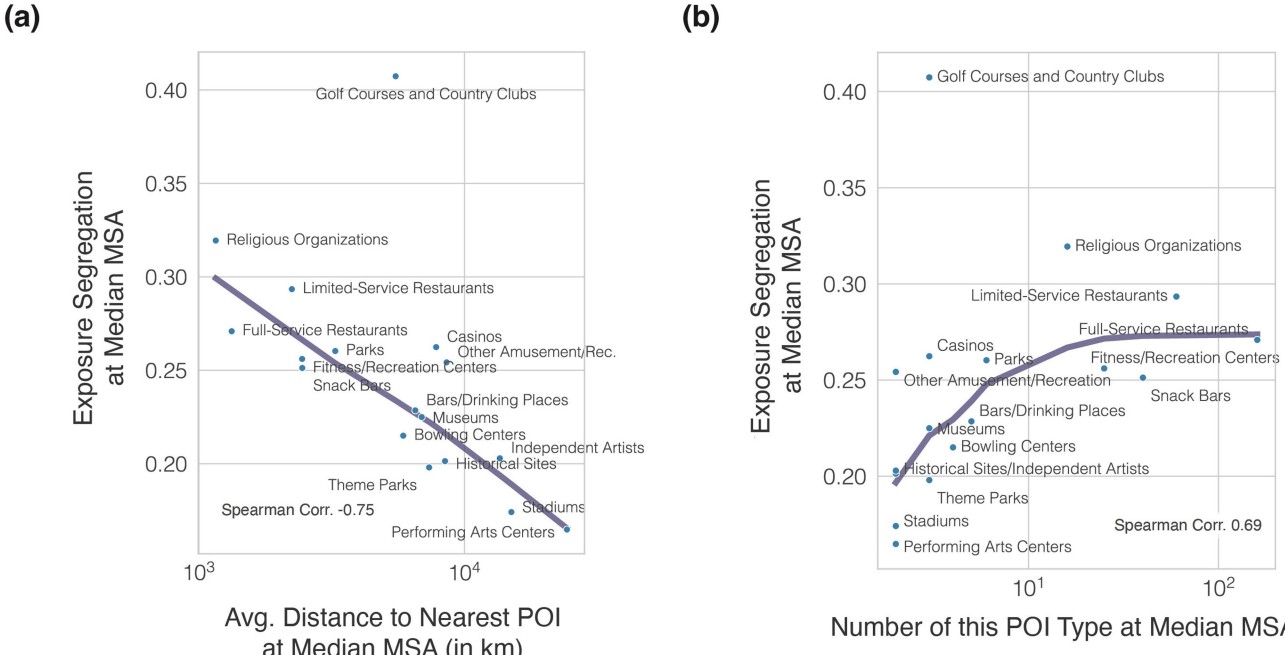

(a)

(b)

(c)

| Metropolitan Statistical Area (MSA) | POI Name | SES | Minimum Cost of Entrance ($) |
|---|---|---|---|
| Chicago-Naperville-Elgin, IL-IN-WI | Wynstone Golf Club | High | 40,000 |
| | Flagg Creek Golf Course | Low | 19 |
| Los Angeles-Long Beach-Anaheim, CA | Coto de Caza Golf & Racquet Club | High | 45,000 |
| | Rancho Vista Golf Club | Low | 129 |
| Miami-Fort Lauderdale-West Palm Beach, FL | Old Palm Golf Club | High | 199,200 |
| | Eco Golf Club | Low | 17 |
| New York-Newark-Jersey City, NY-NJ-PA | Scarsdale Golf Club | High | 8,900 |
| | Weequahic Park Golf Course | Low | 35 |
| Phoenix-Mesa-Scottsdale, AZ | The Estancia Club | High | 150,000 |
| | Peoria Pines | Low | 28 |

**Extended Data Fig. 3 | Understanding why exposure segregation varies significantly across leisure sites.** We identify three primary facets of socioeconomic differentiation between POIs which explain the heterogeneous segregation levels of different leisure POIs (Fig. 2c): **(a)** localization, **(b)** quantity, and **(c)** stratification. **(a)** Localization (average travel distance[30] to the nearest POI of a category) strongly predicts segregation across all POI categories (Spearman Correlation -0.75, N = 17, p < 0.001 Two-sided Student's t-test; see the 'Hypothesis testing' section of the Methods). POIs which are more locally embedded into neighbourhoods (e.g., religious organizations) are more segregated than POIs which serve multiple neighbourhoods (e.g., stadiums). **(b)** The quantity of POIs also explains segregation (Spearman Correlation 0.69, N = 17, p < 0.01; Two-sided Student's t-test; see the 'Hypothesis testing' section of the Methods). Leisure activities with more options (e.g., restaurants) have differentiated venues catering to a specific socioeconomic groups (e.g., Michelin-star restaurants) compared to POIs which are small in number and cater to the overall city (e.g., stadiums). **(c)** Golf courses and country clubs (golf clubs) are an anomaly in that they have a small number of unlocalized POIs, but are highly segregated. We conduct a case study of the top and bottom golf clubs by mean visitor SES in five of the ten largest MSAs. We find that the high segregation of golf clubs is due to extreme stratification between venues; for instance the minimum cost to play at the high-SES golf course in Miami, FL is 11717 × higher than at the lowest-SES golf course. By contrast, the average cost of a MacDonalds Big Mac ($5.65[74]) is only 63 × higher than the average cost of a Michelin 3-star restaurant ($357[75]). Overall, these findings foreshadow the bridging index, which captures POI localization, quantity, and stratification (Extended Data Fig. 8).

**(a)**

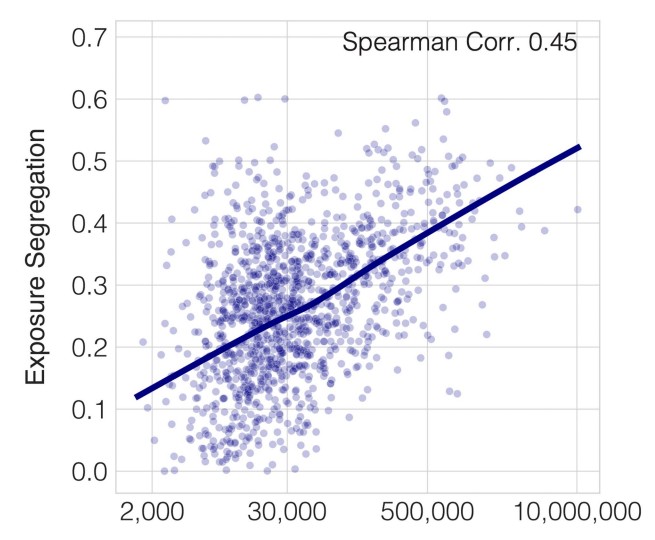

**(b)**

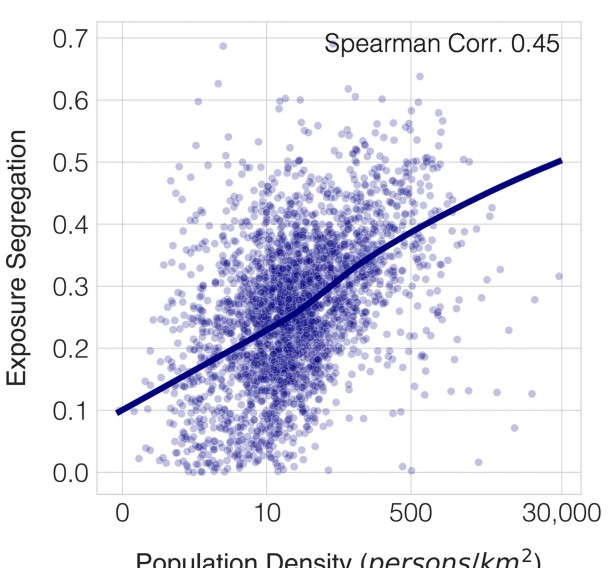

**Extended Data Fig. 4 | Large, dense counties are more segregated.** We compute exposure segregation across 2,829 USA counties (90% of the counties in the USA), excluding counties in which there are less than 50 individuals in our dataset. We find that at the county-level, exposure segregation is also positively correlated with population size (Spearman Correlation 0.45, N = 2829, $p < 10^{-4}$; Two-sided Student's t-test; see the 'Hypothesis testing' section of the Methods) and population density (Spearman Correlation 0.45, N = 2829, $p < 10^{-4}$; Two-sided Student's t-test; see the 'Hypothesis testing' section of the Methods). These correlations reveal that the association between large, dense urban areas and exposure segregation (Fig. 1d) is not an artifact of city boundaries, and may in fact be an emergent property from dynamics of individuals residing in highly populated, dense geographic areas, which persists across multiple scales of granularity.

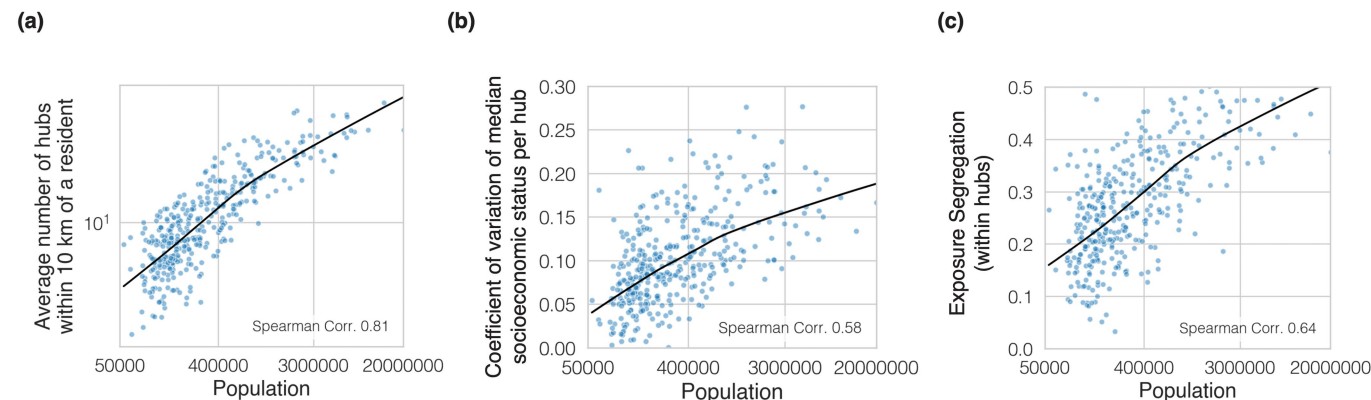

**Extended Data Fig. 5 | At higher levels of scale, spaces in large cities are more differentiated and consequently more segregated: hubs. (a-c)** We conduct an analysis for a city's hubs analogous to that for restaurants in Fig. 3c-e for a city's hubs. We find that higher segregation is driven by an increase in highly differentiated choice of hubs in large cities: (**a**) Larger MSAs have more hubs, giving residents more options to self-segregate (Spearman Correlation 0.81, N = 382, p < 10⁻⁴; Two-sided Student's t-test; see the 'Hypothesis testing' section of the Methods). (**b**) Consequently, hubs in larger MSAs vary more in terms of the mean SES of their visitors (Spearman Correlation 0.58, N = 382, p < 10⁻⁴; Two-sided Student's t-test; see the 'Hypothesis testing' section of the Methods) and as a result, (**c**) exposure segregation within hubs is higher in larger MSAs (Spearman Correlation 0.64, N = 382, p < 10⁻⁴; Two-sided Student's t-test; see the 'Hypothesis testing' section of the Methods). Overall, this analysis suggests that across multiple levels of scale, large cities offer a greater choice of differentiated spaces targeted to specific socioeconomic groups, promoting everyday segregation in exposures.

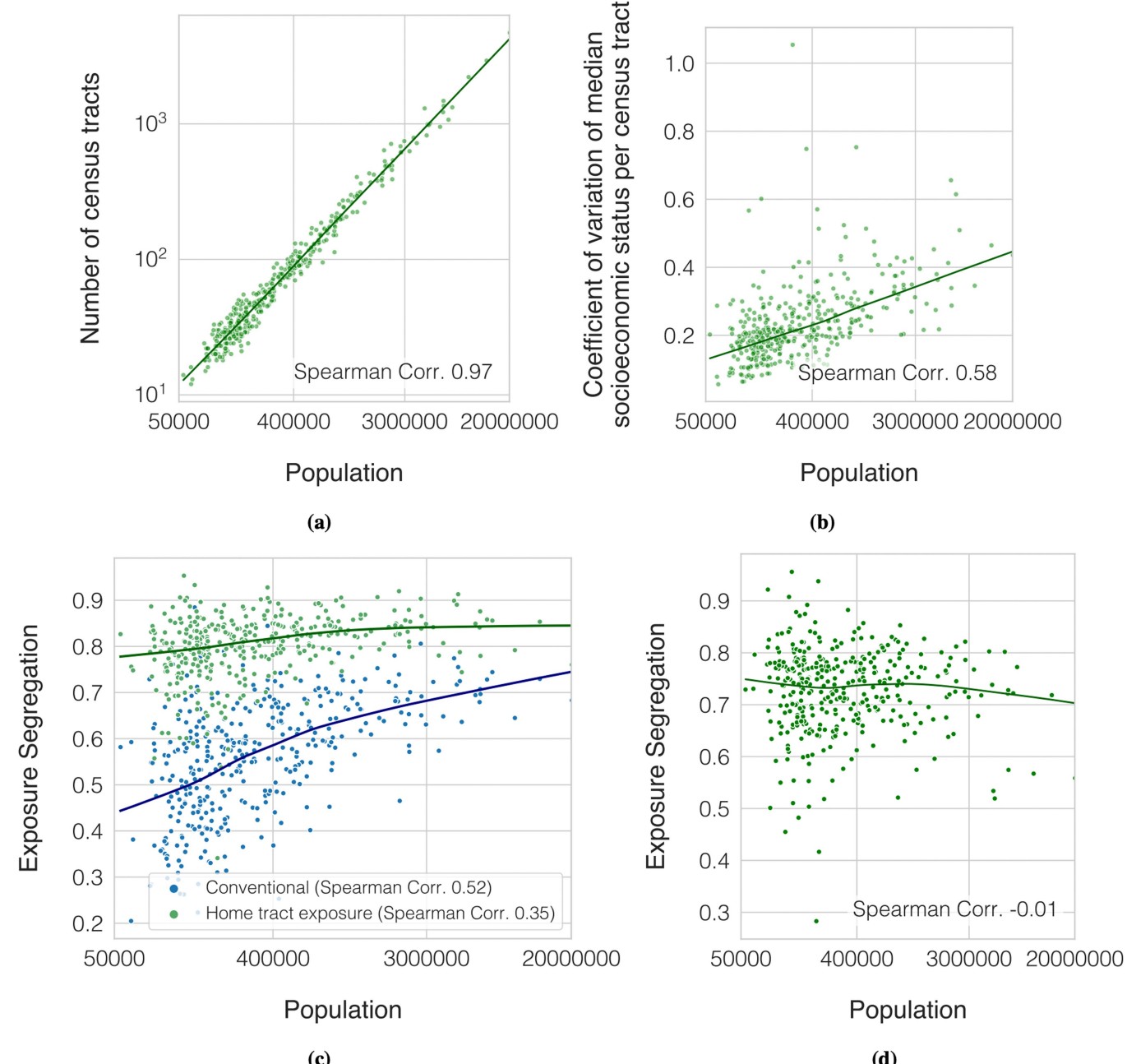

**(a)**

**(b)**

**(c)**

**(d)**

**Extended Data Fig. 6 | At higher levels of scale, spaces in large cities are more differentiated and consequently segregated: home neighbourhoods.** **(a-c)** Similar to the analysis for restaurants in Fig. 3c-e, we find that higher segregation is driven by an increase in highly differentiated choice of neighbourhoods in large cities: (**a**) Larger MSAs have more census tracts, giving residents more options to self-segregate (Spearman Correlation 0.97, N = 382, p < 10$^{-4}$; Two-sided Student's t-test; see the 'Hypothesis testing' section of the Methods). **(b)** Consequently, census tracts in larger MSAs vary more in terms of the mean SES of their residents (Spearman Correlation 0.58, N = 382, p < 10$^{-4}$; Two-sided Student's t-test; see the 'Hypothesis testing' section of the Methods) and as a result, (**c**) both residential segregation (neighbourhood sorting index) and exposure segregation are higher (Spearman Correlations 0.52 and 0.35, N = 382, p < 10$^{-4}$ and p < 10$^{-4}$; Two-sided Student's t-tests; see the 'Hypothesis testing' section of the Methods). However, (**c**) also shows that home

tract exposure segregation (green series) rises more slowly with population than conventional segregation (blue series), suggesting that within-home-tract homophily, which increases exposure segregation but not conventional segregation, is not more pronounced in large MSAs. Substantiating this, **(d)** shows that when home tract exposure segregation is computed using an alternate SES measure so it captures only within-home-tract-homophily, it is not higher in large MSAs (Spearman Correlation -0.01, N = 382, p > 0.1; Two-sided Student's t-test; see the 'Hypothesis testing' section of the Methods). The alternative SES measure is computed by subtracting the mean SES in each census tract. Overall, this analysis suggests that the higher home tract segregation in large MSAs is driven by people's greater choice of neighbourhoods of varying SES in which to live, but not by a greater tendency to cross paths homophilously within their own neighbourhood.

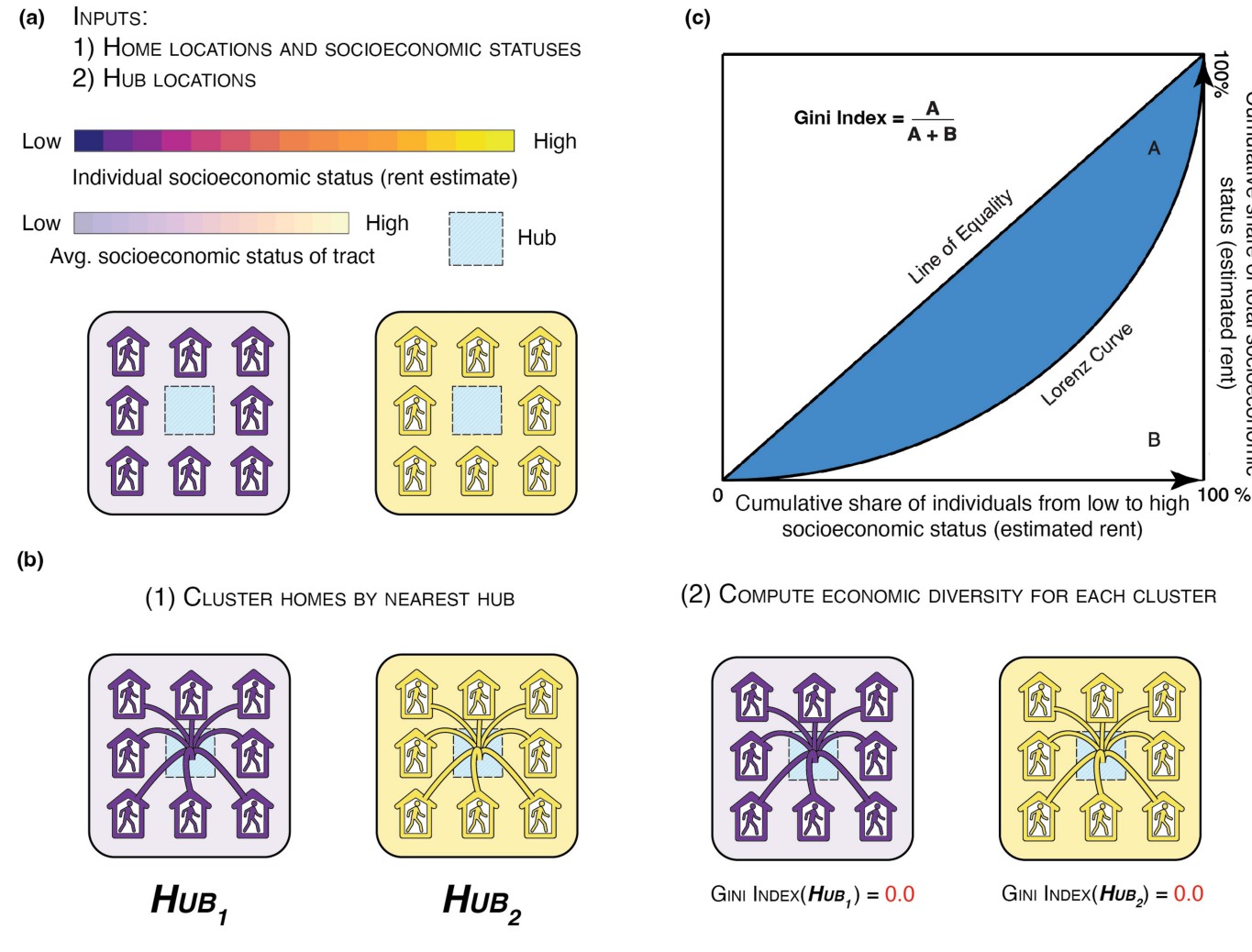

**(a)** INPUTS:
1) HOME LOCATIONS AND SOCIOECONOMIC STATUSES
2) HUB LOCATIONS

Low —— High
Individual socioeconomic status (rent estimate)

Low —— High    Hub
Avg. socioeconomic status of tract

**(c)**

$$\text{Gini Index} = \frac{A}{A+B}$$

Line of Equality

A

Lorenz Curve

B

Cumulative share of total socioeconomic status (estimated rent)

0    Cumulative share of individuals from low to high socioeconomic status (estimated rent)    100 %

**(b)**

(1) CLUSTER HOMES BY NEAREST HUB

$HUB_1$    $HUB_2$

(2) COMPUTE ECONOMIC DIVERSITY FOR EACH CLUSTER

GINI INDEX($HUB_1$) = 0.0    GINI INDEX($HUB_2$) = 0.0

(3) NORMALIZE WEIGHTED AVERAGE BY OVERALL MSA GINI INDEX

$$\text{BRIDGING INDEX} = \frac{\text{WITHIN-HUB ECONOMIC DIVERSITY}}{\text{OVERALL ECONOMIC DIVERSITY}} = \frac{\sum_{i=1}^{K} |HUB_i| \times \text{GINI INDEX}(HUB_i)}{|MSA| \times \text{GINI INDEX}(MSA)} = \frac{9 \times 0.0 + 9 \times 0.0}{18 \times 0.16} = 0.0 \ (\text{NO BRIDGING})$$

**Extended Data Fig. 7 | Computing bridging index.** Illustration of our analytical pipeline for calculating the bridging index. **(a)** Bridging index is computed from the locations and number of POIs in the MSA which are expected to be hubs of exposure (that is, frequently visited POIs), as well the locations and SES values of all homes within MSA boundaries. We intentionally develop bridging index without using mobility data, with the intention of identifying a modifiable extrinsic aspect of a city that can be intervened on to impact mobility patterns and decrease exposure segregation. **(b)** In order, we (1) cluster all homes by nearest hub (using straight line distance from home to hub), partitioning all homes into $K$ clusters, where $K$ is the number of hubs in the MSA (2) compute the weighted average economic diversity (i.e., Gini index) of the clusters, normalized by the overall economic diversity of the MSA to allow for comparisons between different MSAs of varying baseline levels of economic diversity (Extended Data Table 1). **(c)** The graphical definition of Gini index is provided, which is a standard measure of economic dispersion[63]. Results are robust to the definition of economic diversity, and hold true when using variance in SES instead of Gini index (Supplementary Fig. S14).

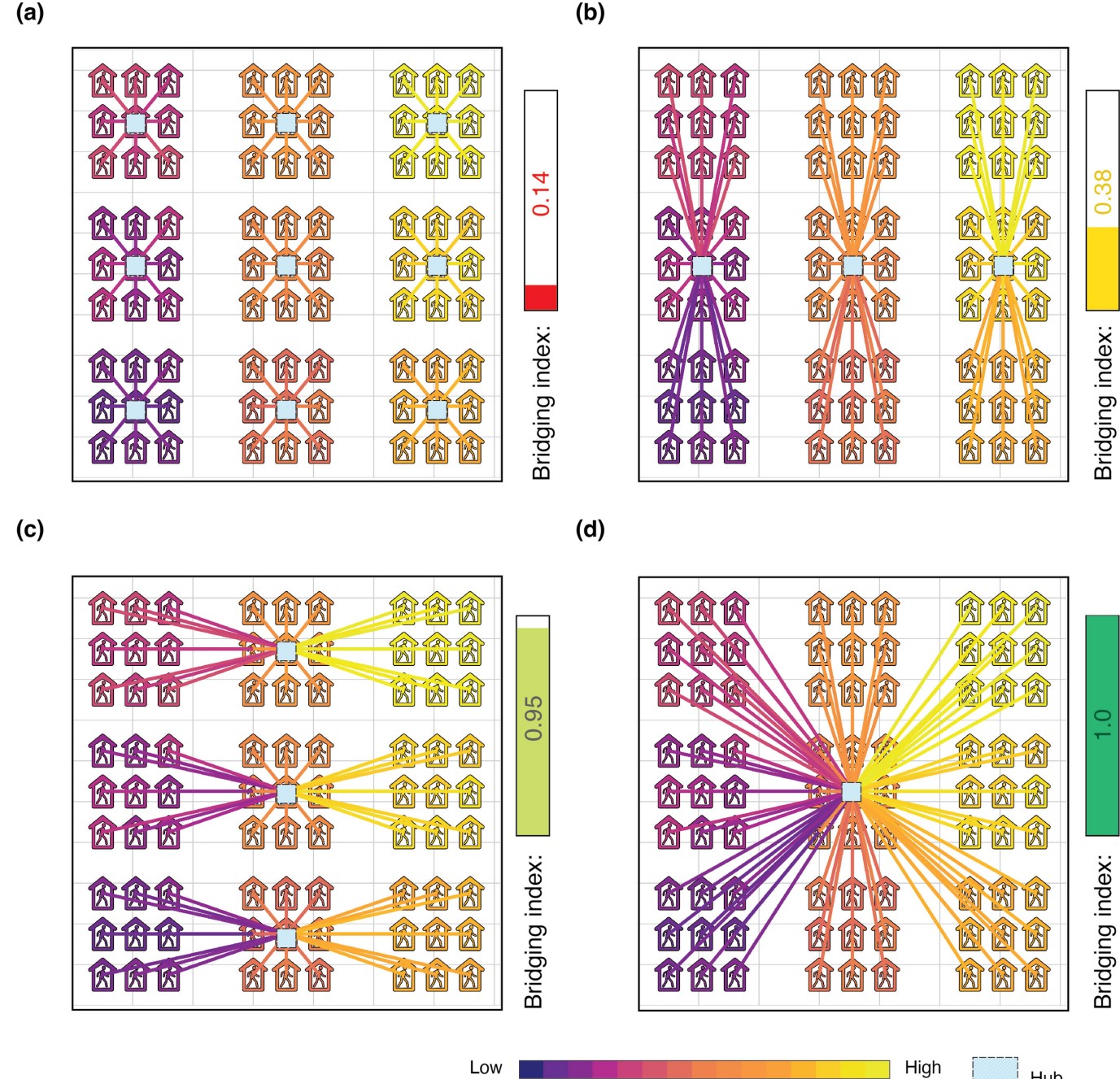

**Extended Data Fig. 8 | Understanding the determinants of the bridging index.** The bridging index is a single metric which captures three important factors of built environment (see Supplementary Fig. S13 for contributions of these factors to explaining exposure segregation): (1) The locations of hubs—if hubs are located in between diverse neighbourhoods, the bridging index will be high as hubs will bridge together diverse individuals. (2) The number of hubs—as number of hubs decreases, bridging index increases (e.g if there is only 1 hub in a city, bridging index will be 1.0 as all individuals are unified by a single hub) (3) Residential segregation, i.e., the locations of homes and their associated SES—as residential segregation decreases we can expect that individuals residing near each hub will be more diverse. This figure builds intuition for how the bridging index may vary for a single simulated city, consisting of highly segregated neighbourhoods. We hold residential segregation (3) constant, and vary the location (1) and number (2) of hubs across panels **(a)**, **(b)**, **(c)**, **(d)**, in order of increasing bridging index. Note that the bridging index in **(c)** is substantially higher than the bridging index in **(b)**, because hubs in **(c)** are better positioned to bridge diverse neighbourhoods—even though the number of hubs remains constant.

**Extended Data Table 1 | Population size is significantly associated with exposure segregation, after controlling for MSA income inequality (Gini index), political alignment (% Democrat in 2016 election), racial demographics (% non-Hispanic White), mean SES, walkability (Walkscore[76]), commutability (% of residents commuting to work), and residential segregation (neighbourhood sorting index)**

| | *Dependent variable:* | | | | Exposure segregation | |
| --- | --- | --- | --- | --- | --- | --- |
| | (1) | (2) | (3) | (4) | (5) | (6) |
| Intercept | 0.355*** | 0.355*** | 0.355*** | 0.355*** | 0.355*** | 0.355*** |
| | (0.004) | (0.004) | (0.003) | (0.003) | (0.003) | (0.003) |
| log(Population size) | 0.059*** | | 0.041*** | 0.044*** | 0.026*** | 0.028*** |
| | (0.004) | | (0.004) | (0.004) | (0.004) | (0.004) |
| Gini index (estimated rent) | | 0.064*** | 0.050*** | 0.051*** | 0.045*** | 0.047*** |
| | | (0.004) | (0.004) | (0.004) | (0.003) | (0.003) |
| Political alignment (% Democrat in 2016 election) | | | | 0.004 | | 0.004 |
| | | | | (0.004) | | (0.004) |
| Racial demographics (% non-Hispanic White) | | | | 0.001 | | 0.006* |
| | | | | (0.004) | | (0.003) |
| Mean SES (estimated rent) | | | | -0.012*** | | -0.005 |
| | | | | (0.004) | | (0.004) |
| Walkability (Walkscore) | | | | | 0.002 | 0.001 |
| | | | | | (0.003) | (0.003) |
| Commutability (% commute to work) | | | | | -0.011*** | -0.010*** |
| | | | | | (0.003) | (0.004) |
| Residential segregation (neighbourhood sorting index) | | | | | 0.042*** | 0.041*** |
| | | | | | (0.003) | (0.003) |
| Observations | 382 | 382 | 382 | 376 | 382 | 376 |
| $R^2$ | 0.350 | 0.419 | 0.567 | 0.578 | 0.704 | 0.705 |
| Adjusted $R^2$ | 0.348 | 0.417 | 0.565 | 0.573 | 0.701 | 0.698 |

*p<0.1; **p<0.05; ***p<0.01

Here we show the regression coefficients (after normalizing each variable via z-scoring to have mean 0 and variance 1) from the key analyses estimating the effect of population size on exposure segregation across all MSAs. Standard errors are displayed in parentheses beneath each coefficient (*p<0.1; **p<0.05; ***p<0.01; Two-sided Student's t-test; see the 'Hypothesis testing' section of the Methods). Columns (1–5) are models specified with different subsets of covariates; Column 6 shows model specification with all covariates. Differences between sample size in models is due to missing Walkscore data in a small number of MSAs.

**Extended Data Table 2 | Bridging index is significantly associated with exposure segregation, after controlling for MSA population size, number of hubs, income inequality (Gini index), political alignment (% Democrat in 2016 election), racial demographics (% non-Hispanic White), mean SES, walkability (Walkscore[76]), commutability (% of residents commuting to work), and residential segregation (neighbourhood sorting index)**

|  | *Dependent variable:* | | | Exposure segregation | |
|---|---|---|---|---|---|
|  | (1) | (2) | (3) | (4) | (5) |
| Intercept | 0.355*** | 0.355*** | 0.355*** | 0.355*** | 0.355*** |
|  | (0.003) | (0.003) | (0.003) | (0.003) | (0.003) |
| Bridging index | -0.078*** | -0.059*** | -0.058*** | -0.035*** | -0.036*** |
|  | (0.003) | (0.005) | (0.005) | (0.006) | (0.006) |
| log(Population size) |  | 0.003 | 0.008* | 0.010** | 0.017*** |
|  |  | (0.004) | (0.005) | (0.004) | (0.006) |
| Gini index (estimated rent) |  | 0.031*** | 0.032*** | 0.035*** | 0.036*** |
|  |  | (0.003) | (0.003) | (0.003) | (0.003) |
| Political alignment (% Democrat in 2016 election) |  |  | 0.001 |  | 0.002 |
|  |  |  | (0.004) |  | (0.004) |
| Racial demographics (% non-Hispanic White) |  |  | 0.003 |  | 0.005 |
|  |  |  | (0.003) |  | (0.003) |
| Mean SES (estimated rent) |  |  | -0.009** |  | -0.005 |
|  |  |  | (0.004) |  | (0.003) |
| Walkability (Walkscore) |  |  |  | 0.002 | 0.001 |
|  |  |  |  | (0.003) | (0.003) |
| Commutability (% commute to work) |  |  |  | -0.011*** | -0.009** |
|  |  |  |  | (0.003) | (0.004) |
| Residential segregation (neighbourhood sorting index) |  |  |  | 0.028*** | 0.026*** |
|  |  |  |  | (0.004) | (0.004) |
| Number of hubs |  |  |  |  | -0.006 |
|  |  |  |  |  | (0.005) |
| Observations | 382 | 382 | 376 | 382 | 376 |
| $R^2$ | 0.620 | 0.686 | 0.693 | 0.733 | 0.736 |
| Adjusted $R^2$ | 0.619 | 0.684 | 0.688 | 0.729 | 0.729 |

*p<0.1; **p<0.05; ***p<0.01

Here we show the regression coefficients (after normalizing each variable via z-scoring to have mean 0 and variance 1) from the key analyses estimating the effect of bridging index on exposure segregation across all MSAs. Standard errors are displayed in parentheses beneath each regression coefficient (*p<0.1; **p<0.05; ***p<0.01; Two-sided Student's t-test; see the 'Hypothesis testing' section of the Methods). Columns (1–4) are models specified with different subsets of covariates; Column 5 shows model specification with all covariates. Differences between sample size in models is due to missing Walkscore data in a small number of MSAs.

**Extended Data Table 3 | Bridging index strongly predicts exposure segregation**

| Measure | Spearman $\rho^2$ | Pearson $R^2$ |
|---|---|---|
| **Bridging index** | **0.60** | **0.62** |
| log(Population size) | 0.39 | 0.35 |
| Gini index (estimated rent) | 0.41 | 0.42 |
| Political alignment (% Democrat in 2016 election) | 0.06 | 0.05 |
| Racial demographics (% non-Hispanic White) | 0.09 | 0.05 |
| Mean SES (estimated rent) | 0.09 | 0.05 |
| Walkability (Walkscore) | 0.01 | 0.02 |
| Commutability (% commute to work) | 0.04 | 0.03 |
| Residential segregation (neighbourhood sorting index) | 0.44 | 0.42 |
| Number of hubs | 0.44 | 0.16 |

and does so more accurately ($p < 10^{-4}$; Two-sided Steiger's Z-test; see the 'Hypothesis testing' section of the Methods) than MSA population size, number of hubs, income inequality (Gini index), political alignment (% Democrat in 2016 election), racial demographics (% non-Hispanic White), mean SES, walkability (Walkscore[76]), commutability (% of residents commuting to work), and residential segregation (neighbourhood sorting index).

# Reporting Summary

## Statistics

For all statistical analyses, confirm that the following items are present in the figure legend, table legend, main text, or Methods section.

| n/a | Confirmed | |
|---|---|---|
| ☐ | ☒ | The exact sample size (*n*) for each experimental group/condition, given as a discrete number and unit of measurement |
| ☐ | ☒ | A statement on whether measurements were taken from distinct samples or whether the same sample was measured repeatedly |
| ☐ | ☒ | The statistical test(s) used AND whether they are one- or two-sided<br>*Only common tests should be described solely by name; describe more complex techniques in the Methods section.* |
| ☐ | ☒ | A description of all covariates tested |
| ☐ | ☒ | A description of any assumptions or corrections, such as tests of normality and adjustment for multiple comparisons |
| ☐ | ☒ | A full description of the statistical parameters including central tendency (e.g. means) or other basic estimates (e.g. regression coefficient) AND variation (e.g. standard deviation) or associated estimates of uncertainty (e.g. confidence intervals) |
| ☐ | ☒ | For null hypothesis testing, the test statistic (e.g. *F*, *t*, *r*) with confidence intervals, effect sizes, degrees of freedom and *P* value noted<br>*Give P values as exact values whenever suitable.* |
| ☒ | ☐ | For Bayesian analysis, information on the choice of priors and Markov chain Monte Carlo settings |
| ☒ | ☐ | For hierarchical and complex designs, identification of the appropriate level for tests and full reporting of outcomes |
| ☐ | ☒ | Estimates of effect sizes (e.g. Cohen's *d*, Pearson's *r*), indicating how they were calculated |

*Our web collection on statistics for biologists contains articles on many of the points above.*

## Software and code

Policy information about availability of computer code

| Data collection | No data collection was performed for this study; all analysis relied on previously collected datasets, as described in the Data section below. |
|---|---|
| Data analysis | Primary data analysis was performed using Python version 3.9.5 and standard libraries. Ping interpolation was performed with SciPy version 1.4.0. Exposure segregation was estimated using R version 3.6.3 and the lme4 library version 1.1-21. Code is available at https://github.com/snap-stanford/exposure-segregation |

For manuscripts utilizing custom algorithms or software that are central to the research but not yet described in published literature, software must be made available to editors and reviewers. We strongly encourage code deposition in a community repository (e.g. GitHub). See the Nature Portfolio guidelines for submitting code & software for further information.

## Data

Policy information about availability of data

All manuscripts must include a data availability statement. This statement should provide the following information, where applicable:
- Accession codes, unique identifiers, or web links for publicly available datasets
- A description of any restrictions on data availability
- For clinical datasets or third party data, please ensure that the statement adheres to our policy

Nationwide exposure segregation and bridging index measures are available at http://segregation.stanford.edu. Census (https://www.census.gov/programs-surveys/acs), Zillow (https://www.zillow.com/howto/api/APIOverview.htm), and TIGER (https://www.census.gov/geographies/mapping-files/time-series/geo/tiger-

geodatabase-file.html) data are publicly available. CoreLogic database is commercially available and may be
requested for research use (https://www.corelogic.com/contact/). Individual cell phone mobility data are not publicly available to preserve privacy, while mobility
data aggregated to the Census block group (CBG level) and SafeGraph places data are commercially available and may be requested for research use (https://www.safegraph.com/contact-us).

## Human research participants

Policy information about studies involving human research participants and Sex and Gender in Research.

| | |
|---|---|
| Reporting on sex and gender | Sex and gender not collected. |
| Population characteristics | Our primary analysis sample was constructed from previously collected, de-identified mobility data provided by the company SafeGraph (https://www.safegraph.com/). We filtered individuals to those with at least a total of 500 pings across three evenly spaced the months in 2017: March, July, and November. We further excluded individuals that shared 80% or more identical pings with another individual, did not have any pings with < 100 meters accuracy, or for whom the Zillow API did not return an estimated rent value. Our final analysis sample consisted of 9,567,559 cell phones. |
| Recruitment | See above. As described by in public documentation, SafeGraph data is collected by: "partner[ing] with mobile applications that obtain opt-in consent from its users to collect anonymous location data." SafeGraph ensures that its mobile application partners obtain consent for data to be used for commercial and research purposes, including academic publication. SafeGraph users are able to opt-out of data collection at any time. Prior work has investigated biases in the SafeGraph dataset (https://www.safegraph.com/blog/what-about-bias-in-the-safegraph-dataset) |
| Ethics oversight | Stanford University IRB |

Note that full information on the approval of the study protocol must also be provided in the manuscript.

## Field-specific reporting

Please select the one below that is the best fit for your research. If you are not sure, read the appropriate sections before making your selection.

☐ Life sciences  ☒ Behavioural & social sciences  ☐ Ecological, evolutionary & environmental sciences

For a reference copy of the document with all sections, see nature.com/documents/nr-reporting-summary-flat.pdf

## Behavioural & social sciences study design

All studies must disclose on these points even when the disclosure is negative.

| | |
|---|---|
| Study description | This is a quantitative, retrospective observational study. |
| Research sample | We study previously collected, de-identified mobility data from provided by the company SafeGraph. As described by SafeGraph in public documentation, SafeGraph data is collected by: "partner[ing] with mobile applications that obtain opt-in consent from its users to collect anonymous location data. This data is not associated with any name or email address. This data includes the latitude and longitude of a device at a given point in time." SafeGraph ensures that its mobile application partners obtain consent for data to be used for commercial and research purposes, including academic publication. SafeGraph users are able to opt-out of data collection at any time. While SafeGraph data is not a random sample, it is geographically well-balanced (i.e., an approximately unbiased sample of different census tracts within each State), and well-balanced along the dimensions of race, income, and education. SafeGraph data was chosen as the study sample because of its scale, geographical breadth, and as it is a widely used standard in previous studies of human mobility. This data was joined with Census (demographics), Zillow (estimated rent), SafeGraph Places (POIs), CoreLogic (addresses), and TIGER (roads and railways) data. |
| Sampling strategy | We did not perform sampling, sample size (N=9,567,559) was determined by the size of the SafeGraph database after filtering (see below). SafeGraph anonymized cell phone data has been shown to be geographically representative across on many key racial, economic, and demographic variables (https://www.safegraph.com/blog/what-about-bias-in-the-safegraph-dataset). |
| Data collection | We did not perform data collection, but relied on previously collected, anonymized geolocation data by the company SafeGraph. |
| Timing | Three (evenly space) months from 2017: March, July, and November. |
| Data exclusions | We apply several filters to improve reliability of the SafeGraph data: all participants logged at least 500 pings, had ping locations with an accuracy < 100 meters, and were at least 20% distinct from other participants (de-duplication). |
| Non-participation | SafeGraph users are able to opt-out of data collection at any time (by changing application settings). Data provided by SafeGraph does not distinguish between user inactivity and opting out from data collection. |
| Randomization | Observational study, participants were not randomized. Key findings are robust to controlling (via regression covariates) for MSA- |

Randomization | level income inequality, political demographics, racial demographics, mean economic standing, walkability, commutability and residential segregation.

# Reporting for specific materials, systems and methods

We require information from authors about some types of materials, experimental systems and methods used in many studies. Here, indicate whether each material, system or method listed is relevant to your study. If you are not sure if a list item applies to your research, read the appropriate section before selecting a response.

## Materials & experimental systems

| n/a | Involved in the study |
|-----|----------------------|
| ☒ ☐ | Antibodies |
| ☒ ☐ | Eukaryotic cell lines |
| ☒ ☐ | Palaeontology and archaeology |
| ☒ ☐ | Animals and other organisms |
| ☒ ☐ | Clinical data |
| ☒ ☐ | Dual use research of concern |

## Methods

| n/a | Involved in the study |
|-----|----------------------|
| ☒ ☐ | ChIP-seq |
| ☒ ☐ | Flow cytometry |
| ☒ ☐ | MRI-based neuroimaging |

