## [Peer Review File · Nature]

Manuscript Title: Human mobility networks reveal increased segregation in large cities

Reviewer Comments & Author Rebuttals

Reviewer Reports on the Initial Version:

Referees' comments:

Referee #1 (Remarks to the Author):

"Human interaction networks reveal that large cities facilitate segregation" is a well-written and impressively researched manuscript which offers up an interesting new measure of segregation, which leads to a number of interesting findings, most notably that large cities tend to have higher segregation than surrounding areas, contrary to what has been speculated in the literature.

It was a pleasure to read the excellent main text and delve into the impressive SI, but before I can recommend publication I have a number of serious concerns about the network of interactions underlying the excellent analysis presented in the paper. The construction of this network (and the network itself) is not sufficiently well described in the paper/SI. And fully understanding the network is important as the entire downstream analysis rests upon that foundation. Understanding the network is important both to ensure that there are no potential biases - and for the interpretation of the results. Concerns around the network also extend to parts of the presentation (for example I think that given the underlying network, the words "interaction" and "contact" need to be eliminated from the text, incl the title). Below I elaborate in detail.

Major concerns

(1) The "network". As mentioned above, I don't think the MS should use the word "contact" or "interaction" to describe what occurs between pairs of individuals in this dataset: that phrasing is frankly misleading. Let me explain why.

While the number of individuals in this dataset is truly impressive, we still only have access to a small fraction (let's say $\sim 2.5\%$) of all Americans. When one has access to only a small fraction of nodes, that implies access to a much smaller fraction of the edges (in the random case, the fraction of edges we sample scales as the fraction of nodes squared) $\sim 0.06\%$ of all links. Combine that with the fact that the data (presumably) was collected over time which introduces further sparsity.

All this implies that *if* we could perfectly collect all real social interactions between pairs of individuals in the dataset, we would only see a very small number of links in the network.

The dataset may or may not be sparse (one cannot tell given the current MS), but it is clear that the authors adopt a generous definition of "interaction", namely that two individuals "interact" if they are observed less than 5 mins apart and within 50 meters of one another. This definition

a) does not necessarily capture all real social links

b) very likely captures many "interactions" that are not real social interactions in any meaningful sense (since the dataset has a lot of interactions: 1.6 billion interactions and 9.6 million nodes).

Let me just motivate one possible (out of several) to motivate point b): In the case of many datasets, the majority of raw location data collected by smartphone apps capture individuals' positions while people are on the move (this is because while they move, location is collected at higher frequency, due to people e.g. using navigation apps). The authors do not describe how

location data is sampled temporally in their dataset (more on this below), but I don't have a reason to assume that it is sampled only when people are still. Thus, it is possible that many of the interactions detected occur while people are driving by each other.

I would like the authors to comment on the arguments above and provide a discussion of which fraction of their measured interactions correspond to underlying social relationships (e.g. some kind of preexisting relationship: two people would consider each other friends, or know of one another's existence, or know each other's name, etc.)

All this also implies (to me at least) that a terminology along the lines of "exposure" would be more honest. (I think what the authors are measuring might be meaningful ... it's just not social interactions. More of a measure of social mixing in space and time. And I also still worry that it might be biased, see below)

Further, as a reader, it is very difficult to gauge which fraction of the inferred data is real and which is noise, since the network is not reported on at all. All we have access to are the derived quantities. And these derived quantities can make a very sparse dataset appear quite dense. To calculate the mean ES of people someone has interacted with, for example, you only need a single datapoint.

To help the reader better understand the results, the MS should also provide information about the inferred networks. Are they meaningful? Here, I would like to see

- * distribution of number of contacts per person (all time)
- * distribution of number of contacts per person (for a single day)
- * average/median number of active nodes over time
- * average/median number of contacts per person over time
- * distribution of link strength (how many observations do we typically have for each link)

(1.5) The network / temporal sampling. Neither the MS nor SI do not say anything about the temporal properties of the GPS pings. We know that individuals have at least 500 pings, but I would like to see

- the distribution of pings per person,
- statistics about how the pings are distributed in time, what does, for example, distribution of the number of days of data coverage per person,
- what is the distribution of interevent times, etc. Is the data evenly collected at some fixed rate - or does datapoints clump together in bursts of collected data?
- a plot that shows the number of individuals with data per day as a function of calendar-time
- a plot of the number of estimated interactions day as a function of calendar-time (also for the various robustness checks)

These temporal properties have important consequences for the inferred interactions.

(2) The probability of an "interaction" depends on the overall density of individuals in an area (a city). Let me illustrate using an example. Let's say we have data from two groups of 100 individuals sampled from low and high population density areas. To make things concrete, let's say that Group A all live within a 1x1 mile area and Group B live distributed across a 20x20 mile area. My common sense would say that the method presented by the authors would detect many more random encounters (= a denser network) among group A than group B, simply because they occupy a smaller geographical space where stop-locations are more densely packed. This dependence density is concerning, since the authors are making claims about how the mixing of social relationships appear to depend on density (metropolitan areas).

I would like the authors to comment on this issue and what it might mean for their results.

(2.5) Density at individual locations. A key takeaway from this paper <https://www.pnas.org/doi/10.1073/pnas.1006155107> is that the background density for two individuals is useful information for inferring social interactions. If two people are observed nearby one another at a Giants Stadium during a game, that co-occurrence is less significant than if they are observed together at midnight in a remote mountain cabin.

I worry that this effect might bias the results for encounters and mixing at specific venues ... and would like to hear the authors' comments to this issue.

(3) Null models (in the MS). In the SI the authors perform several robustness checks and show that their results (in a qualitative sense) hold in spite of many different (including stricter) definitions of a "social interaction". But because I am interested in the *network* and not just (quantities derived from the network), I would like to better understand what happens to the network properties - the metrics listed under (1) above, for the "robustness check networks" shown Fig S2-S8.

(3.5): Null models (that would help clarify what's going on). Even though there are already several null models, I would like to suggest some extra ones.

* To estimate the fraction of random encounters, the authors could randomly displace all trajectories in time (depending on the observation period, it could be ± 10 days). So my trajectory today would be compared to others displaced in time. That would provide an estimate of which fraction of inferred interactions are driven by density, etc.

* The effect of a null model that take into account density of nearby individual in time & space, as discussed here <https://www.pnas.org/doi/10.1073/pnas.1006155107> would also be very interesting to see. Especially for understanding segregation at particular venues.

Medium concerns

The definition of economic segregation as the average of everyone with whom they interact might be problematic. The distribution of income is heavy-tailed, so an average might be driven by large values from the tail. This could be remedied by using rank (or equivalent position) in the income distribution rather than absolute dollar amounts. Even if the authors prefer to use income as a measure, I would like to see results for a rank-based measure.

The MS does not make it clear (at least I couldn't find it) how repeated interactions are weighted in the dataset. For example, let's say my partner and I are both in the dataset and see each other every day. Does each observation of my partner count when the average ES of my interactions is calculated? Or does my partner only count once? I would like the authors to comment on how this choice affects the results.

The MS states that the dataset is "anonymous". But in what sense is the data anonymous if we can identify the home-location of a large fraction individuals in the data ... along with everywhere they go? As far as I know these bits of information would make it possible to re-identify a large fraction of users, e.g. through a side-channel attack.

The authors find that POIs such as religious organizations tend to be more segregated. Does this not contradict Chetty et al.'s finding that people tend to befriend more well-off people at a higher rate than expected at random in religious groups? It would be good if the authors could comment on this explicitly in the text (even if there is no contradiction).

Page 6 line 84: "Large cities facilitate segregation". The authors provide no proof of causal effects. A more accurate claim would be "Interaction segregation correlates with city size and population density". This causal claim occurs several times in the paper and should be corrected throughout.

When inferring home location, the authors use interpolation to fill missing data. This seems reasonable (at least for inferring the home location), but it should probably only be up to a maximum number of missing hours between two recorded data points (it would be nice to have a few more details on what's going on here). Also, the authors filter for users that have stationary nighttime observations on at least 3 nights and with at least 60% of observations within 50 meters radius. 3 nights out of how many nights? All the nights from the 3 months of data? This seems like a very small threshold to use for inferring home location.

There is a section called: "Mechanisms producing higher interaction segregation in larger metropolitan areas", where the authors show that there is an association between the density of amenities and the level of segregation. I see what the authors are hinting at and there are nice examples, but perhaps "mechanism" is a bit strong. The authors do not provide any empirical evidence that higher density of amenities make cities more segregated (it could be the opposite, or it could be that there is an underlying third phenomenon that drives both). Nor they develop a mechanistic model that offers an explicit possible explanation for the observed association. As a final small point on this, here it is even unclear if here, the authors are controlling for the economic diversity in a given city.

Smaller notes

Fig 1a is difficult to understand (at least in the eyes of this reviewer) and could be omitted.

I didn't read this paper carefully, and I see that it is cited, but based on my superficial knowledge, this paper does look at mixing at individual locations in a way that is very close to what happens here <https://www.nature.com/articles/s41467-021-24899-8>

Fig 1b: It is very difficult to see the difference between the red and orange color in this plot.

Caption Fig 2: "Analysis limited to counties with at least 50 individuals". I assume that the authors refer not to the population of counties, but to the number of users present in the dataset for that county. This should be stated more clearly.

Fig 3cd What is the scale here?

Fig 3d: Up to 100m random noise added to preserve anonymity. Is this sufficient -- this is difficult to know without an indication of scale?

Page 10: The sample is representative on many key parameters. The reference says well-sampled to SafeGraph uses the word well-sampled and "representative" seems to be a stretch. The authors are use a more careful choice of words in Section M1, Methods.

Referee #2 (Remarks to the Author):

This was an impressive paper using mobile data to create an improved metric of contact across economic boundaries. Theoretically, most folks were I think aware of limitations of using residential area to estimate contact, and this paper crosses an important technical hurdle to track folks as they go about their days, inferring contact from proximity.

Overall I thought the writing clear, analysis appropriate, and the supporting documents extensive and well organized, I left with thinking I had a clear idea of the idea of the pipeline and what the authors did.

Because of this, my comments are fairly minimal and more invitations for the authors to speculate about the generality/limitations of their results.

Major and minor concerns:

-The main thing I found myself wondering about was the low population areas excluded for the various methodological reasons, and what might be different about the conclusions should these areas be included. The authors find a negative relationship between city size and contact, but I wonder if the evidence for cosmopolitan mixing would come from comparing rural areas (excluded from this analysis) to small/mid-sized cities. This would suggest that there is an overall curvilinear effect of contact on city size when the full range is considered. I think worth it for the authors to mention something about whether they believe this is a possibility (or not) in the general discussion, as it indicates whether the cosmopolitan mixing hypothesis is just totally refuted or it's more a restriction of range problem.

-There are limitations to using residential area to infer interactions, and this undoubtedly seems a strong improvement. Yet there are likely limitations to using spatial proximity to inferring interactions. The authors mention that limitations exist, but don't really say what they are. The contact literature has shown that high quality interactions (romantic partner, longer conversations) have the greatest impact on attitudes toward folks in other groups. I'm thinking of scenarios in which folks might share proximity but lacking meaningful interactions that actually shape attitudes. For example, there is work showing that higher economic status folks look less at homeless people as they pass them on the sidewalk (P Dietze, ED Knowles - Psychological Science, 2016). Or ordering coffee from someone, and the interaction constitutes a service role. A few lines about what the limitations actually are would be useful.

-Minor, but a few times in the statistical section, like on line 430 on page 23 the relationship between an independent variable and dependent var in a mixed effects model is referred to as a Pearson correlation. And later on line 445. It's not a Pearson correlation if it's in a multilevel model, especially when covariates are added to the models later (shared variance partialled out). This can be fixed by just calling this the relationship or association between variables, rather than the Pearson's correlation. Maybe this is a jargon difference between disciplines, but my understanding is Pearson's correlation is just linear relationship between single IV and DV.

I appreciate the opportunity to review this interesting paper.

Referee #3 (Remarks to the Author):

The paper, "Human interaction networks reveal that large cities facilitate segregation", uses cell phone tracking location data to measure to what extent individuals of different socio-economic status cross paths.

The authors first use nighttime location to link individuals to their home and measure an individual's socio-economic status (called economic standing, ES, by the authors) by the rent estimate of their home.

They then define an interaction as taking place whenever two individuals spend time within 50 meters of each other within 5 minutes.

Based on these measures, the authors compute "interaction segregation", a measure of the socio-

economic segregation of individuals crossing paths in a given location. They compute this measure for MSAs across the US, as well as for other locations (called point of interests), such as restaurants, parks, or museums. The paper finds that interaction segregation is higher in larger and denser cities. Using restaurants as an example, they show that in larger and denser cities, individuals have more restaurant choices nearby and restaurants are more stratified by clientele socio-economic status. The authors conclude that interaction segregation is higher in large and dense cities because they are large enough to offer a bigger variety of settings stratified by clientele socio-economic status making it easier for individuals to primarily attend settings attracting people similar to them.

Overview of comments

The paper is well-executed and establishes its findings convincingly. My main concern is that there already exists a large body of work looking at similar outcomes with similar data and the additional insight from the paper at hand is not quite novel enough for a top general interest journal. As I outline below, other work has used GPS data to measure various forms of segregation (mostly focusing on racial) and finds similar results. In addition, I would have liked to see more discussion on what types of interactions are measured and to what extent they are relevant for various outcomes.

Relation to other work using GPS phone data

The paper is part of a growing body of work using GPS data to measure integration and the extent of interaction between different types of individuals.

The MIT media lab has an online tool that shows measure of the extent of inequality for different places across the US using a measure very similar to the current paper. Relative to the tool, the current paper covers a much larger number of MSAs and locations.

The closest related paper that comes to mind is Athey et al (2020), who use similar GPS data to measure what they term "experienced isolation" between white and black Americans. They find that experienced racial isolation is generally lower than standard segregation measures suggest and that it is higher for interactions close to home and lower in places like public parks. The current paper's result mirrors these findings for economic, rather than racial segregation.

The current paper also shows how points of interests that lie between neighborhoods of different demographics can serve as bridges to foster interactions of different types of individuals. A similar idea focusing on NYC parks was recently presented by Abbasov (2021) focusing on racial segregation rather than economic segregation.

In short, the paper's approach and finding are in line with other recent work in this space and reflects those cutting edge data work and methods.

Interactions?

The paper calls its key measure interaction segregation. An interaction is defined as coming within 50 meters of a person within 5 minutes. I would have liked to see a deeper discussion of what type of interactions are captured by this and to what extent and in which settings they matter.

Specifically, the current measure captures whether I cross paths with someone. There is no way of telling with the current data whether two individuals simply spend time in the same restaurant or coffee shop or whether a deeper interaction such as speaking to each other took place.

The reason we care about segregation and interactions of different people is generally because past research has shown that interactions between different types of people are desirable for many outcomes such as economic mobility, health and reduced political polarization (as cited in the paper). So it would be useful to discuss what different types of interactions accomplish and how they help improve different outcomes.

It may be that crossing paths with different types of people is sufficient for reaping the benefits of

increased interactions. Maybe seeing people of different means in the same settings one attends makes them appear less other and helps fostering understanding across class lines, potentially reducing political polarization. However, simply being in the same spaces as others may not actually be sufficient to lead to some of the benefits of reduced segregation. Maybe actual interactions, such as speaking to each other and the exchange of ideas, are necessary to reap those. For instance, recent work by Chetty et al. (2022) suggests that online friendship links, not just exposure, to those of different socio-economic status are associated with higher social mobility.

In short, it would be helpful to show that the reduced extent of crossing paths interactions in bigger and more dense cities are truly associated with less of the relevant mixing across class lines that matters for outcomes. Or whether the increased "interactions" in smaller cities just capture more paths crossings of the type that does not lead to improved outcomes.

It should be noted that the paper provides substantial robustness in how an interaction is defined, using more restrictive definitions with respect to time, length and distance and repeated interactions. Still, a concern is that even though I might pass the same person daily on my way to work, I may never actually talk to or more intensely interact with them.

It may nevertheless be helpful to characterize interactions more in terms of length, distance, repetition and how these characteristics differ across settings and MSAs to better understand which types of crossing paths might be associated with more or less additional interactions (such as speaking) and whether that relates to outcomes.

Mechanism and relationship with race

Related to what types of interactions are captured in the paper, I would have liked to see a bit more discussion on the mechanisms or drivers of the results. There currently is very little in the paper beyond the result on bridging zones.

Given the related work on racial segregation, it would have also been interesting to look more closely at whether there are interaction effects of race and socioeconomic standing. It is well-known that race and socio-economic standing correlate. So it would be very interesting to see to what extent segregation by race and income are driving each other. Is there more interaction across the socio-economic spectrum within race? Are income and racial segregation similar across settings or are some settings more segregated by race or income (e.g., one could imagine restaurants to be more segregated by income)?

Response to all referees

We thank all of the referees for their valuable feedback. We first address two key themes in the reviews: (1) how our measure allows for a direct test of the cosmopolitan mixing hypothesis, and (2) why our findings and methods are a significant advance over previous work. We follow with a summary of revisions and then respond to the individual referees.

Measure of Exposure

It is important to first clarify that the cosmopolitan mixing hypothesis [1-4] pertains to *visual exposure* (e.g., sightings) and not deeper interactions (e.g., speaking). As one of its progenitors, Louis Wirth, writes in *Urbanism as a Way of Life* (1938):

“The heightened mobility of the individual... brings him within the range of stimulation by a great number of diverse individuals... Typically, our physical contacts are close but our social contacts are distant. The urban world puts a premium on visual recognition.”

The cosmopolitan mixing hypothesis has become so entrenched in our understanding of the world that it’s sometimes stated as fact despite a lack of evidence on its behalf. The main unresolved question is whether urban life actually exposes us to diverse people. Given the structure of the hypothesis, it is unproblematic that the encounters we capture are happenstance (R1) or that we cannot determine whether people are holding conversations with one another (R3). Indeed, to test the cosmopolitan hypothesis, it is *necessary* to capture happenstance exposures and include *all* exposures—regardless of whether individuals spoke to each other or simply walked past each other. To clarify this point, we adopt R1’s suggestion to replace the terminology “interaction” with “exposure” in our revision.

Exposure is important because it affects mental health, happiness, political views, and much more [5-17]. It is also a *necessary prerequisite* for deeper forms of interaction. We validate that our measure of exposure is also linked to deeper interactions by showing that it strongly predicts (Spearman Corr 0.73-0.86) friendship ties from a large-scale external dataset compiled by other scholars [18] (**Extended Data Figure 1**, copied below). For instance, our exposure network explains 76.3% of the variation in friendship between counties, whereas distance explains only 53.9% of the variation (**Supplementary Tables S23-S24**). Furthermore, we adopt R3’s suggestion and expand our analysis of the associations between exposure segregation and downstream outcomes (which now includes political polarization, in addition to economic mobility, which was included in our initial manuscript; response pp. 77-81).

Our work is able to directly test the cosmopolitan hypothesis because our temporal co-location measures are exceedingly fine-grained (i.e., as fine-grained as two people being 10 meters apart within 60 seconds; Supplementary Figures S8). By contrast, all prior work uses much looser

definitions of co-location. For instance, a study of close family ties published in *Science* defines exposures as co-location within a 3x3 mile vicinity [19]. Just as problematically, Athey et al. consider two people exposed to each other if they *ever* visited the same 153 x 153m grid cell anytime within the 4 month observation time [20], an approach that leads to serious measurement error (a point we address in the next section.).

Advances over prior work

Our investigation of the cosmopolitan mixing hypothesis – which is long-standing and yet has never been well assessed – is our main contribution. This work overturns the conventional wisdom on urban life and challenges Athey et al.’s conclusions [20]. Our key discoveries are that:

- (1) Large, dense cities are *more* segregated (thus refuting the long-standing cosmopolitan mixing hypothesis);
- (2) Venue differentiation accounts for this increased segregation;
- (3) Citywide segregation is significantly reduced when city hubs (e.g., shopping malls) are positioned to bridge diverse neighborhoods.

All prior studies (besides Athey et al.) are unable to assess these questions because they only measure exposure diversity in one or a small number of cities, and thus cannot draw nationwide comparisons across cities. Athey et al. study multiple cities but only briefly examine the cosmopolitan mixing hypothesis (for cross-race exposure) in a side analysis. However, they arrive at the *opposite* conclusion as our study:

“Experienced isolation is relatively lower in MSAs with higher population density... consistent with the fact that in dense areas residents from different neighborhoods are less separated by physical space, and may reflect the role of urban amenities such as parks and public facilities in facilitating diverse interactions (Jacobs 1961).”

We have included an extended discussion of why Athey et al. arrive at this opposite conclusion in our response to R3 (response pp. 68-70) and in Extended Data Figure 9. We have also now highlighted this key distinction in the results section of our main manuscript (p. 5, lines 69-71). We thank R3 for their suggestion to use our exposure network to also investigate racial exposure segregation, as this allows for direct comparison to Athey et al.

In summary, Athey's conclusion is an artifact of (1) an imprecise (time-indifferent and low-resolution) measure of exposure, and (2) downweighting of small cities in their analyses (in their regression analysis they weight cities by population size, which means that the regression essentially fits to only big cities and ignores the overall relationship; more details in response pp. 68-70). When we instead use our more precise (time-sensitive and high-resolution) measure of exposure and examine *all* cities, the correlation between population density (as well as population size) and segregation becomes positive, strong, and *very* robust. This result holds across different co-location thresholds (10/25/50 meters distance), time thresholds (1/2/5 minutes apart), tie strengths (repetition and duration), segregation types (SES or racial), and area granularities (MSA or county).

Our findings rely on several methodological contributions:

- (1) Building a nationwide, person-to-*person* exposure network of (time-sensitive and fine-grained) exposures across 382 MSAs and 2829 counties. Prior work (including Athey et al.) only constructs static person-to-*place* networks, which are different from a person-to-person network because they are low-resolution, and indifferent to time and tie strength;
- (2) Measuring socioeconomic status at the individual-level (via individualized rent estimates) rather than imputing it from neighborhood averages (as prior scholars have);
- (3) Securing unbiased estimates of exposure segregation even in small cities and towns with limited data (via a mixed model approach that corrects for network sparsity).

Based on our robust results, we have no choice but to reject the cosmopolitan mixing hypothesis (Figure 1c-d). In our revision, we re-ordered our results section to emphasize that this is our key discovery, and condensed our discussion of other exploratory analyses (Figure 2a-c).

We are also the first to explain why the hypothesis falls short; our data show that, as a city scales, segregation increases as spaces become more differentiated to target specific socioeconomic groups (Figure 2d-f; e.g., Michelin-star restaurants). Contrary to what Athey et al. argue, we likewise show that big cities have more small parks localized to each neighborhood and, by extension, targeted to specific socioeconomic groups. We discover that this differentiation of urban spaces, which also occurs at higher levels of scale (i.e., city hubs and neighborhoods become more differentiated too), accounts for increased segregation in big cities.

As our third main contribution, we explore how urban design can overcome this perverse trend (Figure 3). By contrast, Athey et al. do not explore any strategies to reduce segregation. We discover a tractable solution that could be directly implemented through changes in urban development

policy. Because city hubs (e.g., shopping malls) are few in number and high in traffic, when they are positioned as bridges between diverse neighborhoods, citywide segregation is significantly reduced. This implies that interventions to build bridging hubs (i.e. zoning laws or subsidies) may counter venue differentiation and ensure that cities *do* deliver the desired mixing. We showcase several large cities that could serve as role models, as their hubs are (perhaps unwittingly) already positioned in such bridging locations and account for an increase in socioeconomic mixing. These findings motivate future research to assess the causal effects of building such bridging hubs.

The upshot: Our paper not only reveals that cities aren't delivering the diverse exposures that we've long expected them to deliver, but also offers a solution to circumvent this problem and thus reintegrate our cities. Although we didn't feature Athey et al.'s inconsistent side analyses very prominently in our writeup (as Athey et al. didn't make much of them), we have now incorporated a larger writeup in our Supplementary Materials that shows why Athey et al. were led astray.

Summary of revisions

In response to the referees' suggestions, we have also conducted several new analyses and made substantial edits to the manuscript to clarify our dataset, methods, and the novelty of our findings. The most important revisions to our manuscript are that we:

- Investigate the sparsity of our underlying network and our ability to draw statistical comparisons between large and small cities (R1)
- Describe the properties of our underlying exposure network in detail (R1)
- Conduct new robustness checks to account for background density, stationary pings, and varying thresholds for home identification (R1)
- Demonstrate that the cosmopolitan mixing hypothesis can be rejected even in rural areas (R2)
- Distinguish the types of ties captured within our network, and analyze the relationship between socioeconomic segregation, race, and downstream outcomes (R3)
- Differentiate our primary novel findings from secondary exploratory analyses which we conducted to provide a holistic overview of exposure segregation (R3)
- Clarify the terminology in our manuscript, replacing "interaction" with "exposure," and "economic standing" with "socioeconomic status" (R1, R3)

The resulting revision provides novel insights about everyday segregation using terabyte-scale location data. We believe this paper will be of interest to a broad community of scholars and policymakers interested in taking on one of the most pressing inequality problems of our time.

We have, in total, conducted **36** new analyses and included **59** new figures in the revised manuscript. Our response is structured as follows:

- Response to Referee 1 (page 5)
- Response to Referee 2 (page 53)
- Response to Referee 3 (page 58)
- References (page 91)

Referee comments are in boxes, followed by our response to each comment. **Bolded blue** denotes a new analysis or figure. **Bolded black** indicates an analysis previously included in the original manuscript. All page and figure numbers refer to the newly revised manuscript.

Responses to Referee 1

"Human interaction networks reveal that large cities facilitate segregation" is a well-written and impressively researched manuscript which offers up an interesting new measure of segregation, which leads to a number of interesting findings, most notably that large cities tend to have higher segregation than surrounding areas, contrary to what has been speculated in the literature. It was a pleasure to read the excellent main text and delve into the impressive SI, but before I can recommend publication I have a number of serious concerns about the network of interactions underlying the excellent analysis presented in the paper. The construction of this network (and the network itself) is not sufficiently well described in the paper/SI. And fully understanding the network is important as the entire downstream analysis rests upon that foundation. Understanding the network is important both to ensure that there are no potential biases - and for the interpretation of the results. Concerns around the network also extend to parts of the presentation (for example I think that given the underlying network, the words "interaction" and "contact" need to be eliminated from the text, incl the title). Below I elaborate in detail.

We thank R1 for such generous observations. We also appreciate the opportunity to address R1's concerns here. In the comments that follow, we take on each of these concerns, as well as others that R1 has raised. We believe we have satisfactorily addressed all of the issues raised.

"While the number of individuals in this dataset is truly impressive, we still only have access to a small fraction (let's say $\sim 2.5\%$) of all Americans. When one has access to only a small fraction of nodes, that implies access to a much smaller fraction of the edges (in the random case, the fraction of edges we sample scales as the fraction of nodes squared) $\sim 0.06\%$ of all links. Combine that with the fact that the data (presumably) was collected over time which introduces further sparsity... It is very difficult to gauge which fraction of the inferred data is real and which is noise, since the network is not reported on at all. All we have access to are the derived quantities. And these derived quantities can make a very sparse dataset appear quite dense. To calculate the mean ES of people someone has interacted with, for example, you only need a single datapoint."

R1 raises an important point that the quality of our analysis depends critically on the quality of the underlying exposure network. Notably, a sparse exposure network can potentially result in (1) *biased* and (2) *noisy* estimates of exposure segregation, both of which are issues that we treat with care in our study.

To address these two points, we have revised our manuscript to elaborate on

- (1) how our methods produce unbiased estimates of exposure segregation, and
- (2) new analyses in which we quantify the precision, meaning the robustness to noise, of both our MSA-level exposure segregation estimates and macro-level correlations.

We elaborate on these two points in the pages below. We demonstrate that our methods allow for unbiased estimates of exposure segregation even in small cities with limited data. We also show that our network size is sufficient to compare large and small cities and to assess the cosmopolitan mixing hypothesis. The results show that larger cities are significantly more segregated ($p < 0.001$) across the board (i.e., when network nodes are re-sampled or downsampled).

(1) *Unbiased estimates of exposure segregation.*

Our mixed model approach enables unbiased estimates of exposure segregation even in MSAs with limited data. Although our original manuscript briefly addressed issues of *bias* in the section titled “Estimating exposure segregation” (**Methods M3**, pp. 25-27), we have now revised the main section of the manuscript to include a more detailed discussion of our mixed model and the rationale for fitting it (p. 2, lines 41-42). Because this model (copied below) corrects for network sparsity, we ought to have emphasized it more in the main writeup (rather than relegating it to the methods section). We have now done so.

$$y_{ij} = ax_i + b + \epsilon_i^{(1)} + \epsilon_{ij}^{(2)}$$

where x_i = SES of person i

y_{ij} = SES of person j who was exposed to person i

a, b = model parameters

$\epsilon_i^{(1)}$ = person-specific noise term

$\epsilon_{ij}^{(2)}$ = noise for each data point

In summary, our mixed model (**Methods M3**, pp. 25-27) corrects for the effects of sparsity on the correlation between an individual’s SES and the mean of SES of those they are exposed to (which is used to characterize the extent of socioeconomic homophily within a geographic area).

As R1 observes, directly computing correlations with the sample mean SES (per person) is problematic because it will result in noisy estimates when an individual has a lower number of edges in the network. Thus, *we do not directly calculate the sample mean SES for each person*, because doing so (and then relying on uncorrected Pearson correlations) would result in estimates of exposure segregation that are *downwardly biased* in MSAs or counties with limited data.

Instead, we explicitly model the *true* mean SES of the exposure set for each person as $ax_i + b + \epsilon_i^{(1)}$. Individual exposures $y_{i,j}$ are then modeled as noisy draws from a distribution centered at this true mean. We then compute the correlation between each person’s SES and the modeled true mean (rather than the estimated noisy mean).

$$\text{corr} \left(x_i, ax_i + b + \epsilon_i^{(1)} \right)$$

We show that this corrects for the downward bias sparsity induces (**Methods M3, Figure 1**, copied below).

Methods Figure 1: Our estimates compared with naive estimates of the Pearson correlation. We took people who crossed paths with at least 500 other people and computed the Pearson correlation coefficient (the “gold standard estimate”). Then, for each person we randomly sampled 5, 10, 50, 100, and 200 people from the 500+ people and computed segregation estimates based on the reduced sets of people. The left plot shows the ratio of the estimates to the gold standard, for each MSA. The right plot shows the overall number of people in the dataset with $\leq N$ exposures.

(2) Precise estimates of exposure segregation and subsequent findings

We agree with R1 that issues of sparsity are important enough to include more discussion and more analyses of its possible effects. While our original manuscript addressed the effect of network sparsity on *bias* (addressed above), we did not initially address the effect of network sparsity on the *noise* of our exposure segregation estimates and key findings. We have now done so, and included a number of new analyses on this topic. The analyses show that our exposure segregation estimates and key findings are indeed quite precise.

Below, we include additional analyses to quantify the precision (i.e., robustness to noise) of two key estimates:

- (a)** Our MSA-level exposure segregation estimates; and
- (b)** Our primary findings (i.e., nationwide correlations between population size, exposure segregation, and hub bridging)

The first figure quantifies the precision of our segregation estimates for each individual MSA, via 95% bootstrapped confidence intervals obtained by re-sampling network nodes (Supplementary Figure S24). Confidence intervals between the largest and smallest MSAs are non-overlapping, meaning that we have sufficient data to be able to compare exposure segregation estimates between large MSAs and small MSAs.

The second figure illustrates the precision of our macro-level correlations, by downsampling the network nodes and re-computing the correlations between population size, exposure segregation, and hub bridging (Supplementary Figure S25). Asymptotic convergence of both correlations under downsampled networks shows that our network size is more than sufficient to estimate the value of these correlations. This means that we have sufficient data to be able to test the cosmopolitan hypothesis.

(2a) Precision of exposure segregation estimates (Supplementary Figure S24, copied below)

We interpret R1’s feedback, in part, as suggesting that our initial manuscript could better clarify the precision (i.e., robustness to noise) of the derived quantities within our analysis (i.e., our estimates of exposure segregation). In addition to better reporting on our underlying exposure network (in subsequent responses below), we have now revised our manuscript to provide readers with information on the precision of our segregation estimates (via confidence intervals).

Because our dataset is so large, we did not initially provide confidence intervals for our estimates. We have now computed 95% bootstrap confidence intervals for exposure segregation in each MSA (N=1000 replications, sampling nodes with replacement, estimating segregation via our mixed model). These illustrate the precision of our exposure segregation estimates and establish that they allow us to assess the cosmopolitan mixing hypothesis.

As Supplementary Figure S24 (a) (copied below) indicates, the intervals may overlap among very large MSAs (i.e., the 10 MSAs with the most individuals) and among very small MSAs (i.e., the 10 MSAs with the least individuals). However, it is clear that large MSAs are much more segregated than small ones, a result that is again consistent with our main conclusion that the cosmopolitan hypothesis is off the mark. In Supplementary Figure S24 (b), we compare the mean ES of large and small MSAs and find that the mean of the top 10 is higher across all 1000 replicates ($p < 0.001$).

(2b) Precision of our primary findings (Supplementary Figure S25, copied below)

R1 also raises concerns about sampling only a small *proportion* of the nodes in the network (as this further limits the number of exposure edges observed, due to the exponential relationship between the # of observed nodes and # of observed edges in the network). Here, we examine the effect of the size of the network sample on our primary findings.

To do so, we downsample the nodes in the network and recompute our key findings using only the downsampled network. The main result: both correlations are statistically significant ($p < 0.001$) with as little as 10% of nodes sampled. When 50% of nodes are sampled, the correlations for our primary results converge to approximately 0.6 (correlation between exposure segregation and population) and approximately -0.8 (correlation between exposure segregation and hub bridging). This asymptotic convergence suggests that our estimates of these macro-level correlations are precise, i.e., we would expect essentially the same findings if we (i) expanded the sampled network in our study beyond the current dataset, or even if we (ii) analyzed a smaller exposure network (i.e., correlations already begin to stabilize with as little as 30-40% of our data).

In the case of many datasets, the majority of raw location data collected by smartphone apps capture individuals' positions while people are on the move (this is because while they move, location is collected at higher frequency, due to people e.g. using navigation apps). The authors do not describe how location data is sampled temporally in their dataset (more on this below), but I don't have a reason to assume that it is sampled only when people are still. Thus, it is possible that many of the interactions detected occur while people are driving by each other.

We also thought long and hard about whether to include exposures while people are driving. In the end, we decided to include them in our primary measure, on the argument that “cars are like clothes” and that exposure to different types of cars (e.g., a Taurus vs. a Tesla) may be consequential (just as it matters whether one is exposed to people dressed in Gucci, Givenchy, or Prada). But we also ensure that our findings are robust to this decision.

In our initial manuscript we did, therefore, include a robustness check, which excludes exposures occurring on roads (**Supplementary Figure S4**, below). This check (copied below) showed that our findings were similar when using this definition of exposure, which is highly correlated with our primary measure (Spearman Corr 0.98).

We do appreciate R1’s very reasonable concern about this point and have now added an additional robustness check (below) to include only stationary individuals (new row in Supplementary Figure S4, below). To do so, we only consider exposures for when individuals are *stationary* (defined as 2 pings < 10 meters apart, between 1-10 minutes apart). This measure is also highly correlated with our primary measure (Spearman Corr. 0.87) and further highlights the robustness of our key results.

"Authors adapt a generous definition of "interaction", namely that two individuals "interact" if they are observed less than 5 mins apart and within 50 meters of one another. This definition

a) does not necessarily capture all real social links

b) very likely captures many "interactions" that are not real social interactions in any meaningful sense (since the dataset has a lot of interactions: 1.6 billion interactions and 9.6 million nodes).

I would like the authors to comment on the arguments above and provide a discussion of which fraction of their measured interactions correspond to underlying social relationships (e.g. some kind of preexisting relationship: two people would consider each other friends, or know of one another's existence, or know each other's name, etc.)

All this also implies (to me at least) that a terminology along the lines of "exposure" would be more honest. (I think what the authors are measuring might be meaningful ... it's just not social interactions. More of a measure of social mixing in space and time."

We appreciate R1's questions about (1) whether it is more appropriate to refer to the edges in our network as "exposure" and (2) to what extent edges within our network correspond to pre-existing social relationships (e.g., acquaintances) rather than happenstance mixing (e.g. individuals in the same café). We first address these questions, and then (3) show that the exposures we capture are socially meaningful in that they strongly predict externally validated friendship outcomes (Spearman Corr. 0.73-0.86). Finally, we (4) break down our exposure network by tie strength, showing that our key findings are robust to varying thresholds of tie strength.

(1) Terminology of "interaction" has been changed to "exposure"

We agree with R1 that the term "exposure" will clarify that our aim is to assess the cosmopolitan mixing hypothesis, which concerns *all* visual exposures in large cities (including everything from happenstance path-crossings to pre-existing relationships). We have thus rewritten the manuscript to replace "interaction" with "exposure" throughout. We thank R1 for highlighting this concern.

(2) Relationships captured within our network

As one would expect, the *vast majority* of the exposures captured are happenstance and not pre-existing social relationships (response pp. 16-19, 74-76). Capturing these happenstance exposures is an important part of our study design, as it is precisely these exposures which large, dense cities are hypothesized to influence (by constraining space and pushing individuals into unplanned exposure with one another). This type of happenstance exposure is of long-standing interest to social scientists for three reasons:

- (a) it has short-term and long-term effects on the social psychology of group relations [5, 6, 7], emotional well-being [10, 11, 12], political view formation [13], transmission of infectious diseases [8], vulnerability to crime [9], and more [14, 15, 16, 17];

- (b) the U.S. has a longstanding normative commitment to integration that values intermixing in and of itself [21, 22] and thus calls for high-quality measurement of the extent to which we're living up to that commitment; and
- (c) happenstance interactions are a *prerequisite* for many deeper social relationships, such as friendships.

(3) Exposure strongly predicts friendship formation

To further demonstrate that our measure of exposures is meaningful, we've shown that it predicts important external outcomes. In our initial manuscript, we showed in **Extended Data Figure 1** (copied below) that our exposure network strongly predicts friendship formation between zip codes and counties, using the Facebook social connectedness index (Spearman Corr. 0.73-0.86).

In our revision, we further investigate these findings. Most notably, we show that our exposure network is a stronger predictor of friendship formation than distance ($p < 1e-4$, Steiger's Z-test). For instance, our exposure network explains 76.3% of the variance in friendship formation between counties, whereas distance explains only 53.9% of the variance (Supplementary Tables S23 and S24; S23 is copied below).

	Dependent variable: Log(FB Social Connectedness)		
	(1)	(2)	(3)
const	11.123*** (0.003)	11.123*** (0.002)	11.123*** (0.002)
Log(Distance km)	-1.163*** (0.003)		-0.253*** (0.003)
Log(Exposure Network Social Connectness)		1.383*** (0.002)	1.190*** (0.003)
Observations	118,559	118,559	118,559
R^2	0.539	0.763	0.773
Adjusted R^2	0.539	0.763	0.773

* $p < 0.1$; ** $p < 0.05$; *** $p < 0.01$

The results above are intuitive because exposure is a necessary prerequisite to friendship formation and all other deeper forms of social interaction. This further motivates our focus on cross-class exposure, as increasing cross-class exposure is the first step towards increasing cross-class conversations, friendships, and so forth.

(4) Breaking down exposure by tie strength

Although our focus is on overall exposure, we do consider the subsets of our exposure network in which relationships are stronger. We observe that our key findings are robust to the strength of the exposure. In our Supplementary Information, we isolate:

- Prolonged exposure – multiple consecutive co-located pings on the same day
- Repeated exposure – co-located pings on multiple distinct days

Repeated exposure indicates a higher likelihood of individuals being acquainted with each other. Although repeated exposures will encompass ties that range from passing familiarity (e.g., sharing the same bus ride to work each morning) to bona fide friendships (e.g., golf partners), the level of familiarity is clearly increased from the one-off exposures that constitute the majority of ties in our network.

We agree with R1 that we should have provided more information on the different types of ties that our exposure measure encompasses. To this end, we have added columns to Supplementary Table S6 (copied below) which display the % of Pairs and % People retained after applying these filters to the exposure network.

Feature	Pearson Corr. w/ Primary	Spearman Corr. w/ Primary	Median	Mean	% Pairs	% People
Primary Measure	—	—	0.35	0.35	100.00	100.00
Primary Measure (+ Up-weight Multiple Exposures)	0.89	0.91	0.46	0.45	100.00	100.00
SES Definition: Rent Zestimate Percentile	0.88	0.89	0.42	0.42	100.00	100.00
SES Definition: Within-MSA Rent Zestimate Percentile	0.81	0.83	0.54	0.53	100.00	100.00
SES Definition: Census Median Household Income	0.75	0.77	0.47	0.46	100.00	100.00
SES Definition: Educational Attainment (% College or Higher)	0.70	0.71	0.52	0.52	100.00	100.00
Exclude Pri/Sec Roads	0.99	0.99	0.37	0.37	74.30	99.87
Exclude Roads	0.98	0.98	0.37	0.37	38.66	98.56
Stationary Individuals (2 pings < 10 meters in 1-10 min)	0.86	0.87	0.44	0.43	5.39	79.46
Exclude Same-home exposures	0.98	0.98	0.34	0.34	99.71	99.78
Work/Leisure (Neither in Home Tract)	0.93	0.93	0.31	0.31	86.26	95.55
Leisure (inside POI)	0.85	0.84	0.28	0.29	16.15	76.32
Minimum Distance Between Pings: < 25 meters	0.97	0.97	0.34	0.34	49.53	97.71
Minimum Distance Between Pings: < 10 meters	0.95	0.94	0.33	0.33	20.93	92.28
Minimum Time Between Pings: < 2 minutes	0.97	0.97	0.34	0.34	53.59	98.92
Minimum Time Between Pings: < 60 seconds	0.97	0.97	0.35	0.35	33.67	97.83
Minimum Tie Strength: 2 consecutive exposures	0.94	0.95	0.35	0.35	18.25	94.80
Minimum Tie Strength: 3 consecutive exposures	0.83	0.83	0.37	0.37	3.06	65.46
Minimum Tie Strength: 2 unique days of exposure	0.88	0.90	0.47	0.46	7.46	83.56
Minimum Tie Strength: 3 unique days of exposure	0.73	0.77	0.56	0.54	2.32	62.81
Dist. < 25 meters, Time < 2 min., >= 2 consec. exposures	0.92	0.93	0.36	0.35	8.80	86.64
Dist. < 25 meters, Time < 2 min., >= 2 unique days	0.87	0.89	0.46	0.44	3.84	70.57
Dist. < 10 meters, Time < 60 sec., >= 3 consec. exposures	0.78	0.79	0.39	0.38	0.94	38.16
Dist. < 10 meters, Time < 60 sec., >= 3 unique days	0.68	0.68	0.52	0.51	0.58	28.92
Downweight Simultaneous Exposures	0.97	0.97	0.34	0.34	99.99	99.99
Exclude Simultaneous Exposures	0.94	0.95	0.46	0.46	22.87	99.61
Tie Strength: 1 Exposure	0.97	0.98	0.31	0.32	74.24	98.66
Tie Strength: 2 Exposures	0.95	0.96	0.33	0.33	16.14	93.20
Tie Strength: 3 Exposures	0.89	0.90	0.36	0.36	4.09	74.67
Tie Strength: 4 Exposures	0.85	0.86	0.37	0.36	1.82	58.51
Tie Strength: 5+ Exposure	0.83	0.85	0.45	0.45	3.42	70.62
Minimum Stationary Nights: 6 nights	0.99	0.99	0.35	0.35	77.45	83.91
Minimum Stationary Nights: 9 nights	0.99	0.99	0.35	0.36	68.64	73.42
Minimum Stationary Nights: 12 nights	0.98	0.98	0.35	0.36	59.16	63.66
Racial Segregation (% Non-White)	0.59	0.60	0.46	0.46	100.0	100.0
Economic Segregation (White Overall)	0.94	0.95	0.34	0.34	55.97	63.83
Economic Segregation (White Within-group)	0.93	0.94	0.34	0.34	39.84	63.69
Economic Segregation (Non-White Overall)	0.55	0.52	0.28	0.28	43.76	35.50
Economic Segregation (Non-White Within-group)	0.47	0.44	0.28	0.30	27.50	35.31

As one would expect, a minority of relationships fit these stricter thresholds (see % pairs column; red box). This reflects the reality that the vast majority of individuals that a city resident mixes with in their day-to-day life are one-off and ephemeral—and hence very likely to also be happenstance.

We find that the exposure segregation of stronger ties is highly correlated to our primary measure, which includes all exposures (Spearman Corr. 0.77-95; see above table, Spearman and Pearson Corr. w\ Primary columns). Our rejection of the cosmopolitan mixing hypothesis, and discovery of the hub-bridging effect, generalizes to these stronger ties as well (*Supplementary Figures S7-S8*, see next pages). This is expected because, when one’s day-to-day social environment favors homophilous mixing, some of this mixing will “grow” into stronger ties that are also more homophilous.

Minimum Tie Strength: 2 consecutive exposures

Minimum Tie Strength: 3 consecutive exposures

Minimum Tie Strength: 2 unique days of exposures

Minimum Tie Strength: 3 unique days of exposures

Distance: < 25 meters, Time: < 2 minutes, Minimum Tie Strength: 2 consecutive exposures

Distance: < 25 meters, Time: < 2 minutes, Minimum Tie Strength: 2 unique days of exposures

Distance: < 10 meters, Time: < 60 seconds, Minimum Tie Strength: 3 consecutive exposures

Distance: < 10 meters, Time: < 60 seconds, Minimum Tie Strength: 3 unique days of exposures

To help the reader better understand the results, the MS should also provide information about the inferred networks. Are they meaningful? Here, I would like to see

- * distribution of number of contacts per person (all time)
- * distribution of number of contacts per person (for a single day)
- * average/median number of active nodes over time
- * average/median number of contacts per person over time
- * distribution of link strength (how many observations do we typically have for each link)

We agree with R1 that our manuscript should include further details on the underlying exposure network. We did include some of these details (i.e. distribution of number of exposures per person) in our initial manuscript (**Supplementary Table S1**, copied below), but we agree that providing additional details on the inferred network can only strengthen the manuscript.

	Accurate pings	Unique days	Distinct user pairs	Exposures	Accurate pings	Distinct user pairs	Exposures
count	8,609,406				382		
mean	3,273	35	184	363	73,757,695	2,577,322	4,845,144
std	16,507	20	374	1,073	163,848,305	8,872,464	16,838,938
min	11	2	1	1	2,196,084	27,326	53,350
10%	570	13	8	17	8,398,875	140,251	313,803
50%	1,471	30	76	141	22,054,930	504,525	1,031,691
90%	5,857	63	436	785	175,295,175	4,573,152	8,954,800
max	4,755,081	95	42,323	193,193	1,605,070,032	94,140,015	215,183,409

Supplementary Table S1: Combined descriptive statistics for all individuals residing in 382 Metropolitan Statistical Areas (MSAs). 8,609,406 individuals reside in a Metropolitan Statistical Area (90% of the overall 9,567,559 individuals in our study). The remaining 958,153 users live outside of MSAs, influencing the **exposure** segregation of an MSA by coming into contact with MSA residents. Descriptive statistics are grouped by individual (left) and MSA (right). At least one of two users in each **exposure** pair must live in an MSA to be included in this table.

In the following pages, we highlight the new tables and figures, which provide all of the requested information.

distribution of number of contacts per person (all time)	Supplementary Table S8
distribution of number of distinct contacts per person (all time)	Supplementary Table S18
distribution of number of contacts per person (for a single day)	Supplementary Table S10
distribution of number of distinct contacts per person (for a single day)	Supplementary Table S19
average/median number of active nodes over time	Supplementary Figure S26

average/median number of contacts per person over time	Supplementary Figure S27
average/median of number of distinct contacts per person (for a single day)	Supplementary Figure S35
distribution of link strength (how many observations do we typically have for each link)	Supplementary Table S11

Percentile	# of Exposures	Percentile	# of Exposures
0.0	1.0	51.0	146.0
1.0	2.0	52.0	152.0
2.0	4.0	53.0	157.0
3.0	5.0	54.0	163.0
4.0	7.0	55.0	168.0
5.0	8.0	56.0	174.0
6.0	10.0	57.0	181.0
7.0	12.0	58.0	187.0
8.0	14.0	59.0	194.0
9.0	16.0	60.0	201.0
10.0	17.0	61.0	208.0
11.0	19.0	62.0	215.0
12.0	21.0	63.0	223.0
13.0	23.0	64.0	231.0
14.0	25.0	65.0	240.0
15.0	28.0	66.0	249.0
16.0	30.0	67.0	258.0
17.0	32.0	68.0	268.0
18.0	34.0	69.0	278.0
19.0	36.0	70.0	289.0
20.0	39.0	71.0	300.0
21.0	41.0	72.0	312.0
22.0	44.0	73.0	325.0
23.0	46.0	74.0	338.0
24.0	49.0	75.0	353.0
25.0	51.0	76.0	368.0
26.0	54.0	77.0	384.0
27.0	57.0	78.0	402.0
28.0	59.0	79.0	420.0
29.0	62.0	80.0	440.0
30.0	65.0	81.0	462.0
31.0	68.0	82.0	485.0
32.0	71.0	83.0	511.0
33.0	74.0	84.0	539.0
34.0	77.0	85.0	569.0
35.0	81.0	86.0	603.0
36.0	84.0	87.0	640.0
37.0	88.0	88.0	683.0
38.0	91.0	89.0	731.0
39.0	95.0	90.0	786.0
40.0	98.0	91.0	851.0
41.0	102.0	92.0	927.0
42.0	106.0	93.0	1018.0
43.0	110.0	94.0	1133.0
44.0	114.0	95.0	1280.0
45.0	118.0	96.0	1482.0
46.0	123.0	97.0	1781.0
47.0	127.0	98.0	2291.0
48.0	132.0	99.0	3480.0
49.0	136.0	100.0	194243.0
50.0	141.0		

Supplementary Table S8: Distribution of number of exposures for all individuals residing in 382 Metropolitan Statistical Areas (MSAs). The median individual had 141 exposures overall. 8,609,406 individuals reside in a Metropolitan Statistical Area (90% of the overall 9,567,559 individuals in our study). The remaining 958,153 users live outside of MSAs, influencing the exposure segregation of an MSA by coming into contact with MSA residents.

Percentile	# Distinct Exposures	Percentile	# Distinct Exposures
0.0	1.0	51.0	79.0
1.0	1.0	52.0	81.0
2.0	2.0	53.0	85.0
3.0	3.0	54.0	88.0
4.0	3.0	55.0	91.0
5.0	4.0	56.0	94.0
6.0	5.0	57.0	98.0
7.0	6.0	58.0	102.0
8.0	7.0	59.0	106.0
9.0	8.0	60.0	109.0
10.0	8.0	61.0	114.0
11.0	9.0	62.0	118.0
12.0	10.0	63.0	122.0
13.0	11.0	64.0	127.0
14.0	12.0	65.0	132.0
15.0	13.0	66.0	137.0
16.0	14.0	67.0	142.0
17.0	15.0	68.0	148.0
18.0	17.0	69.0	154.0
19.0	18.0	70.0	160.0
20.0	19.0	71.0	167.0
21.0	20.0	72.0	173.0
22.0	21.0	73.0	181.0
23.0	23.0	74.0	188.0
24.0	24.0	75.0	196.0
25.0	25.0	76.0	205.0
26.0	27.0	77.0	214.0
27.0	28.0	78.0	224.0
28.0	30.0	79.0	234.0
29.0	31.0	80.0	245.0
30.0	33.0	81.0	257.0
31.0	34.0	82.0	271.0
32.0	36.0	83.0	285.0
33.0	38.0	84.0	300.0
34.0	40.0	85.0	317.0
35.0	41.0	86.0	336.0
36.0	43.0	87.0	357.0
37.0	45.0	88.0	380.0
38.0	47.0	89.0	406.0
39.0	49.0	90.0	436.0
40.0	51.0	91.0	471.0
41.0	53.0	92.0	511.0
42.0	55.0	93.0	560.0
43.0	58.0	94.0	620.0
44.0	60.0	95.0	695.0
45.0	62.0	96.0	796.0
46.0	65.0	97.0	939.0
47.0	67.0	98.0	1171.0
48.0	70.0	99.0	1655.0
49.0	73.0	100.0	42323.0
50.0	76.0		

Supplementary Table S18: Distribution of number of distinct exposures for all individuals residing in 382 Metropolitan Statistical Areas (MSAs). The median individual had 76 exposures overall. 8,609,406 individuals reside in a Metropolitan Statistical Area (90% of the overall 9,567,559 individuals in our study). The remaining 958,153 users live outside of MSAs, influencing the interaction segregation of an MSA by coming into contact with MSA residents.

Percentile	# of Exposures	Percentile	# of Exposures
0.0	1.00	51.0	7.10
1.0	1.00	52.0	7.27
2.0	1.29	53.0	7.44
3.0	1.50	54.0	7.62
4.0	1.60	55.0	7.81
5.0	1.73	56.0	8.00
6.0	1.86	57.0	8.20
7.0	2.00	58.0	8.40
8.0	2.00	59.0	8.62
9.0	2.17	60.0	8.83
10.0	2.27	61.0	9.06
11.0	2.38	62.0	9.30
12.0	2.50	63.0	9.54
13.0	2.57	64.0	9.80
14.0	2.67	65.0	10.06
15.0	2.77	66.0	10.33
16.0	2.88	67.0	10.62
17.0	3.00	68.0	10.93
18.0	3.05	69.0	11.25
19.0	3.17	70.0	11.57
20.0	3.26	71.0	11.93
21.0	3.36	72.0	12.29
22.0	3.47	73.0	12.68
23.0	3.57	74.0	13.09
24.0	3.67	75.0	13.52
25.0	3.77	76.0	14.00
26.0	3.88	77.0	14.49
27.0	4.00	78.0	15.00
28.0	4.08	79.0	15.58
29.0	4.19	80.0	16.19
30.0	4.30	81.0	16.83
31.0	4.41	82.0	17.55
32.0	4.52	83.0	18.32
33.0	4.64	84.0	19.17
34.0	4.75	85.0	20.09
35.0	4.87	86.0	21.12
36.0	5.00	87.0	22.27
37.0	5.11	88.0	23.57
38.0	5.23	89.0	25.04
39.0	5.36	90.0	26.74
40.0	5.50	91.0	28.71
41.0	5.62	92.0	31.06
42.0	5.75	93.0	33.92
43.0	5.89	94.0	37.48
44.0	6.00	95.0	42.10
45.0	6.17	96.0	48.46
46.0	6.31	97.0	57.88
47.0	6.46	98.0	74.00
48.0	6.62	99.0	112.19
49.0	6.77	100.0	5351.00
50.0	6.94		

Supplementary Table S10: Distribution of average number of exposures (per active day) for all individuals residing in 382 Metropolitan Statistical Areas (MSAs). The median individual had 6.94 exposures on the average day of activity. 8,609,406 individuals reside in a Metropolitan Statistical Area (90% of the overall 9,567,559 individuals in our study). The remaining 958,153 users live outside of MSAs, influencing the interaction segregation of an MSA by coming into contact with MSA residents. Activity is defined as one or more interactions occurring on a given day. For details on activity over time, see Supplementary Figure S29.

Percentile	# of Distinct Exposures	Percentile	# of Distinct Exposures
0.0	1.00	51.0	4.55
1.0	1.00	52.0	4.65
2.0	1.00	53.0	4.76
3.0	1.00	54.0	4.87
4.0	1.14	55.0	5.00
5.0	1.23	56.0	5.09
6.0	1.30	57.0	5.21
7.0	1.36	58.0	5.33
8.0	1.43	59.0	5.46
9.0	1.50	60.0	5.60
10.0	1.56	61.0	5.73
11.0	1.62	62.0	5.87
12.0	1.67	63.0	6.00
13.0	1.75	64.0	6.17
14.0	1.80	65.0	6.32
15.0	1.87	66.0	6.49
16.0	1.94	67.0	6.66
17.0	2.00	68.0	6.83
18.0	2.00	69.0	7.00
19.0	2.10	70.0	7.21
20.0	2.17	71.0	7.41
21.0	2.23	72.0	7.62
22.0	2.29	73.0	7.86
23.0	2.36	74.0	8.09
24.0	2.42	75.0	8.33
25.0	2.50	76.0	8.61
26.0	2.55	77.0	8.89
27.0	2.62	78.0	9.20
28.0	2.68	79.0	9.52
29.0	2.75	80.0	9.87
30.0	2.82	81.0	10.24
31.0	2.89	82.0	10.65
32.0	3.00	83.0	11.08
33.0	3.00	84.0	11.57
34.0	3.10	85.0	12.09
35.0	3.17	86.0	12.67
36.0	3.25	87.0	13.33
37.0	3.33	88.0	14.05
38.0	3.40	89.0	14.88
39.0	3.48	90.0	15.83
40.0	3.56	91.0	16.94
41.0	3.64	92.0	18.25
42.0	3.72	93.0	19.83
43.0	3.81	94.0	21.78
44.0	3.89	95.0	24.31
45.0	4.00	96.0	27.71
46.0	4.07	97.0	32.71
47.0	4.16	98.0	41.00
48.0	4.25	99.0	59.00
49.0	4.35	100.0	1740.25
50.0	4.45		

Supplementary Table S19: Distribution of average number of distinct exposures (per active day) for all individuals residing in 382 Metropolitan Statistical Areas (MSAs). The median individual had 4.45 unique exposures on the average day of activity. 8,609,406 individuals reside in a Metropolitan Statistical Area (90% of the overall 9,567,559 individuals in our study). The remaining 958,153 users live outside of MSAs, influencing the interaction segregation of an MSA by coming into contact with MSA residents. Activity is defined as one or more interactions occurring on a given day. For details on activity over time, see Supplementary Figure S29.

Active Individuals Over Time

of Active Individuals

Supplementary Figure S26: Active individuals over time. Number of active individuals (i.e. nodes in the network) over the study observation period. Activity is defined as one or more exposures occurring on a given day.

Daily Exposures Per Person Over Time

Supplementary Figure S27: Average number of exposures over time. Mean/median exposures per active individuals (i.e. nodes in the network) over the study observation period. Activity is defined as one or more exposures occurring on a given day.

Daily Distinct Exposures Per Person Over Time

Supplementary Figure S35: Average number of distinct exposures over time. Mean/median distinct interactions per active individuals (i.e. nodes in the network) over the study observation period. Activity is defined as one or more exposures occurring on a given day.

Percentile	Tie Strength	Percentile	Tie Strength
0.0	1	51.0	1
1.0	1	52.0	1
2.0	1	53.0	1
3.0	1	54.0	1
4.0	1	55.0	1
5.0	1	56.0	1
6.0	1	57.0	1
7.0	1	58.0	1
8.0	1	59.0	1
9.0	1	60.0	1
10.0	1	61.0	1
11.0	1	62.0	1
12.0	1	63.0	1
13.0	1	64.0	1
14.0	1	65.0	1
15.0	1	66.0	1
16.0	1	67.0	1
17.0	1	68.0	1
18.0	1	69.0	1
19.0	1	70.0	1
20.0	1	71.0	1
21.0	1	72.0	1
22.0	1	73.0	1
23.0	1	74.0	1
24.0	1	75.0	2
25.0	1	76.0	2
26.0	1	77.0	2
27.0	1	78.0	2
28.0	1	79.0	2
29.0	1	80.0	2
30.0	1	81.0	2
31.0	1	82.0	2
32.0	1	83.0	2
33.0	1	84.0	2
34.0	1	85.0	2
35.0	1	86.0	2
36.0	1	87.0	2
37.0	1	88.0	2
38.0	1	89.0	2
39.0	1	90.0	2
40.0	1	91.0	3
41.0	1	92.0	3
42.0	1	93.0	3
43.0	1	94.0	3
44.0	1	95.0	4
45.0	1	96.0	4
46.0	1	97.0	5
47.0	1	98.0	7
48.0	1	99.0	11
49.0	1	100.0	11644
50.0	1		

Supplementary Table S11: Distribution of tie strength (# of exposures) for all pairs of individuals residing in 382 Metropolitan Statistical Areas (MSAs).

The network / temporal sampling. Neither the MS nor SI do not say anything about the temporal properties of the GPS pings. We know that individuals have at least 500 pings, but I would like to see

- the distribution of pings per person,
- statistics about how the pings are distributed in time, what does, for example, distribution of the number of days of data coverage per person,
- what is the distribution of interevent times, etc. Is the data evenly collected at some fixed rate - or does datapoints clump together in bursts of collected data?
- a plot that shows the number of individuals with data per day as a function of calendar-time
- a plot of the number of estimated interactions day as a function of calendar-time (also for the various robustness checks)

R1's comment highlights the importance of providing additional details on the temporal properties of the GPS pings. Some of these details (i.e., distribution of pings per person, distribution of number of days of data coverage per person) were previously available in **Supplementary Table S1** (copied below). Nevertheless, we again agree that additional details on these distributions, as well as further description of other properties of the pings (e.g. interevent times, number of exposures as a function of calendar time), will clarify the results.

	Accurate pings	Unique days	Distinct user pairs	Exposures	Accurate pings	Distinct user pairs	Exposures
count	8,609,406				382		
mean	3,273	35	184	363	73,757,695	2,577,322	4,845,144
std	16,507	20	374	1,073	163,848,305	8,872,464	16,838,938
min	11	2	1	1	2,196,084	27,326	53,350
10%	570	13	8	17	8,398,875	140,251	313,803
50%	1,471	30	76	141	22,054,930	504,525	1,031,691
90%	5,857	63	436	785	175,295,175	4,573,152	8,954,800
max	4,755,081	95	42,323	193,193	1,605,070,032	94,140,015	215,183,409

Supplementary Table S1: Combined descriptive statistics for all individuals residing in 382 Metropolitan Statistical Areas (MSAs). 8,609,406 individuals reside in a Metropolitan Statistical Area (90% of the overall 9,567,559 individuals in our study). The remaining 958,153 users live outside of MSAs, influencing the **exposure** segregation of an MSA by coming into contact with MSA residents. Descriptive statistics are grouped by individual (left) and MSA (right). At least one of two users in each **exposure** pair must live in an MSA to be included in this table.

In the following pages, we highlight the new tables and figures which provide all of this requested information.

distribution of pings per person	Supplementary Table S12
distribution of the number of days of data coverage per person	Supplementary Table S13

what is the distribution of interevent times, etc. Is the data evenly collected at some fixed rate - or does datapoints clump together in bursts of collected data?	Supplementary Table S14
a plot that shows the number of individuals with data per day as a function of calendar-time	Supplementary Figure S27
a plot of the number of estimated interactions a day as a function of calendar-time (also for the various robustness checks)	Supplementary Figure S28 (robustness checks addressed in later response)
a plot of the number of distinct estimated interactions a day as a function of calendar-time (also for the various robustness checks)	Supplementary Figure S36 (robustness checks addressed in later response)

Regarding whether data are evenly-spaced or “bursty” (i.e., clumped together), this of course varies across SafeGraph users. In addition to the quantitative analysis above, we show two illustrative examples (one with evenly-spaced and the other with bursty data)¹. Each row is a day, each blue dot is a ping. We visually inspected a random sample of 100 users, and found that the majority (78%) were “bursty” and a minority (22%) were evenly-spaced. This burstiness among most users is likely because most apps which collect and report data to SafeGraph are used sporadically throughout the day. We appreciate R1’s feedback on the importance of this point, and have added mention of this limitation to the discussion section of the manuscript (p. 11, line 162).

{REDACTED}

Editorial note: as per the footnote, the authors chose to exclude these figures from the primary manuscript out of precaution. Though the dates were obscured, they depicted a random sample of 5 days and 25% of pings were randomly dropped, they felt that the potential to identify users did not justify publishing these figures and so they have been redacted.

¹ For user privacy, dates are obscured, a random sample of 5 days is shown, and 25% of pings are randomly dropped. Nevertheless we exclude these figures from the primary manuscript out of precaution.

Percentile	Accurate Pings	Raw Pings	Percentile	Accurate Pings	Raw Pings
0	11	500	51	1,507	1,668
1	370	513	52	1,544	1,706
2	424	526	53	1,582	1,745
3	458	541	54	1,621	1,786
4	483	555	55	1,661	1,827
5	501	571	56	1,702	1,869
6	514	586	57	1,745	1,913
7	528	602	58	1,789	1,958
8	542	618	59	1,834	2,004
9	556	634	60	1,880	2,052
10	570	651	61	1,927	2,101
11	585	668	62	1,976	2,152
12	600	686	63	2,027	2,204
13	616	703	64	2,080	2,259
14	631	721	65	2,134	2,315
15	647	739	66	2,190	2,374
16	664	757	67	2,249	2,434
17	680	776	68	2,310	2,498
18	697	795	69	2,373	2,565
19	714	814	70	2,440	2,634
20	731	833	71	2,509	2,708
21	749	852	72	2,582	2,785
22	767	872	73	2,659	2,867
23	785	892	74	2,740	2,955
24	803	913	75	2,827	3,048
25	822	933	76	2,919	3,148
26	842	955	77	3,019	3,256
27	861	976	78	3,125	3,372
28	881	998	79	3,241	3,498
29	901	1,021	80	3,367	3,636
30	922	1,043	81	3,504	3,788
31	944	1,067	82	3,656	3,954
32	966	1,090	83	3,824	4,139
33	988	1,114	84	4,011	4,344
34	1,011	1,139	85	4,220	4,576
35	1,034	1,164	86	4,460	4,837
36	1,058	1,190	87	4,733	5,137
37	1,083	1,216	88	5,051	5,479
38	1,108	1,243	89	5,420	5,873
39	1,134	1,271	90	5,857	6,327
40	1,160	1,300	91	6,370	6,862
41	1,187	1,329	92	6,987	7,492
42	1,215	1,360	93	7,735	8,258
43	1,244	1,391	94	8,669	9,211
44	1,274	1,423	95	9,885	10,444
45	1,304	1,456	96	11,543	12,116
46	1,336	1,489	97	14,011	14,602
47	1,369	1,523	98	18,150	18,735
48	1,402	1,558	99	27,407	27,938
49	1,436	1,594	100	4,755,081	4,777,213
50	1,471	1,630			

Supplementary Table S12: Distribution of total pings for all included individuals residing in 382 Metropolitan Statistical Areas (MSAs). The median individual has 1,471 accurate pings. Accurate pings are those with < 100 meters error.

Percentile	Unique Days	Percentile	Unique Days
0	2	51	30
1	5	52	31
2	7	53	31
3	8	54	31
4	9	55	31
5	9	56	31
6	10	57	31
7	11	58	32
8	11	59	32
9	12	60	32
10	13	61	32
11	13	62	32
12	14	63	33
13	15	64	34
14	15	65	34
15	16	66	35
16	16	67	37
17	17	68	38
18	17	69	39
19	18	70	40
20	18	71	41
21	19	72	43
22	19	73	44
23	20	74	45
24	20	75	47
25	21	76	48
26	21	77	49
27	22	78	51
28	22	79	52
29	23	80	53
30	23	81	54
31	24	82	56
32	24	83	57
33	25	84	58
34	25	85	59
35	25	86	60
36	26	87	61
37	26	88	62
38	26	89	62
39	27	90	63
40	27	91	64
41	27	92	67
42	27	93	70
43	28	94	74
44	28	95	78
45	28	96	82
46	29	97	86
47	29	98	90
48	29	99	93
49	30	100	95
50	30		

Supplementary Table S13: Distribution of unique days of ping data coverage for all included individuals residing in 382 Metropolitan Statistical Areas (MSAs). The median individual has 30 days of data coverage.

Percentile	Time Elapsed Between Pings (seconds)	Percentile	Time Elapsed Between Pings (seconds)
0		1	51
1		1	52
2		1	53
3		1	54
4		1	55
5		1	56
6		1	57
7		1	58
8		1	59
9		1	60
10		1	61
11		1	62
12		1	63
13		1	64
14		1	65
15		1	66
16		1	67
17		2	68
18		2	69
19		2	70
20		2	71
21		2	72
22		2	73
23		3	74
24		3	75
25		3	76
26		3	77
27		4	78
28		4	79
29		4	80
30		4	81
31		5	82
32		5	83
33		5	84
34		5	85
35		5	86
36		5	87
37		5	88
38		5	89
39		5	90
40		5	91
41		5	92
42		5	93
43		5	94
44		5	95
45		5	96
46		5	97
47		5	98
48		5	99
49		6	100
50		6	

Supplementary Table S14: Distribution of interevent times (seconds elapsed between pings) for all included individuals residing in 382 Metropolitan Statistical Areas (MSAs). The median interevent time is 6 seconds, and 96% of interevent times are < 60 minutes. Distribution is estimated using a random sample of 1% of users, corresponding to 326,039,078 pings

Individuals With Data Over Time

of Individuals with > 0 Accurate Pings

Supplementary Figure S28: Individuals with data coverage over time. Number of individuals with data (i.e. > 0 accurate pings) over the study observation period. Accurate pings are those with < 100 meters error.

Distinct Exposures Over Time

Total Number of Distinct Exposures

Supplementary Figure S36: Distinct exposures over time across all individuals residing in the 382 MSAs.

The probability of an "interaction" depends on the overall density of individuals in an area (a city). Let me illustrate using an example. Let's say we have data from two groups of 100 individuals sampled from low and high population density areas. To make things concrete, let's say that Group A all live within a 1x1 mile area and Group B live distributed across a 20x20 mile area. My common sense would say that the method presented by the authors would detect many more random encounters (= a denser network) among group A than group B, simply because they occupy a smaller geographical space where stop-locations are more densely packed. This dependence density is concerning, since the authors are making claims about how the mixing of social relationships appear to depend on density (metropolitan areas). I would like the authors to comment on this issue and what it might mean for their results.

There are several important issues raised here.

First, as clarified in our prior responses (response pp. 1-3, 14-19), the cosmopolitan mixing hypothesis primarily concerns happenstance (i.e., unplanned or "random") exposures, which our metric captures by design. Thus, including such exposures is not a problem that needs remedying, but an important part of our study design.

Second, R1's reasoning above would actually lead to an underestimate (not overestimate) of segregation in dense areas. So, if R1's reasoning held true, dense cities would be *less* segregated (due to more "random" exposures), whereas we find exactly the opposite: dense cities are *more* segregated. R1's example does not account for the fact that—both in high and low density cities—most exposures occur in specific POIs and hubs (**Figure 3**) [16]. We find that in large cities, segregation increases, as there are many *more* venues that serve specific socioeconomic groups (e.g., **Extended Data Figure 4**, copied below).

Imagine City A (high density) with two separate hubs for rich and poor residents, and City B (low density) with one shared hub. If individuals spent most of their time at hubs, City A will be more segregated despite high density. This example illustrates why we use interpersonal mobility networks, instead of static city properties (e.g. density, distance), to analyze segregation. Our findings also further emphasize the need for policies that support the development of bridging hubs to integrate diverse communities.

Density at individual locations. A key takeaway from this paper <https://www.pnas.org/doi/10.1073/pnas.1006155107> is that the background density for two individuals is useful information for inferring social interactions. If two people are observed nearby one another at a Giants Stadium during a game, that co-occurrence is less significant than if they are observed together at midnight in a remote mountain cabin.

I worry that this effect might bias the results for encounters and mixing at specific venues ... and would like to hear the authors' comments to this issue.

Because the cosmopolitan hypothesis references exposure, it's important for our primary analysis to include all types of happenstance mixing. We agree, however, with R1 that it is also important to assess whether our results are driven only by exposures in high-density settings (e.g., hubs, stadiums), or if they generalize to exposures in lower density POIs (e.g. parks). In our revision, we design two new robustness checks which confirm that our findings are robust to the background density of venues. Specifically, we (1) downweight exposures by the number of individuals simultaneously exposed to each other, and (2) we use only exposures in which both people *were not exposed to anybody else*.

Our key findings are robust under both metrics (Supplementary Figure S9, copied above), and these metrics are highly correlated to our primary metric (Spearman Corr. 0.97 and Spearman Corr. 0.94; Supplementary Table S6). We have included these robustness checks in our supplementary information, citing the PNAS paper to motivate them. We appreciate R1's feedback on this point.

...a plot of the number of estimated interactions day as a function of calendar-time (also for the various robustness checks)

Null models (in the MS). In the SI the authors perform several robustness checks and show that their results (in a qualitative sense) hold in spite of many different (including stricter) definitions of a "social interaction". But because I am interested in the *network* and not just (quantities derived from the network), **I would like to better understand what happens to the network properties - the metrics listed under (1) above, for the "robustness check networks" shown Fig S2-S8.**

We agree that it is important to assess the robustness of network properties themselves. One challenge in addressing this comment is the sheer number of robustness checks we conduct. There are a total of 38 robustness checks (in our revised manuscript), and 6 metrics highlighted by R1. Rather than presenting all 228 plots, we'll address the spirit of R1's point in the following ways.

- Firstly, as highlighted earlier in this response, we add two columns (% Pairs, % People) to **Supplementary Table S6** (copied below), which show how the robustness checks affect network sparsity.

Feature	Pearson Corr. w/ Primary	Spearman Corr. w/ Primary	Median	Mean	% Pairs	% People
Primary Measure	—	—	0.35	0.35	100.00	100.00
Primary Measure (+ Up-weight Multiple Exposures)	0.89	0.91	0.46	0.45	100.00	100.00
SES Definition: Rent Zestimate Percentile	0.88	0.89	0.42	0.42	100.00	100.00
SES Definition: Within-MSA Rent Zestimate Percentile	0.81	0.83	0.54	0.53	100.00	100.00
SES Definition: Census Median Household Income	0.75	0.77	0.47	0.46	100.00	100.00
SES Definition: Educational Attainment (% College or Higher)	0.70	0.71	0.52	0.52	100.00	100.00
Exclude Pri/Sec Roads	0.99	0.99	0.37	0.37	74.30	99.87
Exclude Roads	0.98	0.98	0.37	0.37	38.66	98.56
Stationary Individuals (2 pings < 10 meters in 1-10 min)	0.86	0.87	0.44	0.43	5.39	79.46
Exclude Same-home exposures	0.98	0.98	0.34	0.34	99.71	99.78
Work/Leisure (Neither in Home Tract)	0.93	0.93	0.31	0.31	86.26	95.55
Leisure (inside POI)	0.85	0.84	0.28	0.29	16.15	76.32
Minimum Distance Between Pings: < 25 meters	0.97	0.97	0.34	0.34	49.53	97.71
Minimum Distance Between Pings: < 10 meters	0.95	0.94	0.33	0.33	20.93	92.28
Minimum Time Between Pings: < 2 minutes	0.97	0.97	0.34	0.34	53.59	98.92
Minimum Time Between Pings: < 60 seconds	0.97	0.97	0.35	0.35	33.67	97.83
Minimum Tie Strength: 2 consecutive exposures	0.94	0.95	0.35	0.35	18.25	94.80
Minimum Tie Strength: 3 consecutive exposures	0.83	0.83	0.37	0.37	3.06	65.46
Minimum Tie Strength: 2 unique days of exposure	0.88	0.90	0.47	0.46	7.46	83.56
Minimum Tie Strength: 3 unique days of exposure	0.73	0.77	0.56	0.54	2.32	62.81
Dist. < 25 meters, Time < 2 min., >= 2 consec. exposures	0.92	0.93	0.36	0.35	8.80	86.64
Dist. < 25 meters, Time < 2 min., >= 2 unique days	0.87	0.89	0.46	0.44	3.84	70.57
Dist. < 10 meters, Time < 60 sec., >= 3 consec. exposures	0.78	0.79	0.39	0.38	0.94	38.16
Dist. < 10 meters, Time < 60 sec., >= 3 unique days	0.68	0.68	0.52	0.51	0.58	28.92
Downweight Simultaneous Exposures	0.97	0.97	0.34	0.34	99.99	99.99
Exclude Simultaneous Exposures	0.94	0.95	0.46	0.46	22.87	99.61
Tie Strength: 1 Exposure	0.97	0.98	0.31	0.32	74.24	98.66
Tie Strength: 2 Exposures	0.95	0.96	0.33	0.33	16.14	93.20
Tie Strength: 3 Exposures	0.89	0.90	0.36	0.36	4.09	74.67
Tie Strength: 4 Exposures	0.85	0.86	0.37	0.36	1.82	58.51
Tie Strength: 5+ Exposure	0.83	0.85	0.45	0.45	3.42	70.62
Minimum Stationary Nights: 6 nights	0.99	0.99	0.35	0.35	77.45	83.91
Minimum Stationary Nights: 9 nights	0.99	0.99	0.35	0.36	68.64	73.42
Minimum Stationary Nights: 12 nights	0.98	0.98	0.35	0.36	59.16	63.66
Racial Segregation (% Non-White)	0.59	0.60	0.46	0.46	100.0	100.0
Economic Segregation (White Overall)	0.94	0.95	0.34	0.34	55.97	63.83
Economic Segregation (White Within-group)	0.93	0.94	0.34	0.34	39.84	63.69
Economic Segregation (Non-White Overall)	0.55	0.52	0.28	0.28	43.76	35.50
Economic Segregation (Non-White Within-group)	0.47	0.44	0.28	0.30	27.50	35.31

(2) Additionally, for the most critical robustness checks that vary the distance, time, and length thresholds of exposure, we show the key metrics below.

Distribution of number of distinct contacts per person (all time)	Supplementary Table S20
Distribution of number of distinct contacts per person (for a single day)	Supplementary Table S21
Number of active nodes over time	Supplementary Figures S37-S44
Mean/median number of distinct contacts per person over time	Supplementary Figures S45-S52
A plot of the number of estimated distinct interactions day as a function of calendar-time	Supplementary Figures S53-S60

We focus on distinct contacts (i.e., the de-duplicated set of exposures to unique people) because our primary metric does not weight by edge strength (to avoid “overweighting” stronger ties). As such, the number of distinct exposures is the best metric to compare the extent of sparsity across different robustness checks.

Below, we copy the distribution tables listed above as well as an illustrative example of the daily metrics for a single robustness check (distance threshold of 25 meters). We refer readers to the supplementary figures of the manuscript for the complete set of robustness check figures. We again thank R1 for this feedback.

Measure	Mean	std	min	10%	25%	50%	75%	90%	max
Primary Measure	184.21	373.98	1.0	8.0	25.0	76.0	196.0	436.0	42323.0
Minimum Distance Between Pings: < 25m	93.23	200.13	1.0	4.0	12.0	37.0	97.0	219.0	26778.0
Minimum Distance Between Pings: < 10m	41.47	99.99	1.0	2.0	5.0	14.0	38.0	94.0	12907.0
Minimum Time Between Pings: < 2 minutes	99.92	212.03	1.0	5.0	14.0	41.0	104.0	232.0	26592.0
Minimum Time Between Pings: < 60 seconds	63.50	141.10	1.0	4.0	9.0	26.0	65.0	144.0	18912.0
Minimum Tie Strength: 2 consecutive exposures	35.51	91.98	1.0	2.0	5.0	13.0	33.0	78.0	13513.0
Minimum Tie Strength: 3 consecutive exposures	8.53	23.31	1.0	1.0	1.0	3.0	7.0	18.0	3107.0
Minimum Tie Strength: 2 unique days	16.39	54.78	1.0	1.0	2.0	5.0	13.0	33.0	15454.0
Minimum Tie Strength: 3 unique days	6.74	22.04	1.0	1.0	1.0	3.0	6.0	13.0	7685.0
Dist. < 25m, Time < 2 minutes, Length 2+ unique days	9.83	33.09	1.0	1.0	2.0	3.0	8.0	19.0	9951.0
Dist. < 25m, Time < 2 minutes, Length 2+ consecutive	18.62	49.25	1.0	1.0	3.0	7.0	17.0	41.0	8447.0
Dist. < 10m, Time < 1 minutes, Length 3+ unique days	3.30	10.72	1.0	1.0	1.0	1.0	3.0	6.0	2019.0
Dist. < 10m, Time < 1 minutes, Length 3+ consecutive	4.31	12.40	1.0	1.0	1.0	2.0	4.0	9.0	2893.0

Supplementary Table S20: Distribution of number of distinct exposures per person (by time, distance, and length threshold). We compute the number of distinct pairs of exposures for all residents of the 382 Metropolitan Statistical Areas (MSAs), for each robustness check which varies the time, distance, length thresholds for the definition of exposure. Summary of the per person number of distinct exposures.

Measure	Mean	std	min	10%	25%	50%	75%	90%	max
Primary Metric	7.90	13.90	1.0	1.56	2.50	4.45	8.33	15.83	1740.25
Minimum Distance Between Pings: < 25 meters	3.25	5.39	1.0	1.00	1.25	1.90	3.20	6.06	907.75
Minimum Distance Between Pings: < 10 meters	2.54	4.58	1.0	1.00	1.00	1.40	2.17	4.33	510.14
Minimum Time Between Pings: < 2 minutes	3.17	5.25	1.0	1.00	1.29	1.95	3.23	5.86	973.75
Minimum Time Between Pings: < 60 seconds	2.49	3.79	1.0	1.00	1.13	1.62	2.50	4.33	674.50
Minimum Tie Strength: 2 consecutive interactions	2.19	3.59	1.0	1.00	1.00	1.40	2.11	3.67	622.00
Minimum Tie Strength: 3 consecutive interactions	1.61	2.18	1.0	1.00	1.00	1.00	1.50	2.45	344.00
Dist. < 25 meters, Time < 2 minutes, Length 2+ consecutive interactions	1.76	2.30	1.0	1.00	1.00	1.18	1.75	2.89	1148.00
Dist. < 10 meters, Time < 60 minutes, Length 3+ consecutive interactions	1.43	2.03	1.0	1.00	1.00	1.00	1.30	2.00	342.00

Supplementary Table S21: Distribution of average number of distinct exposures (per active day) per person across all days of activity (by time, distance, and length threshold). Summary statistics are shown for for all residents of the 382 Metropolitan Statistical Areas (MSAs), for each robustness check which varies the time, distance, length thresholds for the definition of exposure. Activity is defined as one or more exposure occurring on a given day. Summary of the per person number of distinct exposures.

Active Individuals Over Time
 (Minimum Distance Between Pings: < 25 meters)

(a)

Supplementary Figure S37: Active individuals over time (exposure distance threshold: 25 meters). Number of active individuals (i.e. nodes in the network) over the study observation period. Activity is defined as one or more exposures occurring on a given day.

**Daily Distinct Exposures Per Person Over Time
(Minimum Distance Between Pings: < 25 meters)**

Supplementary Figure S45: Average number of distinct exposures over time (distance threshold: 25 meters). Mean/median distinct interactions per active individuals (i.e. nodes in the network) over the study observation period. Activity is defined as one or more exposures occurring on a given day.

Distinct Exposures over Time
 (Minimum Distance Between Pings: < 25 meters)

Total Number of Distinct Exposures

Supplementary Figure S53: Number of distinct exposures (distance threshold: 25 meters) over time across all individuals residing in the 382 MSAs.

Null models (that would help clarify what's going on). Even though there are already several null models, I would like to suggest some extra ones.

* To estimate the fraction of random encounters, the authors could randomly displace all trajectories in time (depending on the observation period, it could be ± 10 days). So my trajectory today would be compared to others displaced in time. That would provide an estimate of which fraction of inferred interactions are driven by density, etc.

* The effect of a null model that take into account density of nearby individual in time & space, as discussed here <https://www.pnas.org/doi/10.1073/pnas.1006155107> would also be very interesting to see. Especially for understanding segregation at particular venues.

We thank R1 for these suggestions of additional robustness checks.

Regarding the first robustness check, as we have stressed throughout our responses (pp. 16-19, 74-76), the vast majority of exposures in a city are indeed happenstance (i.e. “random” and unplanned). Capturing these happenstance exposures is necessary to assess the cosmopolitan mixing hypothesis (and not a limitation of our metric).

Regarding the second robustness check, we agree that it is important to understand the influence of background density on our metric. As mentioned above (in response to R1’s concern about background density on p. 38), we have added two new robustness checks to address this point (*Supplementary Figure S9*, copied below).

The definition of economic segregation as the average of everyone with whom they interact might be problematic. The distribution of income is heavy-tailed, so an average might be driven by large values from the tail. This could be remedied by using rank (or equivalent position) in the income distribution rather than absolute dollar amounts. Even if the authors prefer to use income as a measure, I would like to see results for a rank-based measure.

This concern, which we agree is an important one, is already addressed by our robustness check using rent percentiles as the measure of socioeconomic status (**Supplementary Figure S3**, copied below). Our results are robust under this definition of segregation and it correlates highly (Spearman Corr. 0.89; **Supplementary Table S6**, copied below) to our primary measure of segregation.

The MS does not make it clear (at least I couldn't find it) how repeated interactions are weighted in the dataset. For example, let's say my partner and I are both in the dataset and see each other every day. Does each observation of my partner count when the average ES of my interactions is calculated? Or does my partner only count once? I would like the authors to comment on how this choice affects the results.

We thank R1 for raising this clarifying question. Our primary metric does not weight by edge strength (to avoid “overweighting” stronger ties). We have revised our manuscript to clarify this point (p. 3, line 44). Our primary findings are unaffected by this choice, as shown in our existing robustness check (**Supplementary Figure S2**) in which we upweight repeated exposures.

The MS states that the dataset is "anonymous". But in what sense is the data anonymous if we can identify the home-location of a large fraction individuals in the data ... along with everywhere they go? As far as I know these bits of information would make it possible to re-identify a large fraction of users, e.g. through a side-channel attack.

We initially used the phrase "anonymous" to convey that the GPS ping data does not explicitly contain an individual's name, address, or date of birth. We appreciate R1's concern that this is arguably a nominal definition and that there's a threat of a side-channel attack. We have replaced the phrase "anonymous" with "de-identified" in the manuscript (e.g. p. 15, line 185), which more accurately describes the data [23].

More generally, we agree that our data are sensitive. For this reason, all analysis was conducted on a secure server behind a firewall, in accordance with the guidelines of the Stanford University Institutional Review Board. Furthermore, individual-level data will of course not be released, as indicated in our reporting summary.

The authors find that POIs such as religious organizations tend to be more segregated. Does this not contradict Chetty et al.'s finding that people tend to befriend more well-off people at a higher rate than expected at random in religious groups? It would be good if the authors could comment on this explicitly in the text (even if there is no contradiction).

R1 draws an interesting connection between our exploratory analysis of POI-level segregation and the recent study of social capital by Chetty et al. [24]. Our findings are consistent with Chetty et al., as exposure segregation is an orthogonal concept to "friendship bias", which we elaborate on below.

Chetty et al. define "friending bias" as friendship formation *conditioned on affiliation with the same religious organization* (operationalized as liking the organization's page on Facebook). As R1 highlights, friending bias is shown to be lower in religious organizations. So, for instance, a rich and poor person are more likely to become friends *if they are affiliated with the same Church*, compared to if they are affiliated with the same University.

By contrast, we investigate overall cross-class exposure within each POI category. So for instance, the high exposure segregation in religious organizations indicates that rich and poor people in a given city typically visit different religious venues. Chetty et al. actually re-affirm this point when they write that: "high-SES people tend to attend different religious institutions and colleges compared with low-SES people". The key difference, of course, is that we directly measure which POIs individuals attend, whereas Chetty et al. use social media likes of an organization's Facebook page as (an arguably very weak) proxy for attendance.

Similar results to ours can be seen in the Moro et al. paper cited by R1, in which religious organizations are shown to have above-average income segregation (by visitation) [25]. We agree that we should explicitly comment on this in the text, and have now cited both Moro et al. and

Chetty et al. to indicate that we are corroborating their findings (p. 7, lines 89-90). We have also revised our manuscript to distinguish, as noted earlier, between these efforts to explore our measure and our novel contribution (i.e., the first systematic assessment of the cosmopolitan mixing hypothesis).

Page 6 line 84: "Large cities facilitate segregation". The authors provide no proof of causal effects. A more accurate claim would be "Interaction segregation correlates with city size and population density". This causal claim occurs several times in the paper and should be corrected throughout.

We thank R1 for highlighting this concern. We have corrected our phrasing throughout the manuscript to clarify that our findings rely on descriptive analysis and are not causal.

When inferring home location, the authors use interpolation to fill missing data. This seems reasonable (at least for inferring the home location), but it should probably only be up to a maximum number of missing hours between two recorded data points (it would be nice to have a few more details on what's going on here).

This is another good point. We interpolate for each hour separately, and as such there is a maximum allowed interval of 60 minutes between two pings for interpolation to occur. We have added additional commentary to clarify this (p. 19, lines 289-290) and the exact method of computation for the interpolation (p. 19, lines 291-292).

Also, the authors filter for users that have stationary nighttime observations on at least 3 nights and with at least 60% of observations within 50 meters radius. 3 nights out of how many nights? All the nights from the 3 months of data? This seems like a very small threshold to use for inferring home location.

Although our overall observation window is 3 months, each individual is typically observed for a subset of these days (as clarified in Supplementary Table S13). We use a threshold of 3 nights overall to make it possible to include users with few nights of data (thus reducing possible selection bias), while yielding high confidence (i.e., 89% of stationary nighttime observations are within 50 meters of the inferred home latitude and longitude). However, our results are robust to choice of exact threshold. We have now added a number of robustness checks in which we increase the threshold to 6, 9, and 12 days (Supplementary Figure S61, copied below)—and we find that these measures correlate highly (Spearman Corr. 0.98-99) with our primary measure, and that our primary findings remain unchanged.

Minimum Stationary Nights: 6 nights

Minimum Stationary Nights: 9 nights

Minimum Stationary Nights: 12 nights

There is a section called: “Mechanisms producing higher interaction segregation in larger metropolitan areas”, where the authors show that there is an association between the density of amenities and the level of segregation. I see what the authors are hinting at and there are nice examples, but perhaps "mechanism" is a bit strong. The authors do not provide any empirical evidence that higher density of amenities make cities more segregated (it could be the opposite, or it could be that there is an underlying third phenomenon that drives both). Nor they develop a mechanistic model that offers an explicit possible explanation for the observed association. As a final small point on this, here it is even unclear if here, the authors are controlling for the economic diversity in a given city.

We of course agree that we shouldn't presume a causal effect. We have modified the language (e.g., p. 7, line 100) throughout the text to ensure that our phrasing does not convey causality.

Regarding controlling for economic diversity: while in our primary figures we show direct, pairwise correlations between two variables (for illustrative purposes), we do control for economic diversity (operationalized as Gini Index) and a variety of other covariates in our extended analyses (referenced on p. 5, line 62). See *Extended Data Tables 1-2, Supplementary Table S7*. We agree with R1 that our analysis of amenity densities would also benefit from such an extended analysis. To this end, we have added another analysis (Supplementary Table S22, copied below) in which we specifically show that higher density of amenities is significantly associated with exposure segregation, even after controlling for economic diversity and other important covariates.

	Dependent variable:			Exposure Segregation		
	(1)	(2)	(3)	(4)	(5)	(6)
Intercept	0.355*** (0.004)	0.355*** (0.004)	0.355*** (0.004)	0.356*** (0.003)	0.355*** (0.003)	0.355*** (0.003)
log(POI Density)	0.055*** (0.004)		0.035*** (0.004)	0.039*** (0.005)	0.020*** (0.004)	0.022*** (0.004)
Gini Index (Estimated Rent)		0.064*** (0.004)	0.051*** (0.004)	0.052*** (0.004)	0.046*** (0.003)	0.047*** (0.003)
Political Alignment (% Democrat in 2016 Election)				0.004 (0.005)		0.003 (0.004)
Racial Demographics (% non-Hispanic White)				-0.003 (0.004)		0.002 (0.003)
Mean SES (Estimated Rent)				-0.015*** (0.004)		-0.006 (0.004)
Walkability (Walkscore)					0.003 (0.003)	0.003 (0.004)
Commutability (% Commute to Work)					-0.010*** (0.004)	-0.010** (0.004)
Conventional Segregation (NSI)					0.045*** (0.003)	0.043*** (0.003)
Observations	382	382	382	376	382	376
R^2	0.307	0.419	0.526	0.540	0.690	0.688
Adjusted R^2	0.305	0.417	0.524	0.534	0.685	0.681

*p<0.1; **p<0.05; ***p<0.01

I didn't read this paper carefully, and I see that it is cited, but based on my superficial knowledge, this paper does look at mixing at individual locations in a way that is very close to what happens here <https://www.nature.com/articles/s41467-021-24899-8>

We further elaborate on the differences between our work and that of Moro et al. in our response to R3 (response p. 71). In short, the key methodological differences are that we use a different unit of analysis (an entire geographical area vs. an individual POI category), and we're working on a different scale (382 MSAs and 2829 counties vs. eleven cities). By virtue of these methodological choices, we can then take on the cosmopolitan hypothesis and further discover how hubs bridge economically diverse neighborhoods.

We do agree that our analysis of POIs in **Figure 2c** is similar to that of Moro et al. This exploratory analysis is intended to build intuition for our primary results in **Figure 2d-f** (by showing that segregation is increased when POIs become more economically differentiated). We have now explicitly clarified that the aim of this analysis is exploratory (p. 5, line 72), and we now cite Moro et al. in the discussion to indicate that they corroborate our results (p. 5, lines 89-90). We have also condensed this section to make it clear that our primary findings are what follows.

Fig 1a is difficult to understand (at least in the eyes of this reviewer) and could be omitted.

Fig 1b: It is very difficult to see the difference between the red and orange color in this plot.

Caption Fig 2: "Analysis limited to counties with at least 50 individuals". I assume that the authors refer not to the population of counties, but to the number of users present in the dataset for that county. This should be stated more clearly.

Fig 3cd What is the scale here?... Up to 100m random noise added to preserve anonymity. Is this sufficient -- this is difficult to know without an indication of scale?

We agree on all of the points above. We have made the following changes:

- We have omitted Fig 1a.
- We have adjusted our color scheme in Fig 1b (now Fig1a) accordingly, to increase the contrast between colors.
- We have clarified the phrasing in Fig 2.
- We have added scale to Fig 3cd.

Page 10: The sample is representative on many key parameters. The reference says well-sampled to SafeGraph uses the word well-sampled and "representative" seems to be a stretch. The authors are use a more careful choice of words in Section M1, Methods.

We thank R1 for highlighting this concern, and have adjusted our phrasing in Sections **M1, Methods**. We have removed the term “representative” and more carefully described the properties of the dataset (i.e., now noting that SafeGraph data is well-balanced across geography and many key demographic attributes (p. 15, line 180)).

Responses to Referee 2

This was an impressive paper using mobile data to create an improved metric of contact across economic boundaries. Theoretically, most folks were I think aware of limitations of using residential area to estimate contact, and this paper crosses an important technical hurdle to track folks as they go about their days, inferring contact from proximity.

Overall I thought the writing clear, analysis appropriate, and the supporting documents extensive and well organized, I left with thinking I had a clear idea of the idea of the pipeline and what the authors did.

Because of this, my comments are fairly minimal and more invitations for the authors to speculate about the generality/limitations of their results.

We thank R2 for this very positive feedback! We respond to R2's specific comments below.

The main thing I found myself wondering about was the low population areas excluded for the various methodological reasons, and what might be different about the conclusions should these areas be included. The authors find a negative relationship between city size and contact, but I wonder if the evidence for cosmopolitan mixing would come from comparing rural areas (excluded from this analysis) to small/mid-sized cities. This would suggest that there is an overall curvilinear effect of contact on city size when the full range is considered. I think worth it for the authors to mention something about whether they believe this is a possibility (or not) in the general discussion, as it indicates whether the cosmopolitan mixing hypothesis is just totally refuted or it's more a restriction of range problem.

R2 raises an important point: the primary plot of our analysis (**Figure 1c**) focuses on MSAs (metropolitan statistical areas) and does not include rural areas. R2 asks whether our results generalize to low-population, rural areas. Our county-level analysis (**Figure 1c**, and **Extended Data Figure 3**, copied below) speaks to these concerns because it includes 90% of USA counties (including rural counties, and counties containing small towns/small cities).

We take R2’s comment as feedback that our paper would be clearer if we explicitly disaggregated counties by extent of urbanization. In our revision, we added a new figure (**Supplementary Figure S30**, copied below), using the RUCA (rural-urban continuum) designations from the USDA. Here, we also find that our results generalize to the full range of counties, from metropolitan to rural, thus fully refuting the cosmopolitan mixing hypothesis.

There are limitations to using residential area to infer interactions, and this undoubtedly seems a strong improvement. Yet there are likely limitations to using spatial proximity to inferring interactions. The authors mention that limitations exist, but don't really say what they are. The contact literature has shown that high quality interactions (romantic partner, longer conversations) have the greatest impact on attitudes toward folks in other groups. I'm thinking of scenarios in which folks might share proximity but lacking meaningful interactions that actually shape attitudes. For example, there is work showing that higher economic status folks look less at homeless people as they pass them on the sidewalk (P Dietze, ED Knowles - Psychological Science, 2016). Or ordering coffee from someone, and the interaction constitutes a service role. A few lines about what the limitations actually are would be useful.

We thank R2 for highlighting that our discussion of the limitations of our method could be improved. We have fleshed out the limitations section of our discussion accordingly, and cited Dietze and Knowles (p. 11, lines 155 and 162)—which is an excellent example of how other factors beyond spatiotemporal proximity are important in determining the strength of an exposure.

Minor, but a few times in the statistical section, like on line 430 on page 23 the relationship between an independent variable and dependent var in a mixed effects model is referred to as a pearson correlation. And later on line 445. It's not a pearson correlation if it's in a multilevel model, especially when covariates are added to the models later (shared variance partialled out). This can be fixed by just calling this the relationship or association between variables, rather than the pearson's correlation. Maybe this is a jargon difference between disciplines, but my understanding is Pearson's correlation is just linear relationship between single IV and DV.

There is indeed a terminological problem here. We interpret R2 as suggesting that we need to draw a clearer distinction between (1) our estimand (i.e., what we are trying to estimate), and (2) our estimation method. We have now significantly revised **Methods M3** to delineate between the two, and clarify the points below (pp. 25-27, lines 395-396, 434-445, 452-455).

In our revision, we clarify that our estimand itself is a standard Pearson correlation (between a single IV and DV), but that in practice (for our estimator) we compute a *corrected* Pearson correlation using a mixed model, in order to obtain unbiased estimates of exposure segregation even when data for a city or county is sparse:

(1) Our estimand itself is a standard Pearson correlation:

$$\text{Exposure Segregation} = \rho_{X,Y} = \frac{\text{cov}(X,Y)}{\sigma_X \sigma_Y}$$

Where:

- X is an individual's SES
- Y is the true mean SES of those they are exposed to.

As such, there is a single independent variable (IV) and a single dependent variable (DV). Thus, the terminology "Pearson correlation" is accurate specifically when referring to our estimand.

(2) Our estimation procedure uses a mixed model to obtain an unbiased estimate of the above

expression, in contrast to the standard Pearson correlation estimator $r = \frac{\sum (x_i - \bar{x})(y_i - \bar{y})}{\sqrt{\sum (x_i - \bar{x})^2 \sum (y_i - \bar{y})^2}}$ which is downwardly biased.

The standard approach would be, for each person, to use the observed sample mean SES of those they have been exposed to. However, doing so would result in noisy estimates of the DV (in areas where data are sparse), in turn downwardly biasing our estimates of the Pearson correlation.

Instead, we explicitly model the *true* mean SES of the exposure set for each person as $ax_i + b + \epsilon_i^{(1)}$. Individual exposures $y_{i,j}$ are then modeled as noisy draws from a distribution centered at this true mean.

$$y_{ij} = ax_i + b + \epsilon_i^{(1)} + \epsilon_{ij}^{(2)}$$

where x_i = ES of person i

y_{ij} = ES of person j who has interacted with person i

a, b = model parameters

$\epsilon_i^{(1)}$ = person-specific noise term

$\epsilon_{ij}^{(2)}$ = noise for each data point

We then compute the Pearson correlation between each person's SES and the modeled true mean (rather than the estimated noisy mean). We show that this corrects for the downward bias that sparsity induces (**Methods Figure 1**, p. 26).

$$\begin{aligned} \text{corr}(x_i, ax_i + b + \epsilon_i^{(1)}) &= \text{corr}(x_i, ax_i + \epsilon_i^{(1)}) \\ &= \frac{\text{cov}(x_i, ax_i + \epsilon_i^{(1)})}{\sqrt{\text{Var}(x_i) \text{Var}(ax_i + \epsilon_i^{(1)})}} \\ &= \frac{\text{cov}(x_i, ax_i)}{\sqrt{\text{Var}(ax_i + \epsilon_i^{(1)})}} \\ &= \frac{a}{\sqrt{a^2 + \text{Var}(\epsilon_i^{(1)})}} \end{aligned}$$

In summary, the quantity we estimate is a standard Pearson correlation between a single IV and DV. However, our single DV is modeled (to achieve an unbiased estimate), rather than computed directly from the mean of the observed data for each person.

As such, we agree with R2 that our terminology here could be improved, specifically to delineate between our estimand (which is a standard Pearson correlation) and estimation method (which is novel). In our revision, we now refer to this estimation method as a *corrected* Pearson correlation for clarity. We thank R2 for pointing out this concern and hope that our revisions clarify our methods for readers across disciplines.

Responses to Referee 3

Overview of comments: The paper is well-executed and establishes its findings convincingly. My main concern is that there already exists a large body of work looking at similar outcomes with similar data and the additional insight from the paper at hand is not quite novel enough for a top general interest journal. As I outline below, other work has used GPS data to measure various forms of segregation (mostly focusing on racial) and finds similar results. In addition, I would have liked to see more discussion on what types of interactions are measured and to what extent they are relevant for various outcomes.

We thank R3 for the positive feedback on the execution of our study. We concur that there are works that study segregation using similar data. However, our key findings (**Figure 1**, **Figure 2d-f**, and **Figure 3**) and methods are fundamentally novel, and are of general public interest because we are the first to refute a widely-accepted theory about urban life. We have revised our manuscript to (1) highlight our advances over prior work (2) showcase new analyses uniquely enabled by our methods and (3) de-emphasize our secondary exploratory analyses (**Figure 2a-c**) which were intended to provide an overview of exposure segregation and set the stage for our primary discoveries.

(1) Why our study advances (and overturns) prior work

Our key contribution is to rigorously test the cosmopolitan mixing hypothesis, explain why it falls short, and highlight a potential solution to counteract the extreme segregation in large cities. We find overwhelming evidence to reject this long-standing hypothesis, which had never been rigorously tested until now. Consequently, our findings also challenge several of Athey et al.'s conclusions (see response pp. 2-3 for a summary, and pp. 68-70 for detailed comparison to Athey et al.), which is one among many studies to take the cosmopolitan mixing hypothesis at face value. We have revised our manuscript to underscore the novelty of this discovery (p. 5, lines 69–71 and **Extended Data Figure 9**).

Rigorously refuting the cosmopolitan hypothesis requires several methodological advances over prior work, including (a) building a time-sensitive, high-resolution *person-to-person* exposure network (as opposed to static *person-to-place* networks from Athey et al. and other prior work), (b) inferring individual-level SES, and (c) developing an unbiased estimator of exposure segregation for data-sparse areas. We elaborate on these points in our next response below, and have emphasized these methodological differences from prior work in our revision (p. 1 lines 29-32, p. 3 lines 45-46, and p. 5, lines 69-71).

(2) New analyses in this revision, which showcase our novel methods

We thank R3 for their suggestions, which have inspired a number of new analyses. These analyses directly showcase the novelty of our person-to-person exposure network by (a) breaking down the different exposure tie strengths (response pp. 74-76) and (b) linking them to types of POIs and downstream outcomes (response pp. 82-86). These analyses require person-to-person exposure networks and would not be possible using the person-to-place networks featured in prior work. We also (c) investigate racial exposure segregation and its relationship to socioeconomic segregation (response pp. 87-90). Our racial exposure segregation measure also allows for direct comparison to Athey et al. This comparison shows that our time-sensitive and high-resolution measure of exposure is necessary to reject the cosmopolitan mixing hypothesis, because conducting a similar analysis with Athey et al.'s time-indifferent and low-resolution measure yields null results (response pp. 68-70).

(3) Explicitly delineating secondary exploratory analyses from new results

We also revise our manuscript to better delineate our exploratory work (**Figure 2a-c**) from our primary discoveries (**Figure 1, Figure 2d-f and Figure 3**). Our exploratory analyses were intended to provide an overview of exposure segregation and build intuition for our metric. We thank R3 for highlighting that it was not clear that these were exploratory analyses, which we have now clarified in our revision.

We have minimized the discussion of these exploratory analyses, by condensing the two sections “*Interaction segregation is lower than previously estimated*” and “*Interaction segregation varies across leisure sites*” into one section: *Exploring exposure segregation*. We now make it explicit at the start of this section that our aim is to explore our metric. We now cite Moro et al. and Athey et al. frequently (*pp. 5-7, lines 83-90, 92*) to indicate which of their findings we are corroborating.

Overall, we greatly thank R3 for their feedback, which has helped us revise the manuscript to more clearly differentiate it from prior work, and take advantage of the new opportunities for analysis presented by our methods. We elaborate on points (1) and (2) in our subsequent responses below.

Relation to other work using GPS phone data

The paper is part of a growing body of work using GPS data to measure integration and the extent of interaction between different types of individuals... In short, the paper's approach and findings are in line with other recent work in this space and reflects those cutting edge data work and methods.

We conduct the first rigorous test of the cosmopolitan mixing hypothesis—which requires several core methodological contributions. We first highlight these methodological differences (response pp. 60-65), and then showcase several new analyses in our revised manuscript that further reveal the uniqueness of our methods (response pp. 66-67).

Methodological contributions

(1) Prior work builds static *person-to-place* networks that cause errors in measuring exposure (see illustrations on next three pages):

(a) Time-indifferent, assuming exposure between *all past visitors* of a place regardless of visit timing. For instance, Athey et al. would conclude that everyone visiting a park in January at midnight was exposed to everyone visiting the park in an April afternoon. This measurement approach is clearly flawed because visit timing is itself segregated.

(b) Unaware of exposure length or repetition, thus treating brief, one-time encounters the same as frequent, prolonged ones. This prevents any subsequent analysis of tie strengths.

(c) Low resolution, deducing exposure from place visits, which leads to inconsistent exposure thresholds. For example, Athey et al. consider individuals 200m apart within opposite corners of a 153m x 153m grid cell as if they're exposed while individuals 1m apart in adjacent cells are overlooked.

We develop *person-to-person* exposure networks to overcome these limitations. Our temporal co-location exposure measure detects whether two individuals are directly exposed to one another within a narrow window of time (1/2/5 minutes) and space (10/25/50 meters). By building a network in which nodes are individuals, and edges are exposures—we are able to also track the time-resolved exposures between people to study length and repetition of exposures.

The precision of our exposure estimates has a profound effect on our findings compared to Athey et al. (response pp. 68-70; Extended Data Figure 9), and our ability to study tie strengths also unlocks many new analyses (response pp. 82-86).

Constructing this network is a major computational challenge at our level of scale, i.e. 1.6 billion exposures across 9.6 million individuals. A naive implementation would necessitate on the order of

~10 years of compute time. We built a large-scale system to efficiently construct this network, leveraging k-d trees and multi-core parallelization. Our system allows us to efficiently construct this network using a single supercomputer (with 12TB RAM, and 288 cores) in under a week. We now elaborate on these computational challenges in our revision (p. 22, Lines 372-376).

**Person-to-Person
Exposure Network (our study)**

Individual Socioeconomic Status (SES)

**Person-to-Place
Visitation Network (Athey et al.)**

- Majority White Census Block
- Majority Non-White Census Block

Our method accounts for temporal and physical proximity

Actual Exposure
(our study)

Prior Work's Inferred Exposure (Athey et al.)

Actual Exposure
(our study)

Prior Work's Inferred Exposure (Athey et al.)

9 a.m.

9 p.m.

Actual Exposure
(our study)

Prior Work's Inferred Exposure (Athey et al.)

Our method accounts for prolonged and repeated exposures

(2) Individual-level measure of socioeconomic status

Prior work measures exposure between coarse groupings of people (e.g. binary grouping of majority white/non-white census block group residents in Athey et al.). Our study uses a continuous and individual-level SES measure (estimated rent). We are able to identify individual-level socioeconomic status by inferring home locations, which we link to estimated rent values (**Methods M2**). This overcomes known inaccuracies [26, 27] of using home neighborhood as an SES proxy, and lends confidence to our (surprising) rejection of the cosmopolitan hypothesis .

(3) Unbiased segregation estimates in small cities, towns, and rural areas with limited data

Refuting the cosmopolitan hypothesis requires estimating segregation for micropolitan areas, towns, and rural areas. Most prior work (except Athey et al.) focus on a small number of cities (e.g. one city in Abbiasov, eleven cities in Moro et al.). Even Athey et al. focus only on metropolitan areas and do not address micropolitan areas, towns, and rural areas. Our study is the first to be truly nationwide, spanning 2829 counties (over 90% of counties) in addition to 382 MSAs. This is important because the cosmopolitan mixing hypothesis pertains to such nationwide variability in city properties (e.g., population size and segregation).

Athey et al.

(focuses on metropolitan areas)

Our Study

(includes metropolitan areas, micropolitan areas, small towns, and rural areas)

A core methodological challenge is that data are sparse in these low-population areas, which can downwardly bias estimates of segregation (response pp. 7-8 for details). Downward bias in low-population areas would confound our assessment of the cosmopolitan mixing hypothesis, because it would be unclear if decreased exposure segregation is an artifact of estimation bias, rather than due to an actual increase in socioeconomic mixing.

We introduce a mixed model approach to solve this problem and secure unbiased estimates of segregation, even when data are sparse. Although our original manuscript addressed issues of bias

in the section titled “Estimating exposure segregation” (*Methods M3*, pp. 25-27), we have now revised the main section of the manuscript to include a more detailed discussion of our mixed model and why it is necessary to fit it (p. 3, lines 45-46). Below, we highlight *Methods Figure 1*, which shows the advantage of our method compared to a standard approach.

Methods Figure 1: Our estimates compared with naive estimates of the Pearson correlation. We took people who crossed paths with at least 500 other people and computed the Pearson correlation coefficient (the “gold standard estimate”). Then, for each person we randomly sampled 5, 10, 50, 100, and 200 people from the 500+ people and computed segregation estimates based on the reduced sets of people. The left plot shows the ratio of the estimates to the gold standard, for each MSA. The right plot shows the overall number of people in the dataset with $\leq N$ exposures.

New Analyses Unique to our Methods

To showcase the unprecedented scale and resolution of analysis enabled by our methods, we include the following new analyses in our revised manuscript:

- We first highlight our capacity to analyze different relationship types in Figure 2b in our revision (copied below). We show that segregation increases with tie strength and is highest for the strongest ties (5+ exposures; median ES 0.57). This validates our measure of tie strength, as homophily is known to increase with tie strength [28].

- We further break down the different types of exposures by length, distance, and repetition – analyzing associations with specific POIs and downstream outcomes. We elaborate on these in our response below (see response pp. 74-86).
- These analyses set the stage for our alternative measures of exposure segregation which isolates repeated and prolonged exposures (which were included in the original manuscript). Most importantly, we are able to show that our rejection of the cosmopolitan hypothesis is robust to these stricter relationship types (see response pp. 18-19, and Supplementary Figures S7-S8; example copied below).

Minimum Tie Strength: 3 unique days of exposures

- We showcase our ability to obtain unbiased estimates of exposure segregation in data-sparse areas such as small cities, towns, and rural areas via a new figure (Supplementary Figure S30, copied below). This figure disaggregates counties by extent of urbanization, and illustrates why we are the first to be able to fully refute the cosmopolitan mixing hypothesis (see response to R2, response p. 55).

We thank R3 for their helpful comments, which have helped us to more clearly differentiate our paper's novel findings and methods from prior work.

The closest related paper that comes to mind is Athey et al (2020), who use similar GPS data to measure what they term “experienced isolation” between white and black Americans. They find that experienced racial isolation is generally lower than standard segregation measures suggest and that it is higher for interactions close to home and lower in places like public parks. The current paper’s result mirror these findings for economic, rather than racial segregation.

While Athey et al. confirm the cosmopolitan mixing hypothesis, we find overwhelming evidence to reject it. Here, we investigate why they arrive at the *opposite conclusion*, as expressed in the following passage from their work:

“Experienced isolation is relatively lower in MSAs with higher population density... consistent with the fact that in dense areas residents from different neighborhoods are less separated by physical space, and may reflect the role of urban amenities such as parks and public facilities in facilitating diverse interactions (Jacobs 1961).”

In summary, Athey et al.’s incorrect conclusion is due to their imprecise measure of exposure. Upon replicating their population density analysis but instead using our time-sensitive, high-resolution exposure measure, we affirm our initial finding that larger, denser cities are more segregated.

Because our findings are surprising (given the frequently-stated, albeit unsubstantiated, claims that cosmopolitan cities promote intermixing), we included numerous exploratory analyses to build intuition for our metric (**Figure 2a-c**). For instance, **Figure 2a** specifically shows consistency with Athey et al. on, and in doing so helps us to establish the exact point at which they were led astray and why. We agree that these should have been more explicitly flagged as exploratory analyses. We have made the necessary adjustments in our revised manuscript (see response to point 3 on response p. 59).

Detailed Investigation

As we detailed earlier (response pp. 60-63), Athey et al.’s measure is imprecise because they consider two people exposed to each other if they *ever* were in the same 153 x 153 meter cell anytime during a 4 month observation period—whereas we directly measure person-to-person exposure. We analyze the consequences of these methodological differences by reproducing the analysis conducted in Athey et al. Their analysis estimates the correlation between population density and their measure of racial segregation, after controlling for residential (racial) segregation. They do so by correlating the residuals of racial segregation measure with the residuals of a population density, where the residuals are derived from regression on (racial) residential segregation controls.

We find that, when using their time-indifferent and low-resolution measure, there is *no significant correlation* (**Extended Data Figure 9a**; copied below) between population size and racial

segregation (Spearman Corr. -0.04, $p=0.49$), nor between population density and experienced segregation (Spearman Corr. 0.00, $p=0.98$). The correct relationship only emerges, as shown below, when person-to-person exposures are directly measured.

Athey et al.'s own robustness checks further corroborate this point, showing no significant association between population density and segregation (*Athey et al. Supplementary Table S8*; below; "Equal weights to all cities" columns).

Weighted by population size
(statistically significant)

Equal weights to all cities
(null findings)

SI Table S8: Regression Coefficients across Samples

	Baseline	Top 50	Top 100	Top 200
Share with Bachelor's	-0.2491 (0.0428)	-0.426 (0.1221)	-0.3269 (0.0812)	-0.1403 (0.0647)
Median income (thousands)	-0.0017 (0.0005)	-0.0032 (0.0014)	-0.0025 (0.0008)	-0.0007 (0.0008)
Unemployment rate	1.72 $p = 0.03$	1.1 $p = 0.10$	1.8 $p = 0.79$	1.1 $p = 0.15$
White mobility measure	-0.23	-0.4	-0.2	-0.17
Black mobility measure	-0.146 (0.106)	-1.3877 (0.374)	-0.5263 (0.2158)	0.0567 (0.525)
log(Population density)	-0.0015 (0.0007)	-0.0044 (0.0026)	-0.0003 (0.0011)	-0.0013 (0.0009)
Public transit use	-0.012 (0.0026)	-0.0273 (0.0073)	-0.0122 (0.0052)	0.0015 (0.0041)
Median age	-0.1236 (0.0295)	-0.3949 (0.0881)	-0.3376 (0.0888)	-0.245 (0.104)

Notes: We report the coefficient and standard error from our baseline population weighted regression of experienced isolation on fifteen residential isolation bin fixed effects and the specified covariate. We also consider the same regression unweighted and estimated on subsamples of the top 50, 100, and 200 most populous MSAs.

Note: 166 small MSAs are discarded from the unweighted analysis (of the 366 MSAs that Athey analyze)

Next, for reasons that are unclear to us, Athey et al. decided to re-weight their correlations by population size ("Baseline" column in the table above). This approach means that, in effect, only the big cities matter (e.g., New York City receives 100x more weight in their regressions than the majority of cities). With this re-weighting, Athey et al. are able to tease out a slight negative association that is barely statistically significant ($p=0.03$).

Although they interpret this as support for the cosmopolitan hypothesis, it only arises because of their (1) imprecise (time-indifferent and low-resolution) measure of exposure and (2) downweighting of small cities. Both of these operational decisions have a profound effect on the estimate. When we run the same analysis across *all* cities (*Extended Data Figure 9b*; copied above on response p. 69), and use our own time-sensitive and high-resolution racial segregation measure, we find positive and significant relationships between racial exposure segregation and population size (Spearman Corr. 0.31, $p < 1e-9$) as well as population density (Spearman Corr. 0.16, $p < 0.001$). This affirms our rejection of the cosmopolitan hypothesis, which we show in our paper holds across different co-location thresholds (10/25/50 meters distance), time thresholds (1/2/5 minutes apart), tie strengths (repetition and duration), segregation types (SES or racial), and area granularities (MSA or county).

Based on our robust observations we have no choice but to reject the cosmopolitan mixing hypothesis (*Figure 1c-d*). We are also the first to explain why the hypothesis falls short (*Figure 2d-f*), and to explore how urban design can potentially help to reintegrate large cities (*Figure 3*). We elaborate on two contributions in our response to all referees (see response pp. 3-4). In our revision, we emphasize that these are our key discoveries..

The MIT media lab has an online tool that shows measure of the extent of inequality for different places across the US using a measure very similar to the current paper. Relative to the tool, the current paper covers a much larger number of MSAs and locations.

The purpose of our paper is to examine the cosmopolitan mixing hypothesis. Because that's our objective, there are key differences between our study and the MIT media lab study by Moro et al. [25] pertaining to the *unit of analysis* (city-wide and county-wide segregation in our study vs. POI-level segregation in Moro et al.) and *scale* (382 MSAs and 2829 counties in our study vs. 11 cities in Moro et al.). These differences (best highlighted in **Figure 1d**, copied below) enable us to compare across cities and counties, resulting in a number of novel findings (i.e., rejecting the cosmopolitan mixing hypothesis, discovering the hub bridging effect).

Moro et al. measure cross-class exposure for *individual POIs across eleven cities*, and are unable to draw statistical comparisons between cities (nor do they attempt to do so). We develop a single measure of segregation across an *entire geographical area*, and amass enough data to draw comparisons between *382 MSAs and 2829 counties*. This means that we are able to test the cosmopolitan mixing hypothesis and understand why it falls short, whereas Moro et al. cannot do either.

A city-wide (or county-wide) measure of exposure segregation is fundamentally different from an individual-level or POI-level one. Namely, it allows us to (a) rigorously evaluate the cosmopolitan mixing hypothesis, (b) examine why large cities are extremely segregated, and (c) understand factors of the cityscape that may counteract the segregative tendencies of large cities. As we have mentioned previously, these analyses (**Figure 1**, **Figure 2d-f**, and **Figure 3**) constitute our novel discoveries. In our revision, we have also referenced Moro et al. in our discussion of **Figure 2c** (p. 7, lines 89-90) to clarify that it is an exploratory analysis.

The current paper also shows how points of interests that lie between neighborhoods of different demographics can serve as bridges to foster interactions of different types of individuals. A similar idea focusing on NYC parks was recently presented by Abbasov (2021) focusing on racial segregation rather than economic segregation.

We have carefully examined Abbasov’s study [33]. It is very different from our study. We study the macro-level effect of *where POIs (or hubs) are located on citywide segregation*, whereas Abbasov studies the micro-level effect of *where people live (i.e., how close to a park they are) on who they encounter*.

Because Abbasov only studies *one city*, they are restricted to an individual-level analysis. Abbasov shows that *individuals* who live near a park have more racially diverse encounters.

Although their analysis might be taken to suggest that more parks should be built, we show that this would likely worsen citywide segregation (insofar as new parks continue to be located as they have been in the past). We find that as the number of parks increases, they become localized to specific neighborhoods, and don’t have an overall integrative effect at the city-level. That is, even if they may do a bit of mixing work *within* the neighborhood (as Abbasov find), the problem is that they do not lead to diverse encounters *across* neighborhoods. Because cross-neighborhood mixing is where the main integrative potential lies (given how segregated neighborhoods are), Abbasov’s findings, taken alone, miss the big picture.

This is why our results are in tension with Abbasov’s results. Our city-level analysis (**Figure 2c** left; **Extended Data Figure 2** right), copied below, shows that parks are more segregated than POIs that serve the broader city (e.g. stadiums), because parks are localized to specific neighborhoods. By extension, building more parks could further fragment a city with more localized spaces, which increases citywide segregation.

Our analysis implies that interventions to reduce segregation need to build POIs *between diverse neighborhoods* (rather than simply building more POIs overall). We also generalize our main conclusions to hubs (e.g., shopping malls), which are more important for integrating cities because they have much higher traffic and are thus associated with more exposures. We develop a city-wide index which measures hub bridging, and discover that when such hubs are located between diverse communities, city-wide segregation is reduced.

This type of city-level analysis is only possible via a nationwide study spanning hundreds of cities, as opposed to Abbasov's analysis of a single city. Our key finding (**Figure 3**, copied below) is that hub bridging is a significantly stronger predictor of exposure segregation (Spearman Corr -0.78; $p < 0.001$) than population size, SES inequality, residential segregation, and racial demographics. Our discovery paves the way for policy solutions to re-integrate large cities (i.e., policies such as zoning laws or subsidies that encourage developers to locate hubs between diverse residential neighborhoods).

Interactions?

The paper calls its key measure interaction segregation. An interaction is defined as coming within 50 meters of a person within 5 minutes. I would have liked to see a deeper discussion of what type of interactions are captured by this and to what extent and in which settings they matter.

Specifically, the current measure captures whether I cross paths with someone. There is no way of telling with the current data whether two individuals simply spend time in the same restaurant or coffee shop or whether a deeper interaction such as speaking to each other took place.

R3 argues – very convincingly – that it’s useful to distinguish between people who simply occupy the same space and time, and deeper forms of interaction such as conversation. As mentioned in our response summary, we agree with R3 that the label “interaction” may have been misleading. We have thus substituted “interaction” with “exposure”.

Our terminology now better reflects the core aim of our study, which is to assess the cosmopolitan mixing hypothesis. As we further clarify in our response summary (response p. 1), the cosmopolitan mixing hypothesis itself pertains to visual exposure (e.g. sightings) and not deeper interactions (e.g., speaking). As we have discussed throughout this memo, the vast majority of exposures in city life are happenstance and do not entail any deeper interaction.

The cosmopolitan hypothesis focuses on exposure because exposure is important. It’s important because of (a) its short-term and long-term effects on the social psychology of group relations [5, 6, 7], emotional well-being [10, 11, 12], political view formation [13], transmission of infectious diseases [8], vulnerability to crime [9], and more [14, 15, 16, 17]; (b) a normative commitment to integration that values intermixing in and of itself [21, 22] and thus calls for high-quality measurement of the extent to which we’re living up to that commitment; and (c) it is a prerequisite for many deeper social relationships, such as friendships. **We explore the associations between the exposures we measure and downstream outcomes in our next responses.**

As was also discussed earlier, we agree with R3 that it’s useful to distinguish between the different types of exposures captured in our network (i.e., tie strength). As shown in Supplementary Figure S6 and Supplementary Table S11 (copied below), the vast majority of exposures are weak ties in the form of one-off path-crossings (74.24% of pairs occurred only a single time). A minority of edges are ties of medium strength (e.g., 2.32% of pairs shared exposures for three or more days), which may indicate acquaintances. Lastly, a small minority of pairs share the same home (0.29%), indicating the possibility of strong ties (e.g., housemates, partners, etc.). The table below reports on these types of exposures as well as many others.

Feature	Pearson Corr. w/ Primary	Spearman Corr. w/ Primary	Median	Mean	% Pairs	% People
Primary Measure	—	—	0.35	0.35	100.00	100.00
Primary Measure (+ Up-weight Multiple Exposures)	0.89	0.91	0.46	0.45	100.00	100.00
SES Definition: Rent Zestimate Percentile	0.88	0.89	0.42	0.42	100.00	100.00
SES Definition: Within-MSA Rent Zestimate Percentile	0.81	0.83	0.54	0.53	100.00	100.00
SES Definition: Census Median Household Income	0.75	0.77	0.47	0.46	100.00	100.00
SES Definition: Educational Attainment (% College or Higher)	0.70	0.71	0.52	0.52	100.00	100.00
Exclude Pri/Sec Roads	0.99	0.99	0.37	0.37	74.30	99.87
Exclude Roads	0.98	0.98	0.37	0.37	38.66	98.56
Stationary Individuals (2 pings < 10 meters in 1-10 min)	0.86	0.87	0.44	0.43	5.39	79.46
Exclude Same-home exposures	0.98	0.98	0.34	0.34	99.71	99.78
Work/Leisure (Neither in Home Tract)	0.93	0.93	0.31	0.31	86.26	95.55
Leisure (inside POI)	0.85	0.84	0.28	0.29	16.15	76.32
Minimum Distance Between Pings: < 25 meters	0.97	0.97	0.34	0.34	49.53	97.71
Minimum Distance Between Pings: < 10 meters	0.95	0.94	0.33	0.33	20.93	92.28
Minimum Time Between Pings: < 2 minutes	0.97	0.97	0.34	0.34	53.59	98.92
Minimum Time Between Pings: < 60 seconds	0.97	0.97	0.35	0.35	33.67	97.83
Minimum Tie Strength: 2 consecutive exposures	0.94	0.95	0.35	0.35	18.25	94.80
Minimum Tie Strength: 3 consecutive exposures	0.83	0.83	0.37	0.37	3.06	65.46
Minimum Tie Strength: 2 unique days of exposure	0.88	0.90	0.47	0.46	7.46	83.56
Minimum Tie Strength: 3 unique days of exposure	0.73	0.77	0.56	0.54	2.32	62.81
Dist. < 25 meters, Time < 2 min., >= 2 consec. exposures	0.92	0.93	0.36	0.35	8.80	86.64
Dist. < 25 meters, Time < 2 min., >= 2 unique days	0.87	0.89	0.46	0.44	3.84	70.57
Dist. < 10 meters, Time < 60 sec., >= 3 consec. exposures	0.78	0.79	0.39	0.38	0.94	38.16
Dist. < 10 meters, Time < 60 sec., >= 3 unique days	0.68	0.68	0.52	0.51	0.58	28.92
Downweight Simultaneous Exposures	0.97	0.97	0.34	0.34	99.99	99.99
Exclude Simultaneous Exposures	0.94	0.95	0.46	0.46	22.87	99.61
Tie Strength: 1 Exposure	0.97	0.98	0.31	0.32	74.24	98.66
Tie Strength: 2 Exposures	0.95	0.96	0.33	0.33	16.14	93.20
Tie Strength: 3 Exposures	0.89	0.90	0.36	0.36	4.09	74.67
Tie Strength: 4 Exposures	0.85	0.86	0.37	0.36	1.82	58.51
Tie Strength: 5+ Exposure	0.83	0.85	0.45	0.45	3.42	70.62
Minimum Stationary Nights: 6 nights	0.99	0.99	0.35	0.35	77.45	83.91
Minimum Stationary Nights: 9 nights	0.99	0.99	0.35	0.36	68.64	73.42
Minimum Stationary Nights: 12 nights	0.98	0.98	0.35	0.36	59.16	63.66
Racial Segregation (% Non-White)	0.59	0.60	0.46	0.46	100.0	100.0
Economic Segregation (White Overall)	0.94	0.95	0.34	0.34	55.97	63.83
Economic Segregation (White Within-group)	0.93	0.94	0.34	0.34	39.84	63.69
Economic Segregation (Non-White Overall)	0.55	0.52	0.28	0.28	43.76	35.50
Economic Segregation (Non-White Within-group)	0.47	0.44	0.28	0.30	27.50	35.31

Importantly, we find that the exposure segregation of these stronger ties is highly correlated to our primary measure, which includes all exposures (Spearman Corr. 0.77-95; see above table, Spearman and Pearson Corr. w/ Primary columns). Our rejection of the cosmopolitan mixing hypothesis, and discovery of the hub-bridging effect, generalizes to these stronger ties as well (see response p. 66, pp. 18-19, and **Supplementary Figures S7-S8**; example copied below). This is expected because, when one’s day-to-day social environment favors homophilous mixing, some of this mixing will “grow” into stronger ties that are also more homophilous.

Minimum Tie Strength: 3 unique days of exposures

Percentile	Tie Strength	Percentile	Tie Strength
0.0	1	51.0	1
1.0	1	52.0	1
2.0	1	53.0	1
3.0	1	54.0	1
4.0	1	55.0	1
5.0	1	56.0	1
6.0	1	57.0	1
7.0	1	58.0	1
8.0	1	59.0	1
9.0	1	60.0	1
10.0	1	61.0	1
11.0	1	62.0	1
12.0	1	63.0	1
13.0	1	64.0	1
14.0	1	65.0	1
15.0	1	66.0	1
16.0	1	67.0	1
17.0	1	68.0	1
18.0	1	69.0	1
19.0	1	70.0	1
20.0	1	71.0	1
21.0	1	72.0	1
22.0	1	73.0	1
23.0	1	74.0	1
24.0	1	75.0	2
25.0	1	76.0	2
26.0	1	77.0	2
27.0	1	78.0	2
28.0	1	79.0	2
29.0	1	80.0	2
30.0	1	81.0	2
31.0	1	82.0	2
32.0	1	83.0	2
33.0	1	84.0	2
34.0	1	85.0	2
35.0	1	86.0	2
36.0	1	87.0	2
37.0	1	88.0	2
38.0	1	89.0	2
39.0	1	90.0	2
40.0	1	91.0	3
41.0	1	92.0	3
42.0	1	93.0	3
43.0	1	94.0	3
44.0	1	95.0	4
45.0	1	96.0	4
46.0	1	97.0	5
47.0	1	98.0	7
48.0	1	99.0	11
49.0	1	100.0	11644
50.0	1		

Supplementary Table S11: Distribution of interactions edge strength (# of interactions) for all individuals residing in 382 Metropolitan Statistical Areas (MSAs).

The reason we care about segregation and interactions of different people is generally because past research has shown that interactions between different types of people are desirable for many outcomes such as economic mobility, health and reduced political polarization (as cited in the paper). So it would be useful to discuss what different types of interactions accomplish and how they help improve different outcomes.

It may be that crossing paths with different types of people is sufficient for reaping the benefits of increased interactions. Maybe seeing people of different means in the same settings one attends makes them appear less other and helps fostering understanding across class lines, potentially reducing political polarization. However, simply being in the same spaces as others may not actually be sufficient to lead to some of the benefits of reduced segregation. Maybe actual interactions, such as speaking to each other and the exchange of ideas, are necessary to reap those. For instance, recent work by Chetty et al. (2022) suggests that online friendship links, not just exposure, to those of different socio-economic status are associated with higher social mobility.

In short, it would be helpful to show that the reduced extent of crossing paths interactions in bigger and more dense cities are truly associated with less of the relevant mixing across class lines that matters for outcomes. Or whether the increased “interactions” in smaller cities just capture more paths crossings of the type that does not lead to improved outcomes.

These are important comments. In addressing them, it’s useful to recognize the extensive literature that treats integration as an end unto itself, a literature that stems from a normative commitment to building a world in which groups live together rather than in isolation from one another (see, e.g., Ellen and Steil 2019 and Patillo 2021 [21, 22]). From a normative perspective, it’s not necessary to claim, for example, that poor people will necessarily be “uplifted” or otherwise benefited by virtue of having contact with rich people. Rather, we simply want to monitor the extent to which our country is realizing its commitment to building a society in which all people – rich and poor alike – live, work, and play together.

Nevertheless, we agree that it is also of interest to try to estimate the long-term and short-term causal effects of exposure and social mixing, and there is already extensive literature that does so [5-17]. It’s beyond the purview of our paper, which is already quite ambitious, to try to contribute to this very different literature by identifying such causal effects.

We did, however, conduct a number of analyses that show significant associations between our exposure network and downstream outcomes. We have also further extended these in our revision. *In summary, exposures captured by our network are significantly associated with (1) friendship formation (2) upward economic mobility and (3) political polarization.* We elaborate on these points in our response below.

Indeed, **Extended Data Figure 1** (copied below) in our initial manuscript shows that exposure segregation is associated with friendship formation [18] and upward economic mobility [29].

Extended Data Figure 1: This studies’ exposure network predicts population-scale friendship formation and upward economic mobility outcomes. We measure the external validity of our definition of exposure, by linking our exposure network to outcomes across two gold-standard, large-scale, datasets. We find at the zip code, county, and MSA-level, our exposure network mirrors population-scale outcomes resulting from dynamic human processes: **(a-b)** the Facebook Social Connectedness Index⁷³ measures the relative probability of a Facebook friendship link between a given Facebook user in location i and a given user in location j . FB Social Connectedness Index has been used social segregation⁷⁴, and has also been linked to economic^{75,76} and public health outcomes⁷⁷. We reproduce the Social Connectedness Index using our exposure network ($\frac{\#ExposurePairs_{i,j}}{\#Individuals_i \cdot \#Individuals_j}$) at the county **(a)** and zip code **(b)** level, and find strong correlations across county pairs (Spearman Correlation 0.85, $N = 121,595$, $p < 10^{-4}$) and zip code pairs (Spearman Correlation 0.73, $N = 1,053,539$, $p < 10^{-4}$). **(c-d)** The Chetty et al. Intergenerational Mobility dataset quantifies upward economic mobility from federal income tax records for each MSA as the mean income rank of children with parents in the bottom half of the income distribution⁷⁸. We find that exposure segregation at the MSA-level **(c)** correlates to (absolute) upward economic mobility (Spearman Correlation -0.37, $N = 379$, $p < 10^{-4}$), and does so significantly more strongly ($p < 10^{-4}$) than **(d)** the conventional segregation measure NSI (Spearman Correlation -0.12, $N = 379$, $p < 0.05$)

Although we do not aim to identify a causal effect, we have expanded this analysis along two dimensions.

(a) In-depth analysis of the relationship between exposure and friendship formation outcomes

(i) In our revision, we further investigate these findings. Most notably, we show that our exposure network is a stronger predictor of friendship formation than distance ($p < 1e-4$, Steiger's Z-test). For instance, our exposure network explains 76.3% of the variance in friendship formation between counties, whereas distance explains only 53.9% of the variance (*Supplementary Tables S23 and S24*, copied below).

	Dependent variable: Log(FB Social Connectedness)		
	(1)	(2)	(3)
const	11.123*** (0.003)	11.123*** (0.002)	11.123*** (0.002)
Log(Distance km)	-1.163*** (0.003)		-0.253*** (0.003)
Log(Exposure Network Social Connectness)		1.383*** (0.002)	1.190*** (0.003)
Observations	118,559	118,559	118,559
R^2	0.539	0.763	0.773
Adjusted R^2	0.539	0.763	0.773

* $p < 0.1$; ** $p < 0.05$; *** $p < 0.01$

Supplementary Table S23: Our exposure network strongly predicts friendship formation (between counties). Here we show the coefficients (after normalizing via z-scoring to have mean 0 and variance 1) and R^2 from predicting FB network friendship strengths between counties. Column (1) uses only county distance, Column (2) uses only exposure network social connectedness, and Column (3) uses the combination of distance and the exposure network. We find that our exposure network alone explains 76.3% of the variance in friendship formation between counties and is a stronger predictor of friendship formation than distance ($p < 10^{-4}$, Steiger's Z-test).

	Dependent variable: Log(FB Social Connectedness)		
	(1)	(2)	(3)
const	12.372*** (0.001)	12.372*** (0.001)	12.372*** (0.001)
Log(Distance km)	-0.965*** (0.001)		-0.175*** (0.002)
Log(Exposure Network Social Connectness)		1.183*** (0.001)	1.051*** (0.002)
Observations	1,038,424	1,038,424	1,038,424
R^2	0.335	0.503	0.508
Adjusted R^2	0.335	0.503	0.508

* $p < 0.1$; ** $p < 0.05$; *** $p < 0.01$

Supplementary Table S24: Our exposure network strongly predicts friendship formation (between zip codes). Here we show the coefficients (after normalizing via z-scoring to have mean 0 and variance 1) and R^2 from predicting FB network friendship strengths between counties. Column (1) uses only zip code distance, Column (2) uses only exposure network social connectedness, and Column (3) uses the combination of distance and the exposure network. We find that our exposure network alone explains 50.3% of the variance in friendship formation between zip codes and is a stronger predictor of friendship formation than distance ($p < 10^{-4}$, Steiger's Z-test).

(ii) Using New York City boroughs as a case study, we further show that there are strong correlations between exposure and friendship formation even within a fine-grained subsection of a city (Supplementary Figure S31, copied below). Specifically, zip codes in which residents have more exposure to each other also have more friendships (on the Facebook platform).

Supplementary Figure S31: Exposure network connections strongly correlate to friendship formation even within fine-grained geographical areas. We reproduce the the Facebook Social Connectedness Index⁷³ at zip code-level (Extended Data Figure 1) for each of the five boroughs of New York City, and find strong correlations to online friendships in all five boroughs (Spearman Correlations 0.66-0.88, all $p < 10^{-4}$). Strong correlations suggest that exposure segregation is likely related to segregation of friendships and other strong social ties.

The results could plausibly speak to causal effects because exposure is a necessary prerequisite for all other deeper forms of interaction, ranging from speaking to strong friendship formation. This further motivates our focus on cross-class exposure, as increasing cross-class exposure is the first step towards increasing cross-class conversations, friendships, and so forth.

(b) Associations with political polarization outcomes (*Supplementary Figure S23*, copied below). Adding additional outcomes, as R3 suggests, is challenging because of limited data availability at the national scale. However, we were able to identify one such measure of partisan prejudice at the county-level from TheAtlantic / PredictWise [30]. We find that among both Democrats and Republicans, prejudice towards the opposite political party is significantly higher ($p < 1e-4$) in counties with higher exposure segregation.

Supplementary Figure S23: Exposure segregation predicts political polarization outcomes. We measure the external validity of our definition of exposure segregation, by linking our measure to outcomes from a large-scale survey of political polarization⁹⁰. We find that county-level exposure segregation correlates to political prejudice among both (a) Democrats (Spearman Correlation 0.30, $N = 2828$, $p < 10^{-4}$) and (b) Republicans (Spearman Correlation 0.26, $N = 2828$, $p < 10^{-4}$). These findings suggest that exposure to diverse others may lead to increased tolerance of inter-group differences, following following prior work¹⁸.

It should be noted that the paper provides substantial robustness in how an interaction is defined, using more restrictive definitions with respect to time, length and distance and repeated interactions. Still, a concern is that even though I might pass the same person daily on my way to work, I may never actually talk to or more intensely interact with them.

It may nevertheless be helpful to characterize interactions more in terms of length, distance, repetition and how these characteristics differ across settings and MSAs to better understand which types of crossing paths might be associated with more or less additional interactions (such as speaking) and whether that relates to outcomes.

We thank R3 for these spot-on comments. As we have stressed, while the cosmopolitan mixing hypothesis motivates our focus on exposure and not on stronger ties (e.g., speaking), we agree with R3 that it is informative to (a) break down the different types of exposures and where they occur, and (b) examine whether the associations with outcomes are robust to exposure type. These analyses further highlight the novelty of our *person-to-person* exposure network (which captures exposure length, distance, and repetition), and would not be possible using earlier research on *person-to-place* networks.

Firstly, as mentioned in our earlier response, we have now added (in Supplementary Table S6, copied below) a breakdown by the % of exposure pairs that fit each robustness check. This speaks to the frequency of each exposure type.

Feature	Pearson Corr. w/ Primary	Spearman Corr. w/ Primary	Median	Mean	% Pairs	% People
Primary Measure	—	—	0.35	0.35	100.00	100.00
Primary Measure (+ Up-weight Multiple Exposures)	0.89	0.91	0.46	0.45	100.00	100.00
SES Definition: Rent Zestimate Percentile	0.88	0.89	0.42	0.42	100.00	100.00
SES Definition: Within-MSA Rent Zestimate Percentile	0.81	0.83	0.54	0.53	100.00	100.00
SES Definition: Census Median Household Income	0.75	0.77	0.47	0.46	100.00	100.00
SES Definition: Educational Attainment (% College or Higher)	0.70	0.71	0.52	0.52	100.00	100.00
Exclude Pri/Sec Roads	0.99	0.99	0.37	0.37	74.30	99.87
Exclude Roads	0.98	0.98	0.37	0.37	38.66	98.56
Stationary Individuals (2 pings < 10 meters in 1-10 min)	0.86	0.87	0.44	0.43	5.39	79.46
Exclude Same-home exposures	0.98	0.98	0.34	0.34	99.71	99.78
Work/Leisure (Neither in Home Tract)	0.93	0.93	0.31	0.31	86.26	95.55
Leisure (inside POD)	0.85	0.84	0.28	0.29	16.15	76.32
Minimum Distance Between Pings: < 25 meters	0.97	0.97	0.34	0.34	49.53	97.71
Minimum Distance Between Pings: < 10 meters	0.95	0.94	0.33	0.33	20.93	92.28
Minimum Time Between Pings: < 2 minutes	0.97	0.97	0.34	0.34	53.59	98.92
Minimum Time Between Pings: < 60 seconds	0.97	0.97	0.35	0.35	33.67	97.83
Minimum Tie Strength: 2 consecutive exposures	0.94	0.95	0.35	0.35	18.25	94.80
Minimum Tie Strength: 3 consecutive exposures	0.83	0.83	0.37	0.37	3.06	65.46
Minimum Tie Strength: 2 unique days of exposure	0.88	0.90	0.47	0.46	7.46	83.56
Minimum Tie Strength: 3 unique days of exposure	0.73	0.77	0.56	0.54	2.32	62.81
Dist. < 25 meters, Time < 2 min., >= 2 consec. exposures	0.92	0.93	0.36	0.35	8.80	86.64
Dist. < 25 meters, Time < 2 min., >= 2 unique days	0.87	0.89	0.46	0.44	3.84	70.57
Dist. < 10 meters, Time < 60 sec., >= 3 consec. exposures	0.78	0.79	0.39	0.38	0.94	38.16
Dist. < 10 meters, Time < 60 sec., >= 3 unique days	0.68	0.68	0.52	0.51	0.58	28.92
Downweight Simultaneous Exposures	0.97	0.97	0.34	0.34	99.99	99.99
Exclude Simultaneous Exposures	0.94	0.95	0.46	0.46	22.87	99.61
Tie Strength: 1 Exposure	0.97	0.98	0.31	0.32	74.24	98.66
Tie Strength: 2 Exposures	0.95	0.96	0.33	0.33	16.14	93.20
Tie Strength: 3 Exposures	0.89	0.90	0.36	0.36	4.09	74.67
Tie Strength: 4 Exposures	0.85	0.86	0.37	0.36	1.82	58.51
Tie Strength: 5+ Exposure	0.83	0.85	0.45	0.45	3.42	70.62
Minimum Stationary Nights: 6 nights	0.99	0.99	0.35	0.35	77.45	83.91
Minimum Stationary Nights: 9 nights	0.99	0.99	0.35	0.36	68.64	73.42
Minimum Stationary Nights: 12 nights	0.98	0.98	0.35	0.36	59.16	63.66
Racial Segregation (% Non-White)	0.59	0.60	0.46	0.46	100.0	100.0
Economic Segregation (White Overall)	0.94	0.95	0.34	0.34	55.97	63.83
Economic Segregation (White Within-group)	0.93	0.94	0.34	0.34	39.84	63.69
Economic Segregation (Non-White Overall)	0.55	0.52	0.28	0.28	43.76	35.50
Economic Segregation (Non-White Within-group)	0.47	0.44	0.28	0.30	27.50	35.31

We have also conducted two new analyses to address these questions:

(a) We analyze how the length, distance, and repetition of exposures vary across different settings (POIs), as suggested by R3 (Supplementary Tables S15-S17, copied below). As GPS pings are sparse, ranking POIs by these measures allows for relative comparison between POIs (but not necessarily the ability to know the true value of length, distance, and repetition).

Overall, we find that *repetition* and *length* appear to relate to the underlying nature of the exposure. For instance, repeated exposure occurs most at religious organizations, golf courses and country clubs, and fitness/recreation centers—which all have relatively static memberships and a norm of frequent attendance. Similarly, longer exposures occur most at performing arts centers and stadiums—venues in which individuals are likely to stay at the same seat for a prolonged period. By contrast, *distance* appears to be a by-product of data sparsity induced by the built environment (e.g., cellphones are discouraged at casinos, so data are sparse and exposures appear more distanced). Thus, our subsequent robustness checks for outcomes focus on repeated exposure and prolonged exposure.

Mean # of Unique Days of Exposure	POI Type
1.027096	Performing Arts Centers
1.029548	Stadiums
1.036697	Theme Parks
1.044692	Bowling Centers
1.050582	Other Amusement/Recreation
1.054688	Bars/Drinking Places
1.068597	Museums
1.068676	Historical Sites
1.073173	Independent Artists
1.087597	Casinos
1.089572	Limited-Service Restaurants
1.092399	Parks
1.097890	Snack Bars
1.102125	Full-Service Restaurants
1.117464	Fitness/Recreation Centers
1.147761	Golf Courses and Country Clubs
1.153269	Religious Organizations

Supplementary Table S15: Exposure repetition by setting. For each leisure POI category, we compute the mean number of unique days of exposure over all pairs of individuals in the exposure network. POIs associated with most repeated exposures (religious organizations, golf courses and country clubs, and fitness centers) are all venues with relatively static membership structures (e.g. religious affiliation, annual gym membership) and in which frequent attendance is a norm (e.g. visiting church every Sunday, weekly workout). By contrast, the POIs with least repetition (performing arts centers and stadiums) are those which are typically attended only special occasions and typically without a commitments (e.g. buying a single ticket to see a sports game).

Mean Length (# of consecutive five minute intervals)	POI Type
1.617740	Museums
1.620565	Theme Parks
1.693283	Other Amusement/Recreation
1.748765	Bars/Drinking Places
1.776146	Independent Artists
1.782905	Limited-Service Restaurants
1.840704	Full-Service Restaurants
1.851464	Snack Bars
1.871852	Casinos
1.882052	Fitness/Recreation Centers
1.910205	Historical Sites
2.039442	Parks
2.061254	Religious Organizations
2.082550	Golf Courses and Country Clubs
2.209579	Bowling Centers
2.436712	Stadiums
2.464210	Performing Arts Centers

Supplementary Table S17: Exposure length by setting. For each leisure POI category, we compute the mean length during exposure over all pairs of individuals in the exposure network (in number of consecutive 5 minute intervals). POIs associated with longest exposures (performing arts centers, stadiums) are those in which attendance is typically prolonged and mobility is restricted (e.g. sitting in the same seat for multiple hours to watch a game). By contrast, the POIs with shortest exposure (museums, theme parks) are those which mobility is a necessary part of the experience (e.g. walking to different exhibits or attractions).

Mean Exposure Distance (meters)	POI Type
14.148335	Museums
20.783428	Historical Sites
23.343675	Golf Courses and Country Clubs
24.539538	Religious Organizations
24.954691	Stadiums
25.063481	Performing Arts Centers
25.168213	Bowling Centers
25.765411	Full-Service Restaurants
25.845487	Parks
25.973838	Other Amusement/Recreation
26.080283	Bars/Drinking Places
26.102137	Fitness/Recreation Centers
26.214838	Theme Parks
26.305469	Limited-Service Restaurants
26.423661	Snack Bars
26.775229	Independent Artists
27.048224	Casinos

Supplementary Table S16: Exposure distance by setting. For each leisure POI category, we compute the mean distance during exposure over all pairs of individuals in the exposure network. POIs associated with furthest exposures (casinos, independent artists) are those in which mobile phone usage is typically restricted (e.g. many casinos do not allow mobile phone usage to ensure fair play), which is likely to lead to sparse GPS pings. By contrast, the POIs with least distance (museums, historical sites) are those which phones may be actively used to enhance the experience (e.g. to take photos, look up information, or use a virtually guided tour).

(b) In newly added *Supplementary Figures S22* and *S32* (copied below), we repeat our analyses of associations between exposure and outcomes from *Extended Data Figure 1* for *prolonged exposure* (i.e., 3+ consecutive exposures over a five minute intervals) and *repeated exposure* (i.e., 3+ unique days of exposure), compared to path-crossings (single exposures between two individuals). Overall we find that associations are robust to choice of exposure definition — highlighting the importance of capturing all forms of exposures, even those which are “mere” path crossings. These results build on

the social theory of the strength of weak ties [31, 32] and validate our choice of primary metric as the set of all exposures (as opposed to only prolonged or repeated exposures).

Supplementary Figure S22: We measure the external validity of alternative measures in which the strictness of our interaction definition is varied. We compute interaction segregation using only prolonged exposure of 3+ consecutive intervals of exposure on the same day, repeated exposure of 3+ consecutive intervals of exposure on different days, and path crossings (i.e. pairs of users that had only one instance of being within proximity of each other). We find all measures of interaction segregation correlate to (absolute) upward economic mobility (Spearman Correlation -0.31, $N = 365$, $p < 10^{-4}$), (Spearman Correlation -0.22, $N = 364$, $p < 10^{-4}$), and (Spearman Correlation -0.37, $N = 382$, $p < 10^{-4}$) respectively. Interestingly, the measure with the weakest definition of exposure (path-crossings) has the strongest correlation, which may reflect the strength of weak ties in shaping upward economic mobility^{86,87}.

Supplementary Figure S32: We measure the external validity of alternative measures in which the strictness of our exposure definition is varied. We filter the exposure network to include (a-b) only prolonged exposure of 3+ consecutive intervals of exposure on the same day, (c-d) repeated exposure of 3+ consecutive intervals of exposure on different days, and (e-f) path crossings (i.e. pairs of users that had only one instance of being within proximity of each other). We find that social connectedness across all three definitions of exposure correlates strongly to social connectedness measured by online friendship linked (detailed in Extended Data Figure 1)

Mechanism and relationship with race

Related to what types of interactions are captured in the paper, I would have liked to see a bit more discussion on the mechanisms or drivers of the results. There currently is very little in the paper beyond the result on bridging zones.

Given the related work on racial segregation, it would have also been interesting to look more closely at whether there are interaction effects of race and socioeconomic standing. It is well-known that race and socio-economic standing correlate. So it would be very interesting to see to what extent segregation by race and income are driving each other. Is there more interaction across the socio-economic spectrum within race? Are income and racial segregation similar across settings or are some settings more segregated by race or income (e.g., one could imagine restaurants to be more segregated by income)?

We focus on the bridging results in our main manuscript because hub bridging is (1) actionable and (2) *strongly* associated with exposure segregation, more so than many other city properties. Nevertheless, R3 is absolutely right that we should undertake further analysis of race and racial segregation. We include four new analyses which do so in our revision (see next pages below).

- (1) Hub locations are an actionable aspect of the built environment. It is possible, in other words, to design interventions (e.g. to increase the Bridging Index of a city via zoning laws or subsidies) and observe their effect. We hope that our findings will motivate future work to assess the causal effect of bridging zones on citywide segregation.
- (2) The Bridging Index is *strongly* associated with exposure segregation (Spearman Corr. -0.78). We did consider a number of other aspects of the built environment and city demographics (including race) in our initial supplementary information (***Extended Data Table 3***, copied below) and found weaker associations. Overall, we consider the focus on a single, strong explanation as a strength of our paper, as opposed to highlighting many moderate-to-weak (but statistically significant) explanations. This is especially so because it addresses the key dynamic (i.e, socioeconomic targeting of urban space) that – throughout our analyses – consistently accounts for extreme segregation in large cities.

Measure	Spearman ρ^2	Pearson R^2
Bridging Index	0.60	0.62
Log(Population Size)	0.39	0.35
Gini Index (Estimated Rent)	0.41	0.42
Political Alignment (% Democrat in 2016 Election)	0.06	0.05
Racial Demographics (% non-Hispanic White)	0.09	0.05
Mean ES (Estimated Rent)	0.09	0.05
Walkability (Walkscore)	0.01	0.02
Commutability (% Commute to Work)	0.04	0.03
Conventional Segregation (NSI)	0.44	0.42
# of Interaction Hubs	0.44	0.16

But of course R3 is absolutely right that we should undertake further analysis of race and racial segregation (that goes beyond, of course, simply replicating the work of Athey et al.). We include three new analyses in our paper (summarized below) to answer the questions highlighted by R3, as well as a fourth analysis which compares our time-sensitive and high-resolution measure of racial segregation with the measure used by Athey et al. (response pp. 68-70).

(a) First, we compute an alternative measure of exposure segregation that uses race (% non-White within home census block group) instead of socioeconomic status (Supplementary Figure S33 top, copied below). We observe that our key findings that large, dense cities are more segregated, and segregation is reduced when hubs bridge diverse neighborhoods—both generalize to racial exposure segregation.

We find that, overall, racial exposure segregation is 31% higher in the median MSA compared to the economic exposure segregation in the median MSA. Moreover, there is much higher variance in racial exposure segregation: the standard deviation of racial exposure segregation is 114% higher than the standard deviation of economic exposure segregation.

(b) We compare how economic vs. racial exposure segregation varies by POI (Supplementary Figure S34, copied below). As R3 insightfully anticipates, restaurants are 47% more segregated by socioeconomic status than by race. At the same time, golf courses and country clubs, as well as religious organizations, are the most segregated across both SES and race.

(c) We examine interaction effects between racial demographics and economic segregation (Supplementary Figure S33, copied below). We label individuals as either “white” / “non-white” (residing in a census block group in which the majority of residents are white / non-white). For each group, we calculate two measures of exposure segregation (1) overall exposure segregation using all exposures, and (2) within-race group exposure segregation using only exposure to individuals belonging to the same group.

Overall exposure segregation is 27% higher for white individuals. However, for both groups, median economic segregation is approximately the same when comparing within-race exposures to overall exposures.

Overall, we thank R3 for highlighting these excellent opportunities for further analysis. These suggestions have resulted in a more informative and expansive manuscript. Thank you!

References

- [1] Jacobs, J. *The Death and Life of Great American Cities* (Random House, 1961).
- [2] Wirth, Louis. "Urbanism as a Way of Life." *American Journal of Sociology* 44.1 (1938).
- [3] Simmel, Georg. "The metropolis and mental life." *The urban sociology reader*. Routledge, 2012. 37-45.
- [4] Milgram, S. The experience of living in cities. *Science* 167, 1461–1468 (1970)
- [5] Lee, Barrett A., Chad R. Farrell, and Bruce G. Link. "Revisiting the contact hypothesis: The case of public exposure to homelessness." *American Sociological Review* 69.1 (2004): 40-63.
- [6] Hässler, Tabea, et al. "A large-scale test of the link between intergroup contact and support for social change." *Nature Human Behaviour* 4.4 (2020): 380-386.
- [7] Christ, Oliver, et al. "Contextual effect of positive intergroup contact on outgroup prejudice." *Proceedings of the National Academy of Sciences* 111.11 (2014): 3996-4000.
- [8] Chang, Serina, et al. "Mobility network models of COVID-19 explain inequities and inform reopening." *Nature* 589.7840 (2021): 82-87.
- [9] Clarke, Ronald V., and Derek B. Cornish. "Modeling offenders' decisions: A framework for research and policy." *Crime and Justice* 6 (1985): 147-185.
- [10] Sandstrom, Gillian M., and Elizabeth W. Dunn. "Social interactions and well-being: The surprising power of weak ties." *Personality and Social Psychology Bulletin* 40.7 (2014): 910-922.
- [11] Huxhold, Oliver, et al. "The strength of weaker ties: An underexplored resource for maintaining emotional well-being in later life." *The Journals of Gerontology: Series B* 75.7 (2020): 1433-1442.
- [12] Gunaydin, Gul, et al. "Minimal social interactions with strangers predict greater subjective well-being." *Journal of Happiness Studies* 22 (2021): 1839-1853.
- [13] Brown, Jacob R., et al. "Childhood cross-ethnic exposure predicts political behavior seven decades later: Evidence from linked administrative data." *Science Advances* 7.24 (2021): eabe8432.
- [14] Charmaz, Kathy, Scott R. Harris, and Leslie Irvine. *The social self and everyday life: Understanding the world through symbolic interactionism*. John Wiley & Sons, 2019.
- [15] Paluck, Elizabeth Levy, Seth A. Green, and Donald P. Green. "The contact hypothesis re-evaluated." *Behavioural Public Policy* 3.2 (2019): 129-158.
- [16] Zajonc, Robert B. "Mere exposure: A gateway to the subliminal." *Current directions in psychological science* 10.6 (2001): 224-228.

- [17] Bornstein, R. F., and C. Craver-Lemley. "Mere exposure effect In Pohl RF (Ed.), Cognitive illusions: A handbook on fallacies and biases in thinking, judgment and memory (pp. 215–234)." (2004).
- [18] Bailey, Michael, et al. "Social connectedness: Measurement, determinants, and effects." *Journal of Economic Perspectives* 32.3 (2018): 259-80.
- [19] Chen, M. Keith, and Ryne Rohla. "The effect of partisanship and political advertising on close family ties." *Science* 360.6392 (2018): 1020-1024.
- [20] Athey, Susan, et al. "Estimating experienced racial segregation in US cities using large-scale GPS data." *Proceedings of the National Academy of Sciences* 118.46 (2021): e2026160118.
- [21] Ellen, Ingrid, and Justin Steil, eds. *The dream revisited: Contemporary debates about housing, segregation, and opportunity.* Columbia University Press, 2019.
- [22] Pattillo, Mary. "Black advantage vision: Flipping the script on racial inequality research." *Issues in Race & Society* 10.1 (2021): 5-39.
- [23] Chevrier, Raphaël, et al. "Use and understanding of anonymization and de-identification in the biomedical literature: scoping review." *Journal of medical Internet research* 21.5 (2019): e13484.
- [24] Chetty, Raj, et al. "Social capital II: determinants of economic connectedness." *Nature* 608.7921 (2022): 122-134.
- [25] Moro, Esteban, et al. "Mobility patterns are associated with experienced income segregation in large US cities." *Nature communications* 12.1 (2021): 1-10.
- [26] Soobader, Mah-jabeen, et al. "Using aggregate geographic data to proxy individual socioeconomic status: does size matter?." *American Journal of Public Health* 91.4 (2001): 632.
- [27] Geronimus, Arline T., and John Bound. "Use of census-based aggregate variables to proxy for socioeconomic group: evidence from national samples." *American Journal of Epidemiology* 148.5 (1998): 475-486.
- [28] McPherson, Miller, Lynn Smith-Lovin, and James M. Cook. "Birds of a feather: Homophily in social networks." *Annual review of sociology* 27.1 (2001): 415-444.
- [29] Chetty, Raj, et al. "Where is the land of opportunity? The geography of intergenerational mobility in the United States." *The Quarterly Journal of Economics* 129.4 (2014): 1553-1623.
- [30] Amanda Ripley, Rekha Tenjarla. "The Geography of Partisan Prejudice." *The Atlantic*, Atlantic Media Company, 3 May 2021, <https://www.theatlantic.com/politics/archive/2019/03/us-counties-vary-their-degree-partisan-prejudice/583072/>.
- [31] Granovetter, Mark S. "The strength of weak ties." *American journal of sociology* 78.6 (1973): 1360-1380.

[32] Rajkumar, Karthik, et al. "A causal test of the strength of weak ties." *Science* 377.6612 (2022): 1304-1310.

[33] Abbasov, Timur. "Do Urban Parks Promote Racial Diversity? Evidence from New York City." (2020): 1-35.

Reviewer Reports on the First Revision:

Referees' comments:

Referee #1 (Remarks to the Author):

I was extremely impressed by this revision.

Having gone through each issue from the original report carefully, I found that in each instance the authors answered my (sometimes substantial) concerns with thoughtfulness accompanied by overwhelming and compelling thoroughness.

I am happy with the current version and recommend publication.

Referee #3 (Remarks to the Author):

Report on "Human mobility networks reveal increased segregation in large cities"

My main concern with the prior version of the paper was that "there already exists a large body of work looking at similar outcomes with similar data and the additional insight from the paper at hand is not quite novel enough for a top general interest journal".

Reading the responses to my more detailed concerns has not changed this assessment and in fact, solidified it.

As an example, the response to my earlier concerns goes into detail in how their work differs from earlier work.

Several pages show that Athey et al. measure the extent to which different types of people visit different types of establishments. In contrast, the paper here shows whether people spend time in the same locations at the same time. This is perhaps a small improvement in terms of implementation.

The paper points out that a key difference to Abbiasov's study is that Abbiasov focuses on interactions of different types of people within a neighborhood, while the paper at hand focuses on differences across neighborhoods.

My original comment was not that the paper at hand is identical to earlier work. It is not and there are differences that are valuable and lead to some additional insights. The issue is that the differences relative to earlier work are not large enough to constitute a big enough contribution to be publishable in Nature. Again, reading through the authors' responses has only solidified this conclusion.

To illustrate this further, the authors show that their exposure measure is closely related to Facebook connections at the zip code and county level. They argue that this shows the value of measuring exposure the way it is done in the paper as it correlates with other meaningful social connections among individuals.

The paper at hand argues that one of its main contributions is to show that "citywide segregation is significantly reduced when city hubs are positioned to bridge diverse neighborhoods".

While not the same, the level of novelty to this insight is similar to that in Bailey et al, cited in a footnote in a different context, who pointed to transit lines (rather than city hubs) affecting social connections (rather than exposure to other people). This paper was published in the Journal of Urban Economics, in line with the extent of its contribution. I see the current paper at a similar level of novelty in contribution.

In short the paper introduces a measure of exposure of different type of individuals to each other and argues that such exposure is lower in bigger cities and largely driven by larger venue segregation. The measure differs slightly from a large number of other papers who have worked

with similar data to construct similar measures to answer similar questions. As such, it contributes to a large literature measuring different aspects of interactions and exposure within and across geographies and trying to understand its determinants. While potentially improving on this prior work in several ways, especially on the measurement front, the additional insights gained do not, in my view, rise to the bar for publication at a top general interest journal like Nature.

Author Rebuttals to First Revision:

Responses to Referee 3

My main concern with the prior version of the paper was that “there already exists a large body of work looking at similar outcomes with similar data and the additional insight from the paper at hand is not quite novel enough for a top general interest journal”.
Reading the responses to my more detailed concerns has not changed this assessment and in fact, solidified it.

Our key contributions are (a) refuting the cosmopolitan mixing hypothesis, which is a widely accepted theory about urban life; (b) explaining why it falls short; and (c) demonstrating the need for careful urban design to transform cities into the 'intermixing zones' that they were long assumed to be.

None of the prior work cited by Referee 3 addresses the cosmopolitan mixing hypothesis. Although Athey et al. do briefly take it on, as we explained in our previous response, they arrive at the wrong conclusion because they use a time-insensitive measure of exposure. Athey et al. write that:

“Experienced isolation is relatively lower in MSAs with higher population density... consistent with the fact that in dense areas residents from different neighborhoods are less separated by physical space, and may reflect the role of urban amenities such as parks and public facilities in facilitating diverse interactions (Jacobs 1961).”

Athey et al. reach the wrong conclusion because their measure of exposure does not account for visit timings. We conduct a new experiment, below, which demonstrates why time is crucial in measuring exposure.

Several pages show that Athey et al. measure the extent to which different type of people visit different types of establishments. In contrast, the paper here shows whether people spend time in the same locations at the same time.

This is perhaps a small improvement in terms of implementation.

A time-sensitive measure of exposure is a major contribution for two reasons:

(1) Time is the key ingredient needed to refute the cosmopolitan mixing hypothesis.

In our revision, we have conducted a new ablation study (copied below; **Supplementary Figure S66, subfigure b**) in which we re-calculate our metric of exposure segregation using Athey et al.'s time-insensitive measure of exposure (visits to the same 153m x 153m Geohash7 at any time). We show that, using this less accurate metric, it is no longer possible to reject the cosmopolitan mixing hypothesis.

Time is central to our findings because visit timings themselves are highly segregated. We provide two illustrative examples in **Supplementary Figure S63** (copied below).

(2) Accounting for time is computationally challenging, whereas a time-insensitive measure is trivial to compute.

Calculating time-insensitive Geohash7 exposures, as Athey et al. does, has a time complexity of $O(n)$, where n is the number of pings. Thus it is possible to calculate Athey's measure of segregation in less than 24 hours. By contrast, calculating a time-sensitive measure of exposure has an $O(n^2)$ time complexity. A naive implementation would necessitate on the order of ~ 10 years of compute time.

We built a large-scale system to efficiently construct this network, leveraging K-D trees and multi-core parallelization. Our system allows us to efficiently construct this network using a single supercomputer (with 12TB RAM, and 288 cores) in under a week. We have expanded our discussion of these computational challenges in our revision (Methods M2, "Constructing exposure network").

(a) Prior Work's Segregation Measure

(b) Our Segregation Measure (time-insensitive, Geohash7 exposures)

(b) Our Segregation Measure (time-sensitive exposures, < 50 meters in < 5 minutes)

Supplementary Figure S66: Our time-sensitive and high-resolution measure of exposure enables us to reject the cosmopolitan mixing hypothesis. We analyze the consequences of different measures of exposure by reproducing Athey et al.'s assessment of the cosmopolitan mixing hypothesis³². Athey et al. estimate the correlation between population density and their racial segregation metric (“experienced isolation”), after controlling for residential (racial) segregation. They do so by correlating the residuals of their racial segregation metric with the residuals of population density, where the residuals are derived from regression on (racial) residential segregation controls. We extend this analysis to include two additional segregation metrics, revealing that a time-sensitive measure of exposure is necessary to reject the cosmopolitan mixing hypothesis (a) We first consider Athey et al.’s original racial segregation metric (“experienced isolation”, which considers two people exposed to each other if they ever visited the same 153 x 153m Geohash7 grid cell within 4 months). Under Athey et al.’s metric, there is no significant correlation between population size and racial segregation (Spearman Corr. -0.04, $p = 0.49$), nor between population density and experienced segregation (Spearman Corr. 0.00, $p = 0.98$). Athey et al.’s robustness checks further corroborate this point, showing no significant association between population density and segregation when MSAs are unweighted by population size (Athey et al. Supplementary Table S8). (b) We then consider a time-insensitive variant of our own racial exposure segregation metric (Supplementary Figure S34), which uses Athey et. al.’s definition of exposure (any two people who visited the same 153 x 153m Geohash7 grid cell are considered exposed). Here, we similarly find that there is no significant correlation between population size and racial segregation (Spearman Corr. 0.06, $p = 0.23$), nor between population density and experienced segregation (Spearman Corr. 0.03, $p = 0.60$) (c) Finally, when we use our own time-sensitive and high-resolution racial exposure segregation metric, we find positive and statistically significant relationships between racial exposure segregation and population size (Spearman Corr. 0.31, $p < 1 \times 10^{-9}$) as well as population density (Spearman Corr. 0.16, $p < 0.001$). This comparison shows that precisely measuring exposure is necessary to reject the cosmopolitan mixing hypothesis.

Supplementary Figure S63: Temporal heterogeneity in exposure participants at the same location: illustrative examples. SES of exposure participants is plotted as a function of calendar month (top) and hour of day (bottom). Top: a high-end hotel in the Lake Tahoe region of California. SES is highest during March due to high hotel demand (March is prime ski season in North Lake Tahoe). SES is lowest in November (weather is cold, but ski resorts are either closed or have little snow). Bottom: A multi-story building in the Manhattan Financial District. SES is lowest during daytime due to out-of-neighborhood visitors (there is a fast food restaurant, Papaya Dog, on the ground floor). SES is highest during nighttime hours due to exposures between residents (the cost of living in the Financial District is extremely high).

To illustrate this further, the authors show that their exposure measure is closely related to Facebook connections at the zip code and county level. They argue that this shows the value of measuring exposure the way it is done in the paper as it correlates with other meaningful social connections among individuals. The paper at hand argues that one of its main contributions is to show that “citywide segregation is significantly reduced when city hubs are positioned to bridge diverse neighborhoods”. While not the same, the level of novelty to this insight is similar to that in Bailey et al, cited in a footnote in a different context, who pointed to transit lines (rather than city hubs) affecting social connections (rather than exposure to other people).

In short the paper introduces a measure of exposure of different type of individuals to each other and argues that such exposure is lower in bigger cities and largely driven by larger venue segregation.

The measure differs slightly from a large number of other papers who have worked with similar data to construct similar measures to answer similar questions.

This example does not support the referee's argument.

Bailey et al. are using entirely different data (online social networks vs. our real-world physical exposure networks) to construct a different measure (edge strength vs. segregation) to answer a different question (concerning the effect of public transit availability vs. city hubs).

Online social network data has been available since the dawn of the internet, and the novelty of analyzing an existing online social network (Facebook in Bailey et al.) is completely different from the novelty of constructing a network of real-world human encounters from cutting edge mobility data.

Moreover, it is unsurprising that public transit from Zipcode A to Zipcode B increases the likelihood of friendships between Zipcode A and Zipcode B, and it is also irrelevant to the cosmopolitan mixing hypothesis. It is far more surprising that the cosmopolitan mixing hypothesis is incorrect, and that properly-located city hubs can overcome the strong (and previously undetected) forces of socioeconomic homophily in large cities.